# Parameter Efficient Fine-tuning via Explained Variance Adaptation

**Fabian Paischer**[1,3*]    **Lukas Hauzenberger**[1*]    **Thomas Schmied**[1]
**Benedikt Alkin**[1,3]    **Marc Peter Deisenroth**[2]    **Sepp Hochreiter**[1,4]

[1] ELLIS Unit, LIT AI Lab, Institute for Machine Learning, JKU Linz, Austria
[2] University College London
[3] EMMI AI, Linz
[4] NXAI GmbH, Linz, Austria
`paischer@ml.jku.at`

## Abstract

Foundation models (FMs) are pre-trained on large-scale datasets and then fine-tuned for a specific downstream task. The most common fine-tuning method is to update pretrained weights via low-rank adaptation (LoRA). Existing initialization strategies for LoRA often rely on singular value decompositions (SVD) of gradients or weight matrices. However, they do not provably maximize the expected gradient signal, which is critical for fast adaptation. To this end, we introduce **E**xplained **V**ariance **A**daptation (EVA), an initialization scheme that uses the directions capturing the most activation variance, provably maximizing the expected gradient signal and accelerating fine-tuning. EVA performs incremental SVD on minibatches of activation vectors and selects the right-singular vectors for initialization once they converged. Further, by selecting the directions that capture the most activation-variance for a given rank budget, EVA accommodates adaptive ranks that reduce the number of trainable parameters. We apply EVA to a variety of fine-tuning tasks as language generation and understanding, image classification, and reinforcement learning. EVA exhibits faster convergence than competitors and achieves the highest average score across a multitude of tasks per domain while reducing the number of trainable parameters through rank redistribution. In summary, EVA establishes a new Pareto frontier compared to existing LoRA initialization schemes in both accuracy and efficiency.

## 1 Introduction

Foundation models (Bommasani et al., 2021, FMs) are usually trained on large-scale data and then fine-tuned towards a particular downstream task. This training paradigm has led to significant advances in the realm of language modeling (OpenAI, 2023; Touvron et al., 2023a; Reid et al., 2024), computer vision (Dehghani et al., 2023; Oquab et al., 2023), and reinforcement learning (Brohan et al., 2023; Zitkovich et al., 2023). With an increasing number of model parameters, fine-tuning (FT) becomes prohibitively expensive. This results in the need for efficient alternatives to fine-tuning *all* parameters of the pre-trained model.

Parameter-efficient fine-tuning (PEFT) approaches are commonly used as an efficient alternative to full fine-tuning (FFT). They modify the pre-trained model by introducing a small number of new trainable parameters, while the pre-trained weights remain frozen. A particularly successful approach, LoRA (Hu et al., 2022), introduces new weights in the form of a low-rank decomposition for each

---

*Equal contribution

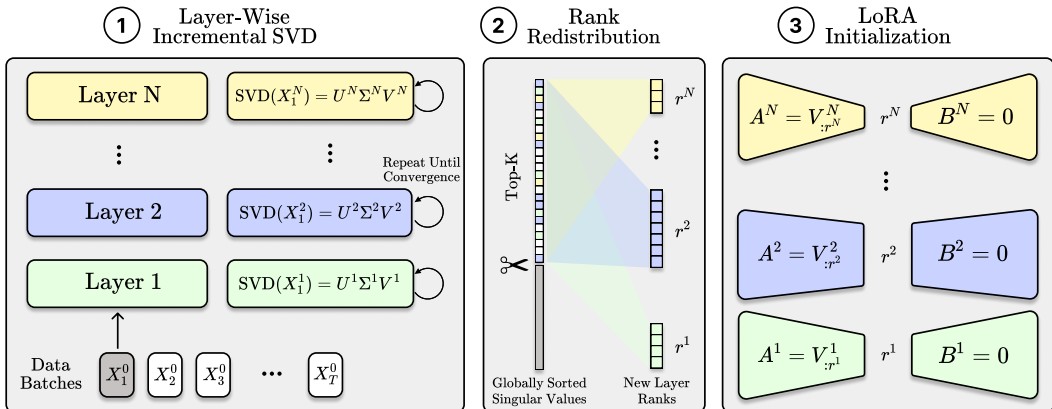

Figure 1: **Left:** We perform incremental SVD on activation vectors for the first $T$ minibatches. **Middle:** We globally sort all right-singular vectors according to their explained variance given by their respective normalized singular values and only keep the top-k. **Right:** We allocate the top-k vectors as initialization for $\boldsymbol{A}$ and continue the standard LoRA fine-tuning procedure.

weight matrix in the pre-trained model. After training, the new weights can be readily merged into the pre-trained weights without any additional inference latency. Recent research has explored various extensions of LoRA, such as different initialization schemes and adaptive rank allocation (see Table 1). Most of them rely on SVD-based approaches on either model weights or gradients. However, these approaches do not optimally maximize the expected gradient signal at the beginning of fine-tuning, still resulting in potentially slow convergence.

We propose **E**xplained **V**ariance **A**daptation (EVA), a method designed to provably maximize the expected gradient signal at the onset of fine-tuning. This optimal initialization is achieved by performing incremental Singular Value Decomposition (SVD) on activation vectors derived from minibatches of the downstream data. Upon convergence of this procedure, we populate the LoRA matrices with the resulting right-singular vectors. These vectors represent the projection onto the principal components, thereby capturing the directions that preserve activation variance. To ensure this maximization of the expected gradient signal within a fixed rank budget, the right-singular vectors are sorted by their explained variance. This process yields an adaptive rank allocation, computed at the beginning of fine-tuning, which assigns greater complexity (i.e., higher rank) to weights where the variance is distributed across more components.

We demonstrate the benefits of EVA on a variety of downstream tasks, namely language generation and understanding, image classification, and reinforcement learning (RL). EVA consistently improves average performance across a multitude of tasks in each domain compared to LoRA and other recently proposed initialization or rank redistribution methods. In addition, we demonstrate that the additional computational overhead for initialization is negligible and it is mostly invariant with respect to the batch size and order, verifying its robustness. Moreover, EVA exhibits improved convergence compared to other initialization methods, and our rank redistribution reduces the number of trainable parameters, since ranks are usually redistributed from higher-dimensional feedforward weights to lower-dimensional attention weights. Overall, we demonstrate that EVA is pareto dominant to competitors, as our rank redistribution reduces the number of trainable parameters while usually improving performance. Our contributions are as follows.

- We propose EVA, a novel data-driven initialization scheme for LoRA that uses incremental SVD on minibatches of activation vectors.

- We propose a data-driven heuristic for adaptive rank allocation to provably maximize the expected gradient signal for a given rank budget.

- We demonstrate pareto-dominance of EVA compared to other initialization schemes across a variety of different domains.

Table 1: Comparison of EVA to existing initialization schemes for LoRA. Existing works focus on initialization *or* adaptive rank allocation. EVA **combines** data-driven initialization with adaptive rank allocation to enhance convergence and downstream performance.

| Method | Initialization | Adaptive ranks |
|---|---|---|
| LoRA (Hu et al., 2022) | Random | ✗ |
| AdaLoRA (Zhang et al., 2023a) | Random | ✓ |
| PiSSA (Meng et al., 2024) | Weight-driven | ✗ |
| MiLoRA (Wang et al., 2024a) | Weight-driven | ✗ |
| OLoRA (Büyükakyüz, 2024) | Weight-driven | ✗ |
| LoRA-GA (Wang et al., 2024b) | Data-driven | ✗ |
| CorDA (Yang et al., 2024) | Data-driven | ✗ |
| EVA (Ours) | Data-driven | ✓ |

## 2   Related Work

**LoRA** (Hu et al., 2022) has sparked widespread interest in leveraging low-rank decompositions for fine-tuning due to its simplicity. Following the success of LoRA, several other variants have been proposed (Kopiczko et al., 2024; Zi et al., 2023; Babakniya et al., 2023; Dettmers et al., 2023; Li et al., 2023; Nikdan et al., 2024; Liu et al., 2024a; Zhang et al., 2023a; Hayou et al., 2024; Chavan et al., 2023). The variants most similar to EVA are CorDA (Yang et al., 2024) and LoRA-GA (Wang et al., 2024b), which are data-driven but do not leverage rank redistribution. Both rely on subsampling training data to estimate either gradients or input-output correlations for initialization. In contrast, EVA provides a variance-optimal initialization that maximizes the expected gradient signal, unified with rank redistribution. Rank redistribution approaches learn gates to switch ranks on/off during fine-tuning (Liu et al., 2024b; Meo et al., 2024) or different adapters with different ranks (Valipour et al., 2023). In contrast, our data-driven heuristic allows redistributing ranks prior to fine-tuning.

**Initialization of LoRA matrices** Common initialization schemes for neural networks (He et al., 2015; Glorot & Bengio, 2010) were designed to stabilize deep neural network training based on activation functions and depth. In the context of PEFT, Hu et al. (2022) and Liu et al. (2022) explored data-driven initialization by pre-training on a different task first, or by unsupervised pre-training on the task at hand. Similarly, Nikdan et al. (2024) utilize a warm-up stage in LoRA fine-tuning, where gradients with respect to LoRA weights are used to initialize a sparse matrix for sparse adaptation (Sung et al., 2021). Alternatively, Babakniya et al. (2023) initialize the LoRA matrices using SVD on the weight matrices obtained after a few steps of full fine-tuning. Weight-driven initializations (Meng et al., 2024; Büyükakyüz, 2024) leverage information from the pre-trained weights for initialization. Current data-driven initialization schemes consider either gradients (Wang et al., 2024b) or input-output correlations (Yang et al., 2024); however neither of them yields optimality with respect to the expected gradient signal. Similar initialization schemes to EVA were proposed for training deep networks from scratch (Mishkin & Matas, 2016; Krähenbühl et al., 2016).

**Increasing efficiency of LoRA** Several works have investigated how to improve the efficiency of LoRA fine-tuning. Kopiczko et al. (2024) decrease the memory complexity by keeping both $A$ and $B$ frozen while only training newly introduced scaling vectors. This way, only random seeds for initializing $A$ and $B$ need to be stored. Another prominent approach is quantization (Dettmers et al., 2022), which has been successfully combined with LoRA (Dettmers et al., 2023). Building on this, other variants of LoRA are also compatible with quantization (Nikdan et al., 2024; Meng et al., 2024). Initialization has also been shown to improve the fine-tuning of quantized models (Li et al., 2023).

## 3   Method

Our aim is to provide a data-driven initialization for LoRA weights that aligns the parameter update space with directions that capture the most variance in activations. Hence, for any downstream task using these activations, the update space is biased toward the most *informative* directions, providing better starting conditions. We first briefly explain LoRA in Section 3.1. Then, we explain the two essential steps conducted in EVA, namely (i), computing a variance optimal initialization for the

LoRA matrices via incremental SVD on activation vectors (Section 3.2) and (ii), adaptive assignment of ranks across layers to maximize the expected gradient signal for a given rank budget (Section 3.5).

## 3.1 Low-Rank Adaptation (LoRA)

LoRA adds new trainable weights that are computed using an outer product of low-rank matrices (Hu et al., 2022). This is motivated by the low intrinsic dimensionality of language models (Aghajanyan et al., 2021) and relies on the assumption that the gradients during fine-tuning are also of low rank (Gur-Ari et al., 2018; Zhang et al., 2023b; Gauch et al., 2022). Let $\boldsymbol{x} \in \mathbb{R}^{d \times 1}$ be the input to a pre-trained weight matrix $\boldsymbol{W} \in \mathbb{R}^{k \times d}$. Then, LoRA introduces new weight matrices $\boldsymbol{A}$ and $\boldsymbol{B}$ as a low-rank decomposition $\boldsymbol{h} = \boldsymbol{W}\boldsymbol{x} + \boldsymbol{B}\boldsymbol{A}\boldsymbol{x}$, where $\boldsymbol{B} \in \mathbb{R}^{k \times r}$ and $\boldsymbol{A} \in \mathbb{R}^{r \times d}$. The rank $r$ is a hyperparameter with $r \ll min(k, d)$. During fine-tuning, $\boldsymbol{W}$ remains frozen while $\boldsymbol{A}$ and $\boldsymbol{B}$ are updated. Usually, $\boldsymbol{B}$ is initialized with zeros and $\boldsymbol{A}$ at random, so that fine-tuning starts from the pre-trained model. In addition, a hyperparameter $\alpha$ is used to scale $\boldsymbol{B}\boldsymbol{A}\boldsymbol{x}$ by $\frac{\alpha}{r}$.

## 3.2 Variance-optimal initialization of Low-Rank Adaptation

For an effective initialization of $\boldsymbol{A}$ that is optimal with respect to propagated activation variance, we utilize incremental SVD (Ross et al., 2008) on minibatches of activation vectors $\boldsymbol{X} \in \mathbb{R}^{b \times d}$ (see Figure 1, left). This process involves collecting activation batches $\boldsymbol{X}^i \in \{\boldsymbol{X}^1, ..., \boldsymbol{X}^N\}$ for $N$ selected pre-trained weight matrices $\boldsymbol{W}^i \in \{\boldsymbol{W}^1, ..., \boldsymbol{W}^N\}$. Naively, we could simply collect batches of activations and stack them into a single matrix and perform SVD. However, this results in excessive memory overhead, as we usually deal with large datasets and models. To reduce memory requirements, we incrementally update $\boldsymbol{V}^i_{:r,:}$ following the approach of Ross et al. (2008), which is based on the sequential Karhunen-Loeve algorithm (Levy & Lindenbaum, 2000). This process is independent of the dataset size; therefore, the computation of the singular values and their respective vectors is constant in time and memory complexity. For each activation batch $\boldsymbol{X}^i$, we compute SVD to obtain the right-singular vectors $\boldsymbol{v}^i_{j,:}$ and their respective singular values $\sigma^i_j$. In practice, we compute truncated SVD (Halko et al., 2011), which is significantly faster. After each SVD computation, we update the right-singular vectors and singular values and check whether $\boldsymbol{V}^i$ has converged by cosine similarity $\text{cossim}(\boldsymbol{v}^{i,t-1}_{j,:}, \boldsymbol{v}^{i,t}_{j,:}) \geq \tau \quad \forall \quad 1 \leq j \leq r$. We illustrate the incremental SVD procedure applied to a sequence of data batches in Algorithm 2 and discuss the complexity of this procedure in Appendix F. Finally, by initializing $\boldsymbol{A}^i = \boldsymbol{V}^i_{:r,:}$ we obtain an optimal initialization for $\boldsymbol{A}$ with respect to the activation variance.

**Theorem 3.1.** *Let $\boldsymbol{X} \in \mathbb{R}^{b \times d}$ be a matrix of activation vectors obtained from a pretrained model, where $b$ is the number of samples and $d$ is the feature dimension. Suppose we wish to adapt a weight matrix $\boldsymbol{W} \in \mathbb{R}^{k \times d}$ using a low-rank update of the form $\Delta \boldsymbol{W} = B A$, where $\boldsymbol{B} \in \mathbb{R}^{k \times r}$, $\boldsymbol{A} \in \mathbb{R}^{r \times d}$, and $r \ll \min(k, d)$. Let $\boldsymbol{X} = \boldsymbol{U}\boldsymbol{\Sigma}\boldsymbol{V}^\top$ be the singular value decomposition (SVD) of the activation matrix with $\sigma_1 \geq \sigma_2 \geq \cdots \geq 0$ being the singular values of $\boldsymbol{\Sigma}$. Then the top $r$ right singular vectors $\boldsymbol{V}_{:r} \in \mathbb{R}^{d \times r}$ solve the following optimization problem:*

$$\boldsymbol{V}_r = \arg \max_{\boldsymbol{V} \in \mathbb{R}^{d \times r}, \boldsymbol{V}^\top \boldsymbol{V} = I} \text{Tr}(\boldsymbol{V}^\top \boldsymbol{X}^\top \boldsymbol{X} \boldsymbol{V}),$$

*and also minimize the Frobenius norm reconstruction error:*

$$\boldsymbol{V}_{:r} = \arg \min_{\boldsymbol{M} \in \mathbb{R}^{b \times d}, \text{rank}(\boldsymbol{M}) \leq r} \|\boldsymbol{X} - \boldsymbol{M}\|_F^2.$$

*Hence, $\boldsymbol{V}_{:r}$ forms the optimal basis for capturing the maximum variance of activations under a rank-$r$ projection.*

The minimization of the reconstruction error under the Frobenius norm in Theorem 3.1 is directly given by the Eckart-Young theorem (Eckart & Young, 1936). For details see Appendix H

## 3.3 Gradient Signal Amplification

We hypothesize that initializing LoRA weights along directions of high variance leads to stronger, more stable gradient signal. To theoretically verify this claim, we consider a simple feedforward layer $\boldsymbol{y} = (\boldsymbol{W} + \boldsymbol{B}\boldsymbol{A})\boldsymbol{x}$. The gradients w.r.t. $\boldsymbol{A}$ and $\boldsymbol{B}$ in this example are

$$\frac{\partial \mathcal{L}}{\partial \boldsymbol{B}} = \frac{\partial \mathcal{L}}{\partial \boldsymbol{y}} \boldsymbol{x}^\top \boldsymbol{A}^\top \quad \text{and} \quad \frac{\partial \mathcal{L}}{\partial \boldsymbol{A}} = \boldsymbol{B}^\top \frac{\partial \mathcal{L}}{\partial \boldsymbol{y}} \boldsymbol{x}^\top, \tag{1}$$

respectively. Using an explained variance optimal initialization of $\boldsymbol{A}$ ensures that LoRA updates are aligned with high-variance directions in activation space. This leads to a higher expected gradient magnitude as shown below.

**Theorem 3.2.** *Let $\Delta \boldsymbol{W} = \boldsymbol{B}\boldsymbol{A}$ be a low-rank adaptation to a pretrained weight matrix $\boldsymbol{W} \in \mathbb{R}^{k \times d}$, where $\boldsymbol{B} \in \mathbb{R}^{k \times r}$, $\boldsymbol{A} \in \mathbb{R}^{r \times d}$, and $r \ll \min(k, d)$. Let $\boldsymbol{x} \in \mathbb{R}^d$ be the activation input to this layer. Assume activations $\boldsymbol{x}$ are drawn from a distribution with covariance matrix $\boldsymbol{\Sigma} = \mathbb{E}[\boldsymbol{x}\boldsymbol{x}^\top]$. Then initializing $\boldsymbol{A}$ with the top right singular vectors of a sample activation matrix $\boldsymbol{X} \in \mathbb{R}^{b \times d}$ maximizes the expected squared gradient norm:*

$$\mathbb{E}\left[\left\|\frac{\partial \mathcal{L}}{\partial \boldsymbol{B}}\right\|_F^2\right] \propto \mathrm{Tr}(\boldsymbol{A}\boldsymbol{\Sigma}\boldsymbol{A}^\top).$$

We provide a proof for Theorem 3.2 in Appendix H. As a result, aligning $\boldsymbol{A}$ with high-variance directions (via SVD), leads to amplification gradient signals and enable more effective low-rank updates. Importantly, this is true for initialization of both $\boldsymbol{B}$ and $\boldsymbol{A}$. Following (Hayou et al., 2024), we initialize $\boldsymbol{A}$ in an explained variance optimal manner and set $\boldsymbol{B} = 0$, as this setup has favorable properties, such as usage of higher learning rates.

### 3.4 Connection to Neural Tangent Kernel and Generalization Error

To provide further theoretical insights into the effect of explained variance optimal initialization on the generalization error, we view the fine-tuning problem through the lens of the Neural Tangent Kernel (Jacot et al., 2018, NTK). For a network with parameters $\theta$, the NTK is defined as

$$K(\boldsymbol{x}, \boldsymbol{x}') = \nabla_\theta f_\theta(\boldsymbol{x})^\top \nabla_\theta f_\theta(\boldsymbol{x}'), \tag{2}$$

and its expectation over the fine-tuning data can be written as

$$\mathbb{E}\left[\nabla_\theta f_\theta(\boldsymbol{x})\nabla_\theta f_\theta(\boldsymbol{x})^\top\right] = \mathbb{E}\left[(\delta(\boldsymbol{x}) \otimes h(\boldsymbol{x}))(\delta(\boldsymbol{x}) \otimes h(\boldsymbol{x}))^\top\right], \tag{3}$$

where $\delta(\boldsymbol{x})$ denotes upstream gradients and $h(\boldsymbol{x})$ represents input activations of a weight matrix, and $\otimes$ denotes the Kronecker product. Under the simplifying assumptions that (i) $h(\boldsymbol{x})$ and $\delta(\boldsymbol{x})$ are weakly correlated, and (ii) the covariance of the upstream gradients $\delta(\boldsymbol{x})$ is approximately isotropic, the NTK simplifies to

$$\mathbb{E}\left[\nabla_\theta f_\theta(\boldsymbol{x})\nabla_\theta f_\theta(\boldsymbol{x})^\top\right] \approx \sigma_\delta^2 \mathbb{E}\left[h(\boldsymbol{x})h(\boldsymbol{x})^\top\right]. \tag{4}$$

Since we only consider the onset of fine-tuning, i.e. the time of initialization, these assumptions are reasonable. Therefore, EVA effectively approximates the empirical NTK's leading subspace. Directions associated with larger NTK eigenvalues $\lambda_i$ are learned faster and generalize more robustly, whereas directions with smaller $\lambda_i$ dominate the residual error and thus the kernel-regression generalization bound (Arora et al., 2019; Lee et al., 2019)

$$\varepsilon_{\mathrm{gen}} \propto \sum_i \frac{(\boldsymbol{u}_i^\top \boldsymbol{y})^2}{\lambda_i^2}, \tag{5}$$

where $\{\boldsymbol{u}_i, \lambda_i\}$ denotes the NTK spectrum and $\boldsymbol{y}$ represents the labels. Consequently, EVA approximately initializes the LoRA adapters along the dominant principal directions of the NTK, thereby minimizing the spectral tail of the NTK-based generalization error.

### 3.5 Adaptive Rank Allocation

The singular values provide an estimate of the amount of variance that each component in $\boldsymbol{V}_{:r,:}^i$ explains. Leveraging this insight, we can redistribute ranks across weight matrices of the pre-trained model such that the maximum amount of variance is explained for a given rank budget $l = Nr$. To achieve this, we sort right singular vectors obtained for all weight matrices according to their explained variance ratio (see Figure 1, middle)

$$\xi_j^i = \frac{\sigma_j^{i^2}}{(M-1)\|\boldsymbol{\sigma}^i\|_1}, \tag{6}$$

where $|| \cdot ||_1$ denotes the $\ell_1$ norm, $\boldsymbol{\sigma}^i$ is a vector containing all $r$ singular values, and $M$ is the total number of samples used for the incremental SVD. Note that $\xi$ is normalized for each weight matrix to ensure comparable ranges. Then, we take the top-$l$ entries of the globally sorted singular vectors and set the rank of each pre-trained weight based on how many of its singular vectors are contained in this selection (see Figure 1, right).

Additionally, we introduce a hyperparameter $\rho \in [1, \infty)$ that controls the uniformity of the rank distribution. $\rho$ determines the number of ranks that we compute during SVD and increasing $\rho$ allows for an increasingly heterogeneous rank distribution. Moreover, $\rho$ controls the maximum number of ranks that a weight matrix can receive. For each $\boldsymbol{W}^i$ we compute the top $r\rho$ components via incremental truncated SVD, resulting in $Nr\rho$ components in total. For redistribution, we then only use the top-$l$ components, according to their explained variance ratio $\xi_j^i$. This ensures that the number of total ranks used is the same for EVA compared to other LoRA variants. Setting $\rho = 1$, results in a uniform rank

---

**Algorithm 1** Fine-tuning via EVA

**Input:** FM $\psi(\cdot)$, $\rho$, rank $r$, dataset $\mathcal{D}$
1: **while** not all_converged($\psi$) **do**
2:     $\boldsymbol{X} \leftarrow \psi(\texttt{next}(\mathcal{D}))$          ▷ get activations
3:     $\boldsymbol{V}_{\text{new}}, \boldsymbol{\xi} \leftarrow$ Incremental-SVD($\boldsymbol{X}, \rho r$)
4:     **if** isclose($\boldsymbol{V}_{\text{old}}, \boldsymbol{V}_{\text{new}}$) **then**
5:         wrap_and_initialize($\boldsymbol{W}_j, \boldsymbol{V}_{\text{new}}$)
6:     **end if**
7:     $\boldsymbol{V}_{old} \leftarrow \boldsymbol{V}_{new}$
8: **end while**
9: redistribute_ranks($\psi, \boldsymbol{\xi}, \boldsymbol{V}_{\text{new}}$)
10: lora_finetune($\psi, \boldsymbol{X}$)

---

distribution as in LoRA, but initialized according to EVA. Therefore, $\rho$ provides us with the means to change the rank distribution in a controlled manner prior to fine-tuning at the initialization stage. In practice, we found that the redistribution converges for values of $\rho > 2$ (see Appendix G). Finally, we set $\boldsymbol{B} = 0$ and perform standard LoRA fine-tuning. In Algorithm 1 we provide pseudocode for EVA.

## 4 Experiments

First, we elaborate on implementation details of EVA in Section 4.1. Then, we show results for fine-tuning large language models (LLMs) on math and reasoning tasks in Section 4.2 and language understanding tasks in Section 4.3. In addition, we show results for image classification in Section 4.4 and decision-making tasks in Section 4.5. Finally, in Section 4.6 we demonstrate that the computational overhead induced by EVA on LoRA is negligible and that incremental SVD converges and is invariant to batch order and batch size.

### 4.1 Implementation Details

We follow the standard LoRA training procedure from Hu et al. (2022). Similarly to Kalajdzievski (2023), we found that LoRA training is very sensitive to the scaling parameter $\alpha$. Therefore, we set $\alpha = 1$ for all our experiments as we found this to be the most stable setting. For EVA with $\rho > 1$ we set $\alpha = \frac{r_{new}}{r_{old}}$ to preserve the scaling factor for different ranks. Following Zhang et al. (2023a), we apply EVA adapters to all pre-trained weight matrices except for the embedding layer. All models we used for fine-tuning are publicly available on the huggingface hub (Wolf et al., 2020). For the implementation of baselines, we utilize the widely used PEFT library (Mangrulkar et al., 2022). Across experiments, we highlight the highest scores in boldface and underline the second-highest.

### 4.2 Language Generation

We fine-tune five different LLMs, namely Llama-2-7B (Touvron et al., 2023b), Llama-3.1-8B (Dubey et al., 2024), Llama-3.1-70B, Gemma-2-9B (Rivière et al., 2024), and Gemma-2-27B on common sense reasoning benchmarks. We follow Liu et al. (2024a) and amalgamate a training set consisting of BoolQ (Christopher et al., 2019), PIQA (Bisk et al., 2020), SIQA (Sap et al., 2019), HellaSwag (Zellers et al., 2019), Winogrande (Sakaguchi et al., 2020), ARC-e and ARC-c (Clark et al., 2018) and OpenBookQA (Mihaylov et al., 2018). We apply all the methods listed in Table 1 to all five models, except LoRA-GA and CorDA, which we do not apply to Llama-3.1-70B and Gemma-2-27B, as it requires an excessive amount of computation for initialization (see Figure 5, left). We train all methods with rank $r = 16$ and a learning rate of $5e - 4$ for three random seeds. For Llama-3.1-70B, we leverage gradient checkpointing and the ZeRO optimizer (Rajbhandari et al., 2020) for optimizer state and gradient offloading. More details on the fine-tuning settings can be found in Appendix B.

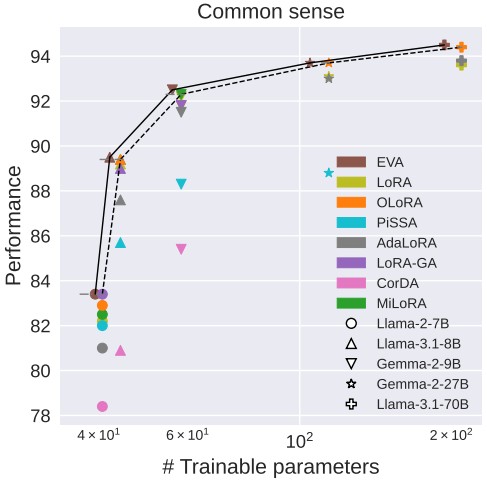
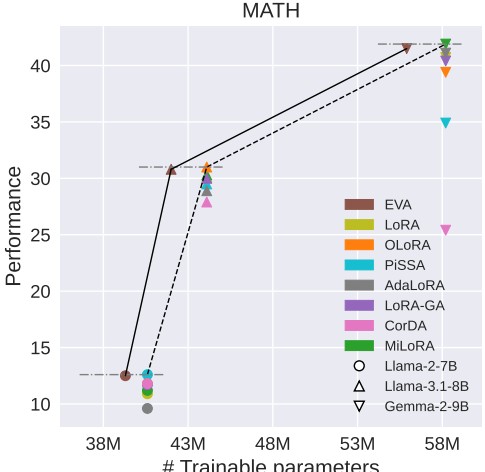

Figure 2: Performance of all methods on eight common sense reasoning tasks (left) and MATH after being finetuned on MetaMathQA (right). EVA reduces the number of trainable parameters while reaching performance on-par or better.

We present average performance for all eight common sense reasoning tasks in Figure 2. Across models, we found that EVA yields the highest performance while also significantly reducing the number of trainable parameters compared to all other LoRA-based methods (see Table 11 in Appendix B), resulting in an improved pareto front. For example, EVA applied to Llama-3.1-70B achieves the highest average score (94.5) while reducing the number of trainable parameters by more than 15M. We report the performance per task in Table 7 in Appendix B and also add a comparison to DoRA (Liu et al., 2024a) and EVA+DoRA, which combines EVA with DoRA. Although there is a fluctuation on a per-task basis, EVA-based methods consistently attain the highest average score across all tasks. Moreover, we conduct experiments where we add rank stabilization (Kalajdzievski, 2023), different learning rates for $A$ and $B$, and different values for $\alpha$ in Table 10 in Appendix B. EVA consistently yields the best results in these settings compared to LoRA. To demonstrate that EVA initialization starts closer

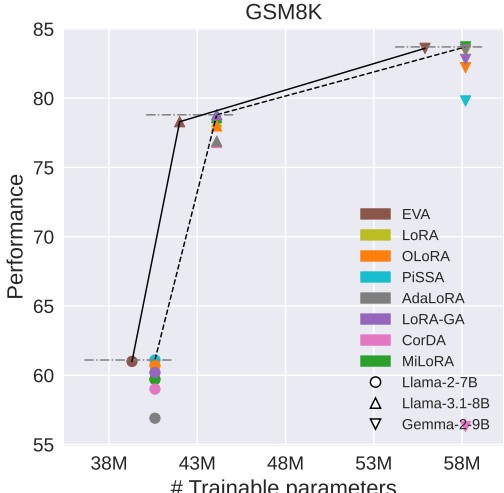

Figure 3: Performance of all methods for fine-tuning Llama-2-7B, Llama-3.1-8B, and Gemma-2-9B on GSM8K after fine-tuning on the MetaMathQA dataset.

to its final solution, we report the distance of EVA to the adapter weights after training compared to the distance of LoRA to the adapter weights after training for different weight matrices of the model in Table 6 (right). The CorDA baseline exhibited high seed sensitivity, prompting us to conduct a light hyperparameter search over the number of initialization examples in consultation with the CorDA authors. Despite these efforts, training performance collapsed for certain seeds, as evidenced in our results. Additionally, we provide results for leveraging the components that explain the *least* amount of variance in Table 12, which results in worse performance compared to EVA, and additional results for training with varying number of ranks for Llama-2-7B in Table 9. We find that across ranks and hyperparameters, EVA is consistently among the best performing methods.

For math fine-tuning experiments, we fine-tune Llama-2-7B, Llama-3.1-8B, and Gemma-2-9B on the MetaMathQA dataset (Yu et al., 2024) for one epoch with the same hyperparameters as for common sense reasoning tasks and evaluate them on MATH (Hendrycks et al., 2021) (see Figure 2, right) and

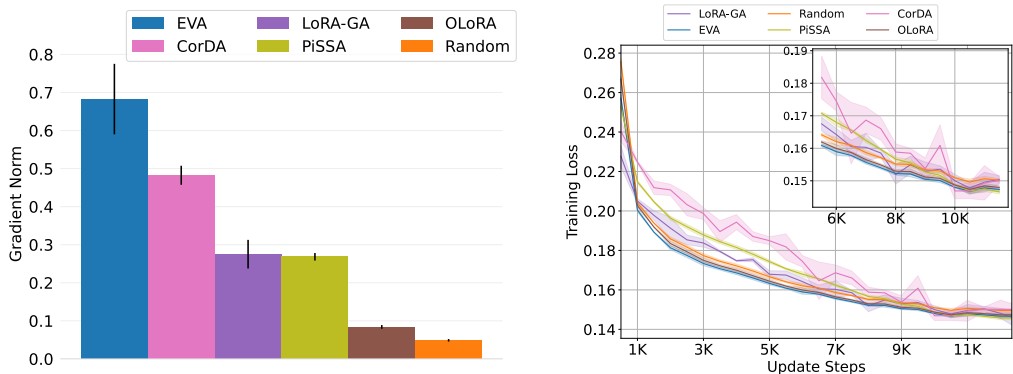

Figure 4: Gradient norm (**left**) and training loss (**right**) for fine-tuning Llama-3.1-8B on the Meta-MathQA dataset. We compare EVA to other initialization methods and random initialization (LoRA). We show mean and standard deviation across three random seeds.

GSM8K (Cobbe et al., 2021) (see Figure 3). We also report the performance of each method on each model and task, again including DoRA and EVA+DoRA, in Table 8 in Appendix B. EVA is pareto-dominant compared to all competitors on both datasets as it trains fewer parameters while resulting in on-par or improved performance. For example, EVA achieves the highest performance on the GSM8K dataset for Gemma-2-9B, while performance is on-par for Llama-2-7B and Llama-3.1-8B. In Figure 4 we show gradient norm and training loss for Llama-3.1-8B on the MetaMathQA dataset. We observe that EVA converges faster than competitors and exhibits the largest gradient norm. We provide additional loss curves in Figure 6. Furthermore, we provide a comprehensive overview on the effect of rank redistribution on different model types for both downstream tasks in Table 11. Our results indicate that the performance of adaptive rank allocation depends on a combination of the selected model and the downstream task. We further analyze the resulting rank distributions for different values of $\rho$ for Llama-2-7B and their effect on downstream performance in Appendix G. Finally, we provide additional results for Llama-2-7B on code fine-tuning tasks in Appendix B.

### 4.3 Language Understanding

We train RoBERTa$_{Large}$ (Liu et al., 2019) and DeBERTav3$_{Base}$ (He et al., 2023) on the GLUE benchmark (Wang et al., 2019). The GLUE benchmark comprises eight downstream tasks, such as natural language inference, or sentiment analysis. In addition to learning rate, we also search for different ranks within a maximal rank budget ($r \leq 16$). For further details on datasets, implementation, or hyperparameters, see Appendix C. We also add FFT as a baseline and report Matthew's correlation for CoLA, Pearson's correlation for STS-B, and accuracy for the remaining tasks in Table 2. EVA achieves the highest average score in all tasks for both RoBERTa$_{Large}$ and DeBERTav3$_{Base}$. Interestingly, DoRA usually only slightly improves over LoRA on low resource tasks (RTE, MRPC), while performing worse on high resource tasks (MNLI, QNLI, QQP, SST2). We also compare LoRA with EVA in Table 17 in Appendix C for different rank budgets, where EVA consistently improves over LoRA. We visualize the resulting rank distribution patterns for different GLUE tasks in Appendix C. More ranks are assigned to higher layers of the query, key, and value projections in self-attention, whereas the remaining weights often receive less ranks. This is a consistent pattern for both DeBERTav3$_{Base}$ and RoBERTa$_{Large}$ and is in line with the reduced number of trainable parameters for larger models.

### 4.4 Image Classification

We evaluate EVA on the VTAB-1K (Zhai et al., 2019) benchmark, which comprises 19 image classification tasks that are divided into natural images, specialized images (medical images and remote sensing), and structured images (e.g. object counting). We fine-tune a DINOv2-g/14 model (Oquab et al., 2023) that consists of around 1.1B parameters. For implementation details and hyperparameters see Appendix D. Our results are shown in Table 20 and we additionally report error bars in Table 21. EVA attains the best average accuracy across all tasks. Interestingly, EVA mainly

Table 2: Comparison of all methods for RoBERTa$_{\text{Large}}$ (top) and DeBERTav3$_{\text{Base}}$ (bottom) on GLUE tasks. We report mean and standard deviation of Matthew's correlation for CoLA, Pearson correlation for STS-B, matched accuracy for MNLI, and accuracy for remaining tasks. For CoLA, RTE, MRPC, and STS-B we average over five seeds and for the remaining tasks over three seeds.

| Method | MNLI | QNLI | QQP | SST2 | CoLA | MRPC | RTE | STS-B | Avg |
|---|---|---|---|---|---|---|---|---|---|
| FFT | 90.2 | 94.7 | **92.2** | **96.4** | 68.0 | 90.9 | 86.6 | 92.4 | 88.9 |
| LoRA | 90.7$_{\pm.1}$ | 94.8$_{\pm.1}$ | 92.0$_{\pm.0}$ | 96.2$_{\pm.3}$ | 69.1$_{\pm.5}$ | 91.1$_{\pm.6}$ | 88.1$_{\pm1.1}$ | 92.3$_{\pm.1}$ | 89.3 |
| AdaLoRA | 90.5$_{\pm.1}$ | 94.8$_{\pm.2}$ | 90.6$_{\pm.1}$ | 96.1$_{\pm.2}$ | 68.2$_{\pm.7}$ | 90.7$_{\pm.6}$ | 84.4$_{\pm.9}$ | 91.8$_{\pm.1}$ | 88.4 |
| PiSSA | 90.1$_{\pm.1}$ | 94.7$_{\pm.0}$ | 91.0$_{\pm.2}$ | 96.1$_{\pm.2}$ | 68.7$_{\pm1.3}$ | 90.4$_{\pm.6}$ | 87.6$_{\pm.5}$ | 92.5$_{\pm.3}$ | 88.9 |
| OLoRA | **90.9$_{\pm.1}$** | 95.0$_{\pm.1}$ | 92.0$_{\pm.2}$ | 96.3$_{\pm.3}$ | 69.0$_{\pm1.5}$ | 91.0$_{\pm1.0}$ | 87.9$_{\pm1.2}$ | 92.4$_{\pm.1}$ | 89.3 |
| LoRA-GA | 90.8$_{\pm.2}$ | 94.9$_{\pm.1}$ | 92.0$_{\pm.0}$ | 96.3$_{\pm.4}$ | 68.4$_{\pm1.9}$ | 91.0$_{\pm.2}$ | 87.0$_{\pm.4}$ | 92.3$_{\pm.3}$ | 89.1 |
| CorDA | 89.3$_{\pm.0}$ | 92.6$_{\pm.0}$ | 89.7$_{\pm.0}$ | 95.5$_{\pm.0}$ | 67.8$_{\pm1.0}$ | 90.1$_{\pm.9}$ | 86.5$_{\pm.8}$ | 91.8$_{\pm.2}$ | 87.9 |
| EVA | 90.8$_{\pm.1}$ | 95.0$_{\pm.2}$ | 92.1$_{\pm.1}$ | 96.2$_{\pm.1}$ | **69.5$_{\pm1.4}$** | 91.4$_{\pm.8}$ | **88.8$_{\pm1.2}$** | 92.6$_{\pm.1}$ | **89.6** |
| DoRA | 89.5$_{\pm.1}$ | 94.6$_{\pm.1}$ | 89.9$_{\pm.1}$ | 96.1$_{\pm.1}$ | 69.3$_{\pm.8}$ | 91.0$_{\pm.6}$ | 88.4$_{\pm1.2}$ | 92.4$_{\pm.1}$ | 88.9 |
| FFT | 90.1 | 94.0 | 92.4 | 95.6 | 69.2 | 89.5 | 83.8 | 91.6 | 88.3 |
| LoRA | 90.5$_{\pm.1}$ | 94.3$_{\pm.1}$ | 92.4$_{\pm.1}$ | 95.2$_{\pm.3}$ | 72.0$_{\pm1.3}$ | 91.4$_{\pm.7}$ | 88.9$_{\pm.5}$ | 91.7$_{\pm.1}$ | 89.6 |
| AdaLoRA | **90.8** | **94.6** | 92.2 | 96.1 | 71.5 | 90.7 | 88.1 | 91.8 | 89.5 |
| PiSSA | 90.1$_{\pm.3}$ | 94.1$_{\pm.1}$ | 91.8$_{\pm.1}$ | 95.8$_{\pm.1}$ | **72.7$_{\pm1.7}$** | 90.9$_{\pm.6}$ | 86.5$_{\pm1.2}$ | 91.6$_{\pm.2}$ | 89.2 |
| OLoRA | 90.5$_{\pm.1}$ | 94.4$_{\pm.1}$ | **92.6$_{\pm.1}$** | **96.2$_{\pm.2}$** | 72.0$_{\pm1.0}$ | 91.6$_{\pm.7}$ | 89.1$_{\pm.9}$ | **92.0$_{\pm.2}$** | 89.8 |
| LoRA-GA | 89.8$_{\pm.7}$ | **94.6$_{\pm.1}$** | 92.2$_{\pm.0}$ | 95.6$_{\pm.8}$ | 72.2$_{\pm.9}$ | 90.8$_{\pm.9}$ | 86.6$_{\pm1.1}$ | 90.5$_{\pm.6}$ | 89.0 |
| CorDA | 90.0$_{\pm.1}$ | 93.8$_{\pm.1}$ | 91.1$_{\pm.1}$ | 95.5$_{\pm.4}$ | 71.8$_{\pm1.2}$ | 89.6$_{\pm.5}$ | 83.9$_{\pm.3}$ | 91.1$_{\pm.2}$ | 88.3 |
| EVA | 90.6$_{\pm.1}$ | 94.4$_{\pm.1}$ | 92.4$_{\pm.0}$ | 96.2$_{\pm.2}$ | 72.5$_{\pm1.3}$ | 91.8$_{\pm.6}$ | **89.4$_{\pm.7}$** | 92.0$_{\pm.2}$ | **89.9** |
| DoRA | 89.0$_{\pm.2}$ | 94.1$_{\pm.1}$ | 88.0$_{\pm.1}$ | 94.6$_{\pm.4}$ | 70.3$_{\pm.5}$ | 91.9$_{\pm.6}$ | 87.8$_{\pm.7}$ | 91.8$_{\pm.1}$ | 88.4 |

improves over competitors on natural tasks, i.e., in-domain datasets. On out-of-distribution datasets, we find that FFT still performs better than most PEFT approaches.

## 4.5 Decision Making

We follow the single task fine-tuning experiments in Schmied et al. (2024) and fine-tune a Decision Transformer (Chen et al., 2021a, DT) on the Meta-World benchmark suite (Yu et al., 2020). Meta-World consists of a diverse set of 50 tasks for robotic manipulation, such as grasping, or pushing buttons. We split Meta-World according to Wolczyk et al. (2021) into 40 pre-training tasks (MT40) and 10 fine-tuning tasks (CW10). We pre-train a 12 M parameter DT on MT40 and fine-tune it on the CW10 holdout tasks. We report success rates and standard errors for each CW10 task in Table 23. We observe that EVA significantly reduces that gap between LoRA and FFT. Furthermore, combining EVA with DoRA improves upon DoRA and attains the best average performance across all tasks. We report results for different rank budgets in Table 24, as well as implementation details and hyperparameters in Appendix E.

## 4.6 Efficiency and convergence

We compare the computational overhead of EVA to competitors. In Figure 5 (left) we show the wall clock time as a fraction of the training time as well as memory requirements. For CorDA we use a sample size of 2560 as recommended by Yang et al. (2024). We observe that EVA with batch size of 16 requires only 0.7% of the training time for initialization, which is the fastest for data-driven initializations. In Figure 5 (right) we provide additional evidence that reducing the batch size for EVA results in the same initialization. Therefore, by reducing the batch size, we can reduce the overhead induced by EVA to 0.2%, resulting in one of the most efficient initializations. Furthermore, we also provide evidence that the incremental SVD is invariant to batch order and consistently converges for different batch sizes in Figure 12 (left), and Figure 11 in Appendix F, respectively.

## 5 Discussion and Limitations

**Low rank setup.** Due to computational constraints we mainly chose lower values for the LoRA rank ($r = 16$). Other initialization schemes (Yang et al., 2024; Meng et al., 2024; Wang et al., 2024b) usually rely on higher ranks ($r \geq 128$). However, for such values, EVA suffers from the computational overhead of repeated SVD computations. In Table 9 we provide results for fine-tuning

| Initialization | Method | Memory (GB) | % of Training |
|---|---|---|---|
| **Weight-driven** | PiSSA | - | 1.5 |
| | OLoRA | - | 0.1 |
| **Data-driven** | LoRA-GA$_{bs=8}$ | 56.95 | 2.4 |
| | CorDA$_{bs=1}$ | 55.64 | 4.5 |
| | EVA$_{bs=16}$ | 32.85 | 0.7 |
| | EVA$_{bs=8}$ | 29.39 | 0.3 |
| | EVA$_{bs=4}$ | 27.51 | 0.2 |

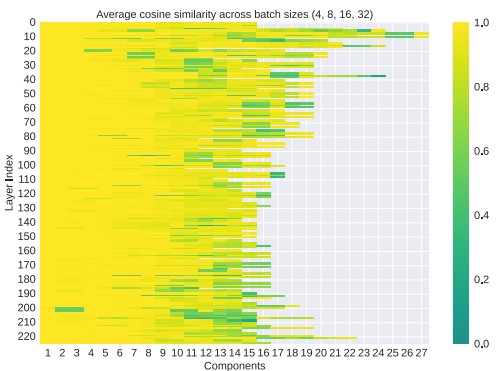

Figure 5: **Left:** Percentage of training time required for computing data-driven initializations for Llama-2-7B on a single A100 GPU on the common sense reasoning tasks. We report the maximum batch size and track peak memory usage. **Right:** Average cosine similarity between components after incremental SVD for different batch sizes. The components strongly correlate indicating that the SVD computation is mostly invariant to the batch size.

with ranks up to $r = 64$, which show that EVA works particularly well on lower-rank setups. This is an advantage for compute-constrained fine-tuning of larger models which is usually only feasible with smaller ranks. Finally, EVA assumes access to a fixed static downstream dataset which may not always be available.

**Effect of rank redistribution.** We provide additional results for the effect of rank redistribution in Appendix I. The results show that rank redistribution in combination with explained variance optimal initialization in principle performs best. We observed cases where there is no improvement in performance for rank redistribution (e.g. decision making). However, since rank redistribution in all our experiments decreased the number of trainable parameters, we recommend using it by default.

**What method performs well in which tasks?** We conducted fine-tuning experiments for 51 tasks and four domains and found that EVA usually performs best on average across multiple tasks per domain. Despite this, there is usually variation in the ranking of methods for single tasks, i.e. LoRA performed better on specialized images and FFT performed best on structured images. Therefore, there is no one algorithm that performs the best on every task, verifying that there is no free lunch (Wolpert & Macready, 1997).

**Reproducibility.** We provide the source code along with the submission (see Appendix A) to ensure reproducibility. In addition, we added support for EVA in the widely used PEFT library (Mangrulkar et al., 2022) to make it more accessible.

# 6 Conclusion and Broader Impact

We propose a novel method named Explained Variance Adaptation (EVA), extending the widely used LoRA with explained variance optimal initialization and rank redistribution to provably maximize the expected gradient signal. EVA performs incremental SVD on minibatches of activation vectors and redistributes ranks across weight matrices according to the amount of variance that they explain. We demonstrate performance gains of EVA over LoRA and initialization schemes thereof in a variety of domains, ranging from language to vision and RL. Moreover, EVA is more efficient than most existing initialization methods while reducing the number of trainable parameters. Our results demonstrate that EVA consistently achieves the highest average performance on a wide range of tasks across a variety of domains.

We believe that EVA can have a significant impact on future research on fine-tuning foundation models because it inherits all the benefits of LoRA while improving performance and reducing the number of trainable parameters at no significant additional cost. In the future, our aim is to additionally incorporate gradient information and exploring ways to enhance interpretability by relating different singular vectors to different forms of pre-trained knowledge. Another fruitful avenue would be combining EVA with mixture-of-experts training to enable more efficient fine-tuning.

## Acknowledgments and Disclosure of Funding

We acknowledge EuroHPC Joint Undertaking for awarding us access to Vega at IZUM, Slovenia, Karolina at IT4Innovations, Czech Republic, MeluXina at LuxProvide, Luxembourg, Leonardo at CINECA, Italy, MareNostrum5 at BSC, Spain. The ELLIS Unit Linz, the LIT AI Lab, the Institute for Machine Learning, are supported by the Federal State Upper Austria. We thank the projects FWF AIRI FG 9-N (10.55776/FG9), AI4GreenHeatingGrids (FFG- 899943), Stars4Waters (HORIZON-CL6-2021-CLIMATE-01-01), FWF Bilateral Artificial Intelligence (10.55776/COE12). We thank NXAI GmbH, Audi AG, Silicon Austria Labs (SAL), Merck Healthcare KGaA, GLS (Univ. Waterloo), TÜV Holding GmbH, Software Competence Center Hagenberg GmbH, dSPACE GmbH, TRUMPF SE + Co. KG. Fabian Paischer acknowledges travel support from ELISE (GA no 951847).

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

# A    Reproducibility Statement

The source code to reproduce the results collected in our work can be found at `https://github.com/ml-jku/EVA`.

# B    Natural language generation

We follow the experiments conducted in Hu et al. (2023) and fine-tune Llama-2-7B, Llama-3.1-8B, Gemma-2-9B, Gemma-2-27Band Llama-3.1-70B on 8 common sense reasoning tasks with Qa-style prompts. We keep the original prompt templates unchanged except for two minor modifications: For BoolQ we prepend the passage field before the question, and for WinoGrande we add a line "Answer format:..." analogous to the other prompts. As done by Hu et al. (2023) and Liu et al. (2024a) we perform joint fine-tuning on all 8 tasks. We furthermore evaluate the pre-trained models mentioned above on the mathematical reasoning tasks GSM8K (Cobbe et al., 2021) and Math (Yu et al., 2024) after fine-tuning on MetaMathQA (Yu et al., 2024) as done in Meng et al. (2024). We keep the original prompt template for fine-tuning and evaluation. For all datasets, we performed fine-tuning for one epoch. For training Llama-3.1-70B, we use 4-bit quantization of the base model and training of adapter weights in bfloat16, as recommended in Dettmers et al. (2023).

## B.1    Implementation details

For fine-tuning our code base leverages PEFT implementations of adapter methods LoRA, AdaLoRA, PiSSA, OLoRA, LoRA-GA, CorDA and DoRA. The initialization step for EVA is a custom implementation, but for fine-tuning we can reformulate EVA as a LoRA adapter leveraging the `rank_pattern` argument of `peft.LoraConfig`. For evaluation, we used scripts provided by the MetaMath github repository (Yu et al., 2024) for math reasoning tasks. For common sense reasoning, we make use of the lm evaluation harness project (Gao et al., 2024) and define custom tasks using the fine-tuning prompts. For the SVD computation for joint fine-tunine on the common sense reasoning tasks, we experiment with random and stratified sampling of examples from the 8 tasks and do not notice a difference in performance. All training and evaluation runs for Llama-2-7B were performed on 4 A100 GPUs. The runs for Llama-3.1-8B and Gemma-2-9B utilized two different nodes, one with 4 A100 GPUs and one with 4 H200 GPUs.

Table 4: hyperparameters for finetuning on common sense reasoning and math reasoning

| Training | |
|---|---|
| Optimizer | AdamW |
| Weight Decay | 0.0 |
| Lora Dropout | 0.0 |
| Batch Size | 32 |
| #Epoch | 1 |
| LR Schedule | Linear |
| Warmup ratio | 0.03 |
| Label Smooth | 0.0 |
| Learning Rate | 5e-4 |
| LoRA Dim | 16 |
| LoRA $\alpha$ | 1 |
| Batch Size SVD (EVA) | 16 |
| $\tau$ | 0.99 |
| Inference | |
| Beam Size | 1.0 |
| Length Penalty | 1.0 |
| repetition penalty | 1.0 |

## B.2    Hyperparameter search

The results reported on language generation tasks in Table 7 and Table 8 are the best setting based on a grid search over different learning rates. We apply adapters to all linear layers including the language modeling head. Furthermore, we set $\alpha = 1$ for all our experiments. We use AdamW with weight decay and a linear learning rate schedule with warm-up. We train for 1 epoch and use the final checkpoint for evaluation. All hyperparameters are summarized in Table 4. As mentioned in 4.2 we tuned the number of samples for initialization for CorDA after consulting with the CorDA authors. Specifically we increased the number of samples from 256 to 2560 after observing weak fine-tuning performance.

Table 3: Prompt templates with examples (red) used for finetuning on common sense and math reasoning tasks.

| Dataset | Fine-tuning Data Template |
|---|---|
| BoolQ | Passage: Drinking in public – Drinking in public is most commonly accepted. After reading this passage, please answer the following question with true or false, question: can you drink on the street in china
Answer format: true/false
the correct answer is true |
| PIQA | Please choose the correct solution to the question: When boiling butter, when it's ready, you can
Solution1: Pour it onto a plate
Solution2: Pour it into a jar
Answer format: solution 1/solution2
the correct answer is solution2 |
| SIQA | Please choose the correct answer to the question: Carson relocated somewhere new. How would you describe Carson?
Answer1: mobile
Answer2: anxious
Answer3: lonely
Answer format: answer1/answer2/answer3
the correct answer is answer1 |
| HellaSwag | Please choose the correct ending to complete the given sentence: Playing drums: People are standing behind large drums. A man
Ending1: is playing a bag pipe.
Ending2: starts to play around the drums.
Ending3: begins playing a drum set.
Ending4: begins playing the drums.
Answer format: ending1/ending2/ending3/ending4
the correct answer is ending4 |
| WinoGrande | Please choose the correct answer to fill in the blank to complete the given sentence: Ian volunteered to eat Dennis's menudo after already having a bowl because _ despised eating intestine.
Option1: Ian
Option2: Dennis
Answer format: option1/option2
the correct answer is option2 |
| ARC-e & ARC-c | Please choose the correct answer to the question: Which factor will most likely cause a person to develop a fever?
Answer1: a leg muscle relaxing after exercise
Answer2: a bacterial population in the bloodstream
Answer3: several viral particles on the skin
Answer4: carbohydrates being digested in the stomach
Answer format: answer1/answer2/answer3/answer4
the correct answer is answer2 |
| OBQA | Please choose the correct answer to the question: The sun is responsible for
Answer1: puppies learning new tricks
Answer2: children growing up and getting old
Answer3: flowers wilting in a vase
Answer4: plants sprouting, blooming and wilting
Answer format: answer1/answer2/answer3/answer4
the correct answer is answer4 |
| MetaMathQA | Below is an instruction that describes a task. Write a response that appropriately completes the request.

### Instruction:
What is the value of the cosine of 90 degrees?

### Response:
s $\\boxed{0}$.The answer is: 0 |

Table 5: Distance between final adapters trained with LoRA or EVA. We report spectral norm ($\sigma$) and average cosine similarity (cos) for Llama-2-7B, Llama-3.1-8B, and Llama-3.1-70B. Our results demonstrate that the effect of different initializations are massive, as the final adapters converge to entirely different solutions, which is indicated by large $\sigma$ and cos around zero.

| Model | Query | | Key | | Value | | Out | | Gate | | Up | | Down | |
|---|---|---|---|---|---|---|---|---|---|---|---|---|---|---|
| | cos | $\ell_2$ | cos | $\ell_2$ | cos | $\ell_2$ | cos | $\ell_2$ | cos | $\ell_2$ | cos | $\ell_2$ | cos | $\ell_2$ |
| Llama-2-7B | -0.01 | 4.98 | 0.00 | 5.00 | 0.01 | 4.00 | 0.00 | 4.05 | 0.00 | 6.64 | -0.00 | 3.67 | -0.00 | 4.02 |
| Llama-3.1-8B | -0.00 | 4.05 | -0.01 | 5.25 | -0.00 | 3.83 | -0.01 | 3.53 | -0.00 | 6.98 | 0.01 | 3.37 | -0.00 | 3.73 |
| Llama-3.1-70B | -0.01 | 7.57 | 0.00 | 7.52 | -0.00 | 6.70 | 0.01 | 5.63 | 0.00 | 12.81 | 0.00 | 6.30 | -0.00 | 6.33 |

Table 6: Distance between initialization of EVA and LoRA with their respective final adapters after training. We report spectral norm ($\sigma$) and average cosine similarity (cos) for Llama-2-7B, Llama-3.1-8B, and Llama-3.1-70B. Our results demonstrate that EVA initialization is a larger constituent of the final adapter than LoRA, indicating that EVA contains more information at initialization.

| Method | Model | Query | | Key | | Value | | Out | | Gate | | Up | | Down | |
|---|---|---|---|---|---|---|---|---|---|---|---|---|---|---|---|
| | | cos($\uparrow$) | $\sigma(\downarrow)$ | cos($\uparrow$) | $\sigma(\downarrow)$ | cos($\uparrow$) | $\sigma(\downarrow)$ | cos($\uparrow$) | $\sigma(\downarrow)$ | cos($\uparrow$) | $\sigma(\downarrow)$ | cos($\uparrow$) | $\sigma(\downarrow)$ | cos($\uparrow$) | $\sigma(\downarrow)$ |
| LoRA | Llama-2-7B | 0.51 | 3.85 | 0.48 | 4.08 | 0.60 | 3.10 | 0.59 | 3.09 | 0.44 | 5.27 | 0.62 | 2.83 | 0.61 | 3.13 |
| | Llama-3.1-8B | 0.51 | 3.46 | 0.47 | 3.96 | 0.59 | 2.93 | 0.61 | 2.73 | 0.35 | 5.88 | 0.60 | 2.58 | 0.59 | 2.98 |
| | Llama-3.1-70B | 0.45 | 4.62 | 0.42 | 5.07 | 0.52 | 3.86 | 0.61 | 3.17 | 0.39 | 6.74 | 0.61 | 3.11 | 0.62 | 3.13 |
| EVA | Llama-2-7B | 0.62 | 3.48 | 0.59 | 3.59 | 0.62 | 2.90 | 0.62 | 2.78 | 0.42 | 4.92 | 0.66 | 2.61 | 0.67 | 2.84 |
| | Llama-3.1-8B | 0.64 | 2.93 | 0.61 | 3.62 | 0.63 | 2.46 | 0.64 | 2.27 | 0.41 | 5.12 | 0.67 | 2.46 | 0.67 | 2.71 |
| | Llama-3.1-70B | 0.53 | 4.27 | 0.52 | 4.62 | 0.53 | 3.68 | 0.58 | 2.91 | 0.33 | 6.53 | 0.59 | 3.24 | 0.59 | 3.16 |

## B.3 Additional results

To demonstrate the effect of initialization, we measure the distance between the final adapters trained via LoRA and EVA and report cosine similarity and frobenius norm in Table 5. Our results demonstrate that depending on the initialization the two methods converge to substantially different solutions as there is almost no similarity between them. Furthermore, to highlight that EVA initialization starts closer to its final solution, we report the distance of EVA to the adapter weights after training compared to the distance of LoRA to the adapter weights after training for different weight matrices of the model in Table 6. In Figure 12, right we also visualize this finding for the three variants of Llama. Our results consistently indicate that (i) initialization has a tremendous impact on the final solution, and (ii) EVA initialization results in less information being learned than for LoRA, as it initializes the adapters to contain most of the information at initialization.

We present the per-task performance for the eight common sense reasoning tasks in Table 7. The respective standard deviations are shown in Table 14. Further, we show the results for all methods on the two math reasoning datasets in Table 8.

To investigate whether the observed improvement in performance depends on the rank, we conducted an additional experiment in which we vary the rank. Recall that in Section 4.2 we only used $r = 16$. Therefore, we conduct experiments for $r \in \{8, 16, 32, 64\}$ for Llama-2-7B on the eight common sense reasoning tasks. We report the results in Table 9. Our results demonstrate that EVA or EVA+DoRA are consistently the best performing methods for all ranks. Also, perhaps surprisingly, we find that a higher number of ranks does not always perform better. Our intuition is that the final performance strongly depends on the dataset size, i.e. the more parameters are introduced, the more likely the model tends to overfit.

We present additional loss curves for Llama-2-7B, Llama-3.1-8B, and Gemma-2-9B in common sense and math reasoning tasks in Figure 6. We find that EVA converges the fastest for all different models on the different tasks.

Another experiment we conduct is to apply recently proposed changes to the scaling factor and learning rate. In Table 10 we show results for changing the scaling factor to $\alpha = \frac{2r}{\sqrt{r}}$ which results in rank stabilization (Kalajdzievski, 2023). In addition, we present results for the regular setting $\alpha = 2r$ as proposed in Hu et al. (2022). Finally, we also show different learning rates for the two matrices $A$ and $B$ as proposed by Hayou et al. (2024). We make the following observations.

Table 7: Comparison of LoRA and DoRA to different initialization and rank re-distribution methods on NLG tasks. We report average performance across three seeds and respective standard deviation in Table 14. EVA+DoRA and EVA consistently attain the highest average performance across all tasks.

| Model | Method | BoolQ | PIQA | SIQA | HellaSwag | Winogrande | ARC-e | ARC-c | OBQA | Avg. |
|---|---|---|---|---|---|---|---|---|---|---|
| Llama-2-7B | LoRA | 67.2 | 83.9 | 82.0 | 94.7 | 84.0 | 87.8 | 74.1 | 84.0 | 82.2 |
| | AdaLoRA | 74.8 | 82.2 | 80.5 | 93.3 | 79.4 | 86.1 | 71.1 | 80.6 | 81.0 |
| | PiSSA | 62.6 | 84.8 | 81.2 | 94.5 | 84.8 | 87.8 | 74.8 | 85.4 | 82.0 |
| | MiLoRA | 65.0 | 84.8 | 82.3 | 94.9 | 84.5 | 88.2 | 74.9 | 85.3 | 82.5 |
| | OLoRA | 68.7 | 84.8 | 82.2 | 95.0 | 85.0 | 88.1 | 74.9 | 85.2 | 82.9 |
| | LoRA-GA | 69.0 | 85.6 | 82.3 | 95.0 | 85.0 | 88.7 | 75.9 | 85.8 | 83.4 |
| | CorDA | 68.7 | 80.4 | 79.7 | 91.7 | 77.8 | 82.5 | 67.9 | 78.4 | 78.4 |
| | EVA | 68.3 | 85.3 | 82.9 | 95.2 | 85.2 | 88.6 | 75.8 | 86.3 | 83.4 |
| | DoRA | 68.3 | 85.1 | 82.2 | 94.9 | 84.3 | 88.7 | 74.8 | 86.3 | 83.1 |
| | EVA+DoRA | 73.5 | 85.3 | 82.4 | 95.2 | 84.8 | 88.9 | 76.0 | 87.3 | 84.2 |
| Llama-3.1-8B | LoRA | 85.7 | 90.3 | 83.0 | 96.9 | 88.4 | 94.2 | 84.8 | 90.1 | 89.2 |
| | AdaLoRA | 83.9 | 89.5 | 81.7 | 96.2 | 86.3 | 93.7 | 82.7 | 86.8 | 87.6 |
| | PiSSA | 72.9 | 87.3 | 81.6 | 95.3 | 87.8 | 91.7 | 81.2 | 87.6 | 85.7 |
| | MiLoRA | 85.7 | 90.8 | 83.0 | 96.8 | 88.8 | 94.4 | 84.9 | 90.5 | 89.4 |
| | OLoRA | 86.0 | 90.4 | 83.9 | 97.0 | 88.6 | 94.5 | 84.7 | 90.3 | 89.4 |
| | LoRA-GA | 83.7 | 89.7 | 83.1 | 96.7 | 88.8 | 94.2 | 85.3 | 90.4 | 89.0 |
| | CorDA | 69.1 | 82.8 | 79.4 | 91.5 | 82.4 | 86.3 | 73.7 | 82.3 | 80.9 |
| | EVA | 85.3 | 90.4 | 83.4 | 97.0 | 89.0 | 94.4 | 86.0 | 90.3 | 89.5 |
| | DoRA | 86.2 | 90.8 | 83.4 | 96.9 | 88.6 | 94.3 | 84.9 | 89.4 | 89.3 |
| | EVA+DoRA | 85.8 | 90.8 | 83.9 | 97.1 | 89.2 | 94.4 | 85.9 | 90.5 | 89.7 |
| Gemma-2-9B | LoRA | 88.3 | 92.9 | 85.2 | 97.8 | 92.3 | 97.2 | 89.9 | 94.4 | 92.2 |
| | AdaLoRA | 87.3 | 91.8 | 84.6 | 97.3 | 91.3 | 97.0 | 90.0 | 92.6 | 91.5 |
| | PiSSA | 81.4 | 90.0 | 82.5 | 95.5 | 89.0 | 93.6 | 83.5 | 90.8 | 88.3 |
| | MiLoRA | 88.2 | 93.0 | 85.0 | 97.8 | 92.9 | 97.4 | 90.2 | 93.9 | 92.3 |
| | OLoRA | 87.7 | 92.5 | 85.2 | 97.5 | 92.5 | 96.6 | 88.7 | 93.7 | 91.8 |
| | LoRA-GA | 87.3 | 92.1 | 84.5 | 97.4 | 93.2 | 96.4 | 89.2 | 94.3 | 91.8 |
| | Corda | 63.1 | 87.2 | 82.2 | 94.0 | 87.9 | 93.7 | 84.4 | 90.8 | 85.4 |
| | EVA | 88.6 | 93.0 | 85.3 | 97.9 | 92.8 | 97.5 | 90.5 | 94.5 | 92.5 |
| | DoRA | 88.3 | 92.6 | 84.9 | 97.7 | 92.2 | 97.1 | 89.9 | 94.5 | 92.1 |
| | EVA+DoRA | 88.6 | 93.1 | 85.1 | 97.9 | 92.5 | 97.3 | 89.6 | 94.8 | 92.4 |
| Gemma-2-27B | LoRA | 89.0 | 93.6 | 85.9 | 98.0 | 93.6 | 97.5 | 92.1 | 95.2 | 93.1 |
| | AdaLoRA | 89.6 | 93.7 | 85.2 | 97.9 | 93.0 | 97.7 | 92.1 | 94.9 | 93.0 |
| | PiSSA | 82.0 | 89.9 | 82.4 | 95.7 | 90.5 | 93.8 | 84.7 | 91.3 | 88.7 |
| | OLoRA | 89.4 | 94.7 | 86.3 | 98.2 | 94.3 | 97.9 | 92.8 | 96.0 | 93.6 |
| | EVA | 89.4 | 94.6 | 85.8 | 98.3 | 94.4 | 98.0 | 93.0 | 95.9 | 93.7 |
| | DoRA | 89.1 | 94.7 | 85.7 | 98.1 | 93.3 | 98.0 | 92.8 | 95.1 | 93.3 |
| | EVA+DoRA | 89.4 | 94.6 | 85.8 | 98.1 | 94.2 | 97.8 | 92.1 | 95.9 | 93.5 |
| Llama-3.1-70B | LoRA | 85.2 | 95.9 | 86.2 | 98.5 | 94.3 | 98.4 | 93.4 | 97.2 | 93.6 |
| | AdaLoRA | 90.4 | 95.1 | 85.8 | 98.0 | 93.3 | 98.2 | 93.7 | 96.7 | 93.8 |
| | PiSSA | 40.6 | 51.5 | 35.4 | 25.8 | 50.5 | 25.8 | 25.3 | 27.2 | 35.3 |
| | OLoRA | 90.3 | 96.0 | 86.2 | 98.4 | 95.5 | 98.3 | 93.5 | 96.9 | 94.4 |
| | EVA | 90.8 | 96.1 | 86.3 | 98.6 | 95.0 | 98.4 | 93.8 | 96.8 | 94.5 |

1. The standard setting $\alpha = 2r$ from Hu et al. (2022) leads to the worst performance

2. Rank stabilization via $\alpha = \frac{2r}{\sqrt{r}}$ significantly improves the performance of both LoRA and EVA

3. Different learning rates for $A$ and $B$ did not improve the results

To provide a comprehensive comparison of the effect of rank redistribution, we compare uniform ranks ($\rho = 1$) to adaptive ranks ($\rho = 2$) on common sense and math reasoning tasks in Table 11. We find that adaptive ranks consistently improve performance for Gemma-2-9B. For Llama-2-7B and Llama-3.1-8B we observe improvements in common sense reasoning tasks only, while uniform ranks perform better on math fine-tuning tasks. In Table 11 we also show the number of trainable

Table 8: Comparison of EVA to other initialization and adaptive rank methods on GSM8K and MATH datasets. We report mean and standard deviation across three random seeds.

| Model | Method | GSM8K | MATH |
|-------|--------|-------|------|
| Llama-2-7B | LoRA | $59.7_{\pm.8}$ | $10.9_{\pm.2}$ |
| | AdaLoRA | $56.9_{\pm.4}$ | $9.6_{\pm.2}$ |
| | PiSSA | $61.1_{\pm.3}$ | $12.6_{\pm.4}$ |
| | MiLoRA | $59.7_{\pm1.4}$ | $11.2_{\pm.1}$ |
| | OLoRA | $60.7_{\pm.5}$ | $11.8_{\pm.3}$ |
| | LoRA-GA | $60.2_{\pm.6}$ | $11.7_{\pm.4}$ |
| | CorDA | $59.0_{\pm1.2}$ | $11.8_{\pm.5}$ |
| | EVA | $\underline{61.9}_{\pm.5}$ | $\underline{13.1}_{\pm.3}$ |
| | DoRA | $59.8_{\pm.5}$ | $11.5_{\pm.2}$ |
| | EVA+DoRA | $\mathbf{62.5}_{\pm.8}$ | $\mathbf{13.4}_{\pm.01}$ |
| Llama-3.1-8B | LoRA | $78.3_{\pm.6}$ | $30.1_{\pm.5}$ |
| | AdaLoRA | $76.9_{\pm.2}$ | $28.9_{\pm.7}$ |
| | PiSSA | $78.8_{\pm.2}$ | $29.5_{\pm.5}$ |
| | MiLoRA | $78.6_{\pm.1}$ | $30.3_{\pm.3}$ |
| | OLoRA | $78.0_{\pm.1}$ | $\underline{31.0}_{\pm.7}$ |
| | LoRA-GA | $\underline{78.8}_{\pm.1}$ | $30.0_{\pm.1}$ |
| | CorDA | $76.8_{\pm.4}$ | $27.9_{\pm.2}$ |
| | EVA | $\underline{78.8}_{\pm.3}$ | $\mathbf{31.2}_{\pm.3}$ |
| | DoRA | $77.9_{\pm.1}$ | $30.2_{\pm.5}$ |
| | EVA+DoRA | $\mathbf{79.1}_{\pm.5}$ | $30.8_{\pm.4}$ |
| Gemma-2-9B | LoRA | $83.4_{\pm.9}$ | $40.7_{\pm.2}$ |
| | AdaLoRA | $\underline{83.5}_{\pm.5}$ | $41.1_{\pm.4}$ |
| | PiSSA | $79.8_{\pm.5}$ | $34.9_{\pm.2}$ |
| | MiLoRA | $83.7_{\pm.4}$ | $\mathbf{41.9}_{\pm.3}$ |
| | OLoRA | $82.2_{\pm.2}$ | $39.4_{\pm.6}$ |
| | LoRA-GA | $82.8_{\pm.8}$ | $40.4_{\pm.4}$ |
| | CorDA | $56.3_{\pm6.2}$ | $25.4_{\pm4.0}$ |
| | EVA | $\mathbf{83.6}_{\pm.8}$ | $\underline{41.5}_{\pm.3}$ |
| | DoRA | $82.5_{\pm.6}$ | $39.7_{\pm.4}$ |
| | EVA+DoRA | $82.9_{\pm.3}$ | $40.0_{\pm.6}$ |

parameters for EVA ($\rho = 2$) compared to LoRA on common sense and math reasoning tasks. We find that after rank redistribution, EVA leads to improved performance while reducing the parameter count by approximately 1M. The reason for this is that parameters are usually redistributed from higher dimensional projections to lower dimensional ones, i.e. from non-attention weights to attention weights. This results in improved performance while reducing the parameter count.

Finally, to verify our intuition that the LoRA matrix $\boldsymbol{A}$ should be initialized with the projection onto the components that explain the most variance, we compare its performance with initializing EVA with the components that explain the *least* amount of variance. We call this method EVA-minor and present results for it in Table 12. To implement EVA-minor, we sample 20 minibatches of data and perform truncated SVD on those and select the resulting minor components. This incurs substantial additional cost, as we must compute all components, whereas for EVA we only approximate the components that explain the most variance. Hence, incremental SVD is not beneficial in this case anymore and it is also not practical as obtaining the initialization takes hours instead of seconds for EVA. Moreover, our data-driven heuristic for adaptive rank allocation is no longer applicable to this case; therefore, we consider uniform ranks. Finally, we find that EVA consistently improves over EVA-minor, highlighting the importance of initializing EVA with the major components, i.e. the ones that explain the most variance.

In addition we also fine-tune Llama-2-7B on the Code-Feedback dataset Zheng et al. (2024) consisting of multi-turn conversations between user and AI Assistant. Due to limited computational resources and the long sequence lengths of the examples in this dataset we do not fine-tune Llama-3.1-8B and Gemma-2-9B or any DoRA variants. We evaluate the fine-tuned checkpoints on four coding benchmarks: MBPP Austin et al. (2021), HumanEval Chen et al. (2021b), MBPP+ and HumanEval+ Liu et al. (2023). The results are presented in Table 13. EVA shows the best performance on MBPP

Table 9: Comparison of different ranks for fine-tuning Llama-2-7B on the eight common sense reasoning tasks.

| Rank | Method | BoolQ | PIQA | SIQA | HellaSwag | Winogrande | ARC-e | ARC-c | OBQA | Avg. |
|---|---|---|---|---|---|---|---|---|---|---|
| 8 | LoRA | 67.6 | 84.0 | 82.1 | 94.6 | 84.2 | 88.1 | 74.2 | 83.5 | 82.3 |
| | AdaLoRA | 70.0 | 82.4 | 80.7 | 93.4 | 80.1 | 86.4 | 70.9 | 79.9 | 80.5 |
| | PiSSA | 62.5 | 84.9 | 81.2 | 93.9 | 84.2 | 87.0 | 74.4 | 85.4 | 81.7 |
| | OLoRA | 65.4 | 84.5 | 82.3 | 94.9 | 84.8 | 88.4 | 74.7 | 85.5 | 82.6 |
| | LoRA-GA | 69.1 | 84.8 | 82.2 | 94.8 | 84.1 | 87.8 | 73.9 | 85.7 | 82.8 |
| | EVA ($\rho=1$) | 72.6 | 85.4 | 82.3 | 95.2 | 84.9 | 88.8 | 75.2 | 85.3 | 83.7 |
| | EVA ($\rho=2$) | 74.1 | 85.6 | 82.6 | 95.1 | 85.0 | 88.7 | 75.5 | 86.3 | 84.1 |
| | DoRA | 65.0 | 84.6 | 82.3 | 94.9 | 84.3 | 88.7 | 74.7 | 85.6 | 82.5 |
| | EVA+DoRA ($\rho=1$) | 71.6 | 85.8 | 82.5 | 95.2 | 85.3 | 88.9 | 75.3 | 86.2 | 83.9 |
| | EVA+DoRA ($\rho=2$) | 69.9 | 84.7 | 82.3 | 95.2 | 84.0 | 88.3 | 74.8 | 84.3 | 82.9 |
| 16 | LoRA | 68.0 | 84.0 | 82.1 | 94.7 | 83.8 | 87.8 | 73.8 | 84.5 | 82.3 |
| | AdaLoRA | 73.8 | 82.1 | 80.6 | 93.3 | 79.2 | 86.1 | 71.1 | 80.1 | 80.8 |
| | PiSSA | 62.6 | 84.9 | 81.3 | 94.5 | 84.6 | 87.6 | 75.2 | 85.5 | 82.0 |
| | OLoRA | 69.5 | 84.8 | 82.5 | 95.0 | 84.6 | 88.0 | 74.7 | 85.1 | 83.0 |
| | MiLoRA | 65.0 | 84.8 | 82.3 | 94.9 | 84.5 | 88.2 | 74.9 | 85.3 | 82.5 |
| | LoRA-GA | 69.0 | 85.6 | 82.3 | 95.0 | 85.0 | 88.7 | 75.9 | 85.8 | 83.4 |
| | EVA ($\rho=1$) | 71.2 | 85.2 | 82.2 | 95.2 | 84.2 | 88.6 | 75.4 | 84.9 | 83.4 |
| | EVA ($\rho=2$) | 68.3 | 85.3 | 82.9 | 95.2 | 85.2 | 88.6 | 75.8 | 86.3 | 83.4 |
| | DoRA | 68.3 | 85.1 | 82.2 | 94.9 | 84.3 | 88.7 | 74.8 | 86.3 | 83.1 |
| | EVA+DoRA ($\rho=1$) | 73.5 | 85.3 | 82.4 | 95.2 | 84.8 | 88.9 | 76.0 | 87.3 | 84.2 |
| | EVA+DoRA ($\rho=2$) | 74.4 | 85.3 | 82.5 | 95.1 | 85.2 | 88.9 | 75.4 | 85.4 | 84.0 |
| 32 | LoRA | 69.1 | 84.0 | 82.0 | 94.7 | 83.7 | 88.2 | 73.9 | 84.4 | 82.5 |
| | AdaLoRA | 72.6 | 82.2 | 80.6 | 93.2 | 80.3 | 86.2 | 71.1 | 79.9 | 80.8 |
| | PiSSA | 65.1 | 84.7 | 81.0 | 94.1 | 84.5 | 87.6 | 73.5 | 86.2 | 82.1 |
| | OLoRA | 63.6 | 84.8 | 82.4 | 95.0 | 84.7 | 88.6 | 75.2 | 85.7 | 82.5 |
| | LoRA-GA | 69.0 | 85.7 | 82.0 | 95.3 | 84.7 | 88.8 | 75.2 | 86.5 | 83.4 |
| | EVA ($\rho=1$) | 69.2 | 85.1 | 82.9 | 95.0 | 85.3 | 88.6 | 74.9 | 85.3 | 83.3 |
| | EVA ($\rho=2$) | 65.4 | 85.4 | 82.9 | 95.2 | 85.0 | 88.5 | 75.3 | 85.4 | 82.9 |
| | DoRA | 66.9 | 84.9 | 82.1 | 95.0 | 84.5 | 88.6 | 74.7 | 84.7 | 82.7 |
| | EVA+DoRA ($\rho=1$) | 69.0 | 85.8 | 82.7 | 95.2 | 84.8 | 89.1 | 75.7 | 86.9 | 83.7 |
| | EVA+DoRA ($\rho=2$) | 71.0 | 84.2 | 81.9 | 95.0 | 84.3 | 87.8 | 74.3 | 85.0 | 82.9 |
| 64 | LoRA | 74.7 | 84.2 | 82.1 | 94.6 | 84.0 | 88.0 | 75.0 | 83.8 | 83.3 |
| | AdaLoRA | 71.5 | 82.0 | 80.4 | 93.1 | 80.2 | 86.0 | 71.1 | 79.9 | 80.5 |
| | PiSSA | 64.9 | 84.6 | 81.3 | 94.0 | 84.5 | 87.6 | 73.3 | 85.0 | 81.9 |
| | OLoRA | 70.0 | 84.8 | 82.4 | 94.9 | 84.7 | 88.7 | 75.3 | 85.9 | 83.3 |
| | LoRA-GA | 70.5 | 85.2 | 82.4 | 95.1 | 84.6 | 88.7 | 75.4 | 85.5 | 83.4 |
| | EVA ($\rho=1$) | 66.6 | 85.2 | 82.6 | 95.0 | 84.8 | 88.3 | 75.3 | 85.1 | 82.9 |
| | EVA ($\rho=2$) | 71.2 | 84.7 | 82.7 | 95.0 | 84.5 | 88.6 | 74.9 | 85.3 | 83.3 |
| | DoRA | 70.5 | 85.0 | 82.6 | 94.9 | 84.8 | 88.3 | 74.7 | 85.9 | 83.3 |
| | EVA+DoRA ($\rho=1$) | 67.4 | 85.3 | 82.6 | 95.1 | 84.9 | 88.9 | 75.5 | 86.6 | 83.3 |
| | EVA+DoRA ($\rho=2$) | 71.6 | 84.6 | 82.2 | 94.9 | 84.0 | 88.2 | 75.0 | 84.8 | 83.2 |

Table 10: Comparison of EVA to LoRA using recently proposed advancements, such as rank stabilized scaling (Kalajdzievski, 2023) or different learning rates for $B$ and $A$ (Hayou et al., 2024), as well as the originally proposed scaling from Hu et al. (2022).

| Adaptation | Method | BoolQ | PIQA | SIQA | HellaSwag | Winogrande | ARC-e | ARC-c | OBQA | Avg. |
|---|---|---|---|---|---|---|---|---|---|---|
| LoRA+ | LoRA | 64.5 | 84.7 | 81.6 | 94.4 | 83.8 | 87.3 | 73.9 | 85.5 | 82.0 |
| | EVA | 68.6 | 85.0 | 81.2 | 94.2 | 84.7 | 87.4 | 73.5 | 84.1 | 82.3 |
| rsLoRA | LoRA | 71.5 | 85.3 | 82.5 | 95.2 | 84.5 | 89.0 | 75.8 | 86.8 | 83.8 |
| | EVA | 75.5 | 86.1 | 82.7 | 95.4 | 86.1 | 89.3 | 76.3 | 86.3 | 84.7 |
| $\alpha=32$ | LoRA | 77.9 | 82.1 | 80.1 | 93.2 | 79.8 | 86.3 | 71.5 | 79.3 | 81.3 |
| | EVA | 68.6 | 84.9 | 82.2 | 94.6 | 84.1 | 87.8 | 74.7 | 84.4 | 82.7 |

and MBPP+ while also exhibiting good performance on HumanEval and HumanEval+. For the latter two datasets, PiSSA is the best-performing method. For fine-tuning, we use a maximum sequence

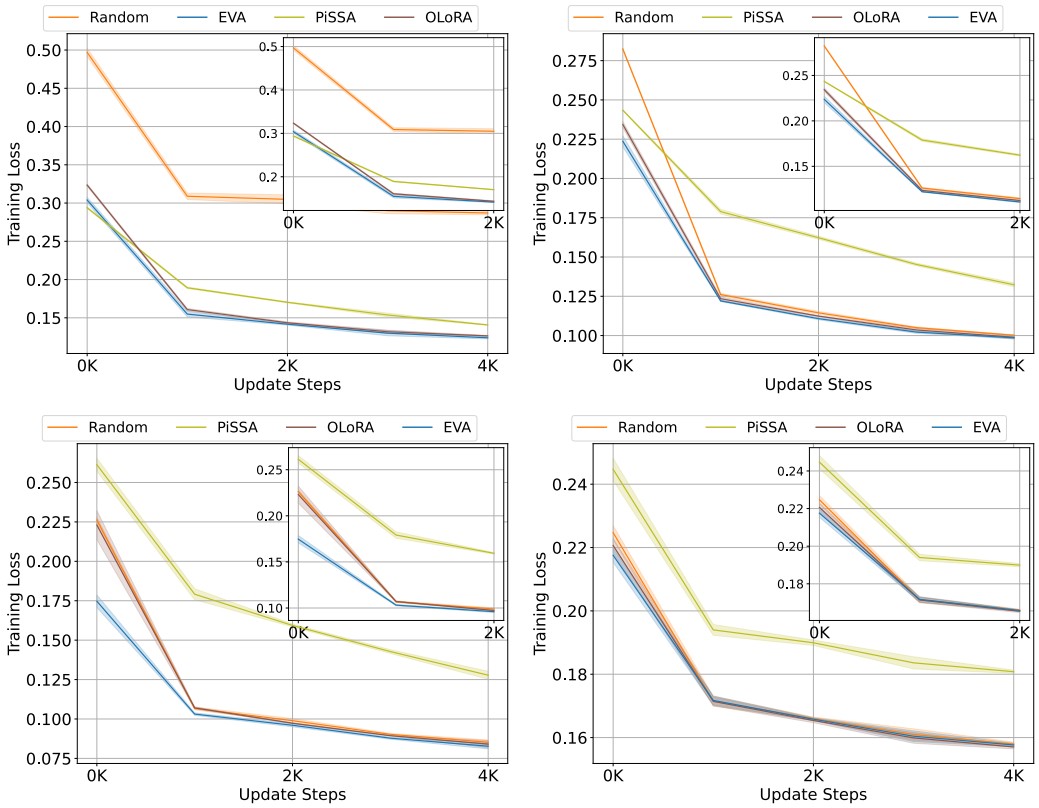

Figure 6: Loss curves for Llama-2-7B on common sense reasoning (top left), Llama-3.1-8B on common sense reasoning (top right), Gemma-2-9B on common sense reasoning (bottom right), and Gemma-2-9B on MetaMathQA. EVA consistently converges the fastest among all competitors.

length of 2028 with a right-hand side truncation. For decoding, we set the temperature to 0.2 and `top_p` to 0.7

## C    Natural language understanding

### C.1    Dataset Statistics

The dataset statistics for each task in the GLUE benchmark (Wang et al., 2019) are shown in Table 15. Generally, GLUE contains four low-resource datasets (RTE, MRPC, STS-B, and CoLA) and four high-resource datasets (SST-2, QNLI, QQP, and MNLI). While CoLA and SST-2 rely on single sentence classification, STS-B evaluates for similarity and the remaining tasks are based on pairwise text classification.

### C.2    Implementation Details

We base our implementation on the LoRA codebase[1]. For these experiments, we initially precompute our initialization prior to the fine-tuning stage and store it as a checkpoint. However, we also provide the possibility to directly compute the initialization during the fine-tuning stage, as done for our experiments on VTAB-1k and Meta-World. By default, we always offload the computation of the initial checkpoint to CPU to save VRAM. We ran all our experiments on nodes with four A100 GPUs and used PyTorch's data-distributed parallel functionality (Paszke et al., 2019). Runtimes range from as little as 10 minutes per run for smaller datasets (RTE, STS-B) to around 15 hours for the largest datasets (QQP, MNLI).

---

[1] https://github.com/microsoft/LoRA

Table 11: Comparison of number of trainable parameters between LoRA-based methods and EVA on the math and common sense reasoning tasks. Common sense reasoning is an average over eight tasks. #Trainable represents the number of trainable parameters. EVA consistently improves performance while decreasing the number of trainable parameters.

| Model | Method | #Trainable | Common sense | GSM8K | MATH |
|---|---|---|---|---|---|
| Llama-2-7B | LoRA | 40.6M | 82.2 | 59.7 | 10.9 |
| | AdaLoRA | 40.6M | 81.0 | 56.9 | 9.6 |
| | PiSSA | 40.6M | 82.0 | 61.1 | 12.6 |
| | MiLoRA | 40.6M | 82.5 | 59.7 | 11.2 |
| | OLoRA | 40.6M | 82.9 | 60.7 | 11.8 |
| | LoRA-GA | 40.6M | 83.4 | 60.2 | 11.7 |
| | EVA ($\rho = 1$) | 40.6M | 83.4 | 61.9 | 13.1 |
| | EVA ($\rho = 2$) | 39.3M | 83.4 | 61.0 | 12.5 |
| Llama-3.1-8B | LoRA | 44.1M | 89.2 | 78.3 | 30.1 |
| | AdaLoRA | 44.1M | 87.6 | 76.9 | 28.9 |
| | PiSSA | 44.1M | 85.7 | 78.8 | 29.5 |
| | MiLoRA | 44.1M | 89.4 | 78.6 | 30.3 |
| | OLoRA | 44.1M | 89.4 | 78.0 | 31.0 |
| | LoRA-GA | 44.1M | 89.0 | 78.8 | 30.0 |
| | EVA ($\rho = 1$) | 44.1M | 89.4 | 78.8 | 31.2 |
| | EVA ($\rho = 2$) | 42M | 89.5 | 78.3 | 30.8 |
| Gemma-2-9B | LoRA | 58.2M | 92.2 | 83.4 | 40.7 |
| | AdaLoRA | 58.2M | 91.5 | 83.5 | 41.1 |
| | PiSSA | 58.2M | 88.3 | 79.8 | 34.9 |
| | MiLoRA | 58.2M | 92.3 | 83.7 | 41.9 |
| | OLoRA | 58.2M | 91.8 | 82.2 | 39.4 |
| | LoRA-GA | 58.2M | 91.8 | 82.8 | 40.4 |
| | EVA ($\rho = 1$) | 58.2M | 92.4 | 83.6 | 41.3 |
| | EVA ($\rho = 2$) | 55.9M | 92.5 | 83.6 | 41.5 |
| Gemma-2-27B | LoRA | 114.2M | 93.1 | - | - |
| | AdaLoRA | 114.2M | 93.0 | - | - |
| | PiSSA | 114.2M | 88.8 | - | - |
| | OLoRA | 114.2M | 93.7 | - | - |
| | EVA ($\rho = 1$) | 114.2M | 93.7 | - | - |
| | EVA ($\rho = 2$) | 104.8M | 93.7 | - | - |
| Llama-3.1-70B | LoRA | 209.3M | 93.6 | - | - |
| | AdaLoRA | 209.3M | 93.9 | | |
| | PiSSA | 209.3M | 35.2 | - | - |
| | OLoRA | 209.3M | 94.4 | - | - |
| | EVA ($\rho = 1$) | 209.3M | 94.5 | - | - |
| | EVA ($\rho = 2$) | 193.6M | 94.5 | - | - |

## C.3 Hyperparameter search

For LoRA and EVA, we search the number of ranks $r \in \{2, 4, 6, 8\}$ and the different learning rates $\eta \in \{1e-3, 4e-4, 1e-4\}$ for RoBERTa$_{\text{Large}}$ and $\eta \in \{4e-3, 1e-3, 4e-4\}$ for DeBERTav3$_{\text{Base}}$. We report the best hyperparameter settings for both RoBERTa$_{\text{Large}}$ and DeBERTav3$_{\text{Base}}$ for LoRA and EVA in Table 16. For AdaLoRA, we search the same ranks and always start the initial ranks with $r + 4$ that are then redistributed during training. For BOFT we sweep over different combinations of block sizes $b \in \{2, 4, 8, 16\}$ which determine the number of multiplicative matrices. Additionally, for both AdaLoRA and BOFT, we search over the same learning rates as for the other LoRA variants. Further, we introduce hyperparameters that result in additional speed-up of our initialization, namely a threshold $\tau$ that considers components as converged, and a threshold $\delta$ that stops computation of the

Table 12: Comparison of EVA to EVA-minor, which leverages components that explain the *least* amount of variance for initialization of $A$, on the common sense reasoning tasks.

| Method | BoolQ | PIQA | SIQA | HellaSwag | Winogrande | ARC-e | ARC-c | OBQA | Avg. |
|--------|-------|------|------|-----------|------------|-------|-------|------|------|
| EVA | 68.6 | 85.0 | 81.2 | 94.2 | 84.7 | 87.4 | 73.5 | 84.1 | 82.3 |
| EVA-minor | 64.0 | 83.4 | 81.5 | 94.3 | 82.0 | 87.3 | 73.0 | 81.6 | 80.9 |

Table 13: Comparison of EVA to other initialization and rank re-distribution schemes on code fine-tuning datasets. We report mean and standard deviation across three random seeds.

| Method | MBPP | HumanEval | MBPP+ | HumanEval+ |
|--------|------|-----------|-------|------------|
| LoRA | $22.2_{\pm1.1}$ | $\underline{18.9}_{\pm0.6}$ | $30.7_{\pm1.1}$ | $\underline{18.9}_{\pm0.6}$ |
| AdaLoRA | $21.5_{\pm0.2}$ | $17.1_{\pm0.0}$ | $29.4_{\pm0.7}$ | $17.1_{\pm0.0}$ |
| PiSSA | $\underline{22.8}_{\pm1.2}$ | $\mathbf{19.9}_{\pm0.9}$ | $30.8_{\pm0.7}$ | $\mathbf{19.9}_{\pm0.9}$ |
| OLoRA | $22.3_{\pm0.6}$ | $\underline{18.9}_{\pm0.0}$ | $\underline{32.4}_{\pm0.4}$ | $\underline{18.9}_{\pm0.0}$ |
| EVA | $\mathbf{22.9}_{\pm0.7}$ | $\underline{18.9}_{\pm1.2}$ | $\mathbf{32.6}_{\pm0.6}$ | $\underline{18.9}_{\pm1.2}$ |

initialization when a certain percentage of components have converged. By default, we set $\tau = 0.99$ and $\delta = 1$, i.e. we only stop when all components converge. These parameters provide additional leeway to speed up the initialization stage of EVA.

We have explored the sensitivity of LoRA to different initialization schemes and found that, similar to other prominent initialization schemes (He et al., 2015; Glorot & Bengio, 2010), scale plays an important role along with directions. Originally, (Hu et al., 2022) propose to set $\alpha = 2r$, however, we found that this parameter is quite sensitive as also shown in (Kalajdzievski, 2023). Similarly, different ranks lead to very different results on different downstream tasks. Therefore, we suggest that one always search over more ranks and choose the best performing one if the required budget is available. We also experimented with different learning rates for the $A$ and $B$ matrices as proposed in (Hayou et al., 2024), however, this did not result in consistent improvements. Instead, we found that learning rates for LoRA-style training can be surprisingly high ($4e - 3$ for DeBERTav3$_{\text{Base}}$), while for larger models the learning rate needs to be approximately a magnitude smaller. A simple recipe that worked consistently well was to set $\alpha = 1$, which results in a similar scaling factor as in Kalajdzievski (2023), and searching over a set of small learning rates for larger models and higher learning rates for smaller ones. For EVA, the only tunable hyperparameter is the rank budget, which we recommend to tune along with the learning rate.

## C.4 Additional results

We report additional results for EVA compared to LoRA for different rank budgets in Table 17. We find that EVA consistently outperforms LoRA for different rank budgets. This demonstrates the effectiveness of EVA among different compute budgets. In addition, we show additional rank redistributions for CoLA, MRPC, RTE, and STSB tasks for different for $r = 2$ (Figure 7), $r = 4$ (Figure 8), $r = 8$ (Figure 9), and $r = 16$ (Figure 10) for both RoBERTa$_{\text{Large}}$ and DeBERTav3$_{\text{Base}}$. The distributions for the different models show different patterns. For DeBERTav3$_{\text{Base}}$, the higher attention layers usually receive more ranks than the lower ones. For CoLA, there are also a large number of ranks in the very first layer. For RoBERTa$_{\text{Large}}$, it seems to be the opposite, as the very first layers consistently receive more ranks compared to the later layers. There is also a notable difference between tasks for both models, which demonstrates the flexibility of EVA to allocate ranks dependent on the downstream task. Interestingly, for a higher initial rank ($r = 16$), the redistribution for DeBERTav3$_{\text{Base}}$ puts more emphasis on fine-tuning the self-attention specific weight matrices. This is not true for RoBERTa$_{\text{Large}}$, as $W_{f1}$ also receives plenty of ranks across all tasks. Overall, the rank redistribution incurs different fine-tuning paradigms depending on the task and the initial rank.

Additionally, we show results for different rank redistributions that we obtain by using alternative measures for explained variance. Specifically, we compare EVA to using (i) the raw eigenvalues

Table 14: Per-task standard deviation across three seeds for all methods on common sense reasoning tasks.

| Model | Method | BoolQ | PIQA | SIQA | HellaSwag | Winogrande | ARC-e | ARC-c | OBQA |
|-------|--------|-------|------|------|-----------|------------|-------|-------|------|
| Llama-2-7B | LoRA | 1.498 | 0.252 | 0.233 | 0.102 | 0.658 | 0.072 | 0.489 | 0.822 |
| | AdaLoRA | 1.315 | 0.251 | 0.182 | 0.098 | 0.392 | 0.362 | 0.106 | 0.899 |
| | PiSSA | 0.358 | 0.294 | 0.138 | 0.096 | 0.298 | 0.386 | 0.494 | 1.117 |
| | MiLoRA | 3.950 | 0.392 | 0.329 | 0.097 | 0.810 | 0.064 | 1.100 | 0.231 |
| | OLoRA | 4.938 | 0.190 | 0.524 | 0.062 | 0.652 | 0.339 | 0.672 | 0.660 |
| | LoRA-GA | 10.573 | 0.416 | 1.049 | 0.115 | 0.344 | 0.170 | 0.560 | 0.721 |
| | CorDA | 8.801 | 2.039 | 0.253 | 0.549 | 2.009 | 1.756 | 2.836 | 4.243 |
| | EVA | 7.974 | 0.137 | 1.054 | 0.101 | 0.810 | 0.526 | 0.421 | 0.577 |
| | DoRA | 2.599 | 0.290 | 0.483 | 0.113 | 0.244 | 0.215 | 0.489 | 0.525 |
| | EVA+DoRA | 5.281 | 0.273 | 0.293 | 0.034 | 0.853 | 0.110 | 0.494 | 0.249 |
| Llama-3.1-8B | LoRA | 0.472 | 0.194 | 0.419 | 0.070 | 0.197 | 0.052 | 0.563 | 0.189 |
| | AdaLoRA | 0.510 | 0.044 | 0.261 | 0.040 | 0.392 | 0.201 | 0.804 | 0.748 |
| | PiSSA | 6.516 | 0.373 | 0.603 | 0.195 | 0.707 | 0.325 | 0.245 | 0.589 |
| | MiLoRA | 0.511 | 0.163 | 0.300 | 0.125 | 0.613 | 0.445 | 0.887 | 0.503 |
| | OLoRA | 0.298 | 0.245 | 0.397 | 0.057 | 0.451 | 0.173 | 0.329 | 0.189 |
| | LoRA-GA | 0.539 | 0.237 | 0.695 | 0.115 | 0.592 | 0.135 | 0.729 | 0.800 |
| | CorDA | 3.676 | 0.077 | 0.145 | 0.070 | 2.009 | 1.905 | 1.508 | 0.424 |
| | EVA | 0.353 | 0.031 | 0.194 | 0.046 | 0.209 | 0.292 | 0.178 | 0.808 |
| | DoRA | 0.225 | 0.112 | 0.315 | 0.014 | 0.260 | 0.119 | 0.698 | 0.000 |
| | EVA+DoRA | 0.225 | 0.168 | 0.121 | 0.117 | 0.392 | 0.105 | 0.175 | 0.249 |
| Gemma-2-9B | LoRA | 0.095 | 0.277 | 0.386 | 0.062 | 0.324 | 0.072 | 0.070 | 0.589 |
| | AdaLoRA | 0.088 | 0.353 | 0.217 | 0.033 | 0.098 | 0.209 | 0.106 | 0.432 |
| | PiSSA | 2.761 | 0.286 | 0.214 | 0.109 | 0.621 | 0.447 | 0.121 | 0.163 |
| | MiLoRA | 0.284 | 0.191 | 0.325 | 0.089 | 1.008 | 0.239 | 0.903 | 0.115 |
| | OLoRA | 0.066 | 0.451 | 0.501 | 0.099 | 0.501 | 0.267 | 0.448 | 0.573 |
| | LoRA-GA | 0.662 | 0.463 | 0.252 | 0.072 | 0.526 | 0.129 | 0.617 | 1.026 |
| | CorDA | 17.299 | 0.154 | 0.109 | 1.486 | 1.730 | 0.268 | 0.845 | 0.000 |
| | EVA | 0.275 | 0.136 | 0.111 | 0.094 | 0.260 | 0.119 | 0.040 | 0.249 |
| | DoRA | 0.189 | 0.420 | 0.301 | 0.074 | 0.419 | 0.091 | 0.000 | 0.499 |
| | EVA+DoRA | 0.132 | 0.296 | 0.490 | 0.070 | 0.037 | 0.150 | 0.715 | 0.340 |
| Gemma-2-27B | LoRA | 0.202 | 0.045 | 0.424 | 0.109 | 0.196 | 0.155 | 0.600 | 0.497 |
| | AdaLoRA | 0.300 | 0.286 | 0.158 | 0.022 | 0.429 | 0.020 | 0.161 | 0.249 |
| | PiSSA | 3.035 | 0.645 | 0.529 | 0.135 | 0.578 | 0.288 | 0.408 | 0.736 |
| | OLoRA | 0.038 | 0.200 | 0.233 | 0.046 | 0.226 | 0.182 | 0.435 | 0.864 |
| | EVA | 0.250 | 0.277 | 0.147 | 0.031 | 0.322 | 0.292 | 0.707 | 0.432 |
| | DoRA | 0.364 | 0.194 | 0.111 | 0.038 | 0.149 | 0.110 | 0.329 | 0.189 |
| | EVA+DoRA | 0.336 | 0.000 | 0.026 | 0.085 | 0.316 | 0.084 | 0.555 | 0.500 |
| Llama-3.1-70B | LoRA | 7.296 | 0.068 | 0.230 | 0.059 | 0.134 | 0.105 | 0.418 | 0.327 |
| | AdaLoRA | 0.300 | 0.077 | 0.274 | 0.060 | 0.232 | 0.110 | 0.224 | 0.189 |
| | PiSSA | 1.208 | 0.544 | 1.407 | 0.070 | 0.079 | 0.968 | 1.195 | 3.400 |
| | OLoRA | 0.548 | 0.143 | 0.301 | 0.119 | 0.207 | 0.209 | 0.426 | 0.411 |
| | EVA | 0.227 | 0.204 | 0.319 | 0.059 | 0.335 | 0.069 | 0.420 | 0.249 |

(EVA-Raw) and (ii) normalizing by the maximum eigenvalue (EVA-Max). We report results for RoBERTa$_{\text{Large}}$ on four GLUE tasks, namely CoLA, RTE, MRPC, and STS-B in Table 18. Our results show that while EVA-Raw and EVA-Max slightly improve upon LoRA, they perform worse on average than EVA.

## D   Image Classification

### D.1   Dataset statistics

The VTAB-1K benchmark consists of 19 datasets, each containing a subset of 1000 examples of their respective samples. We summarize the statistics for each dataset in Table 19. Although the original

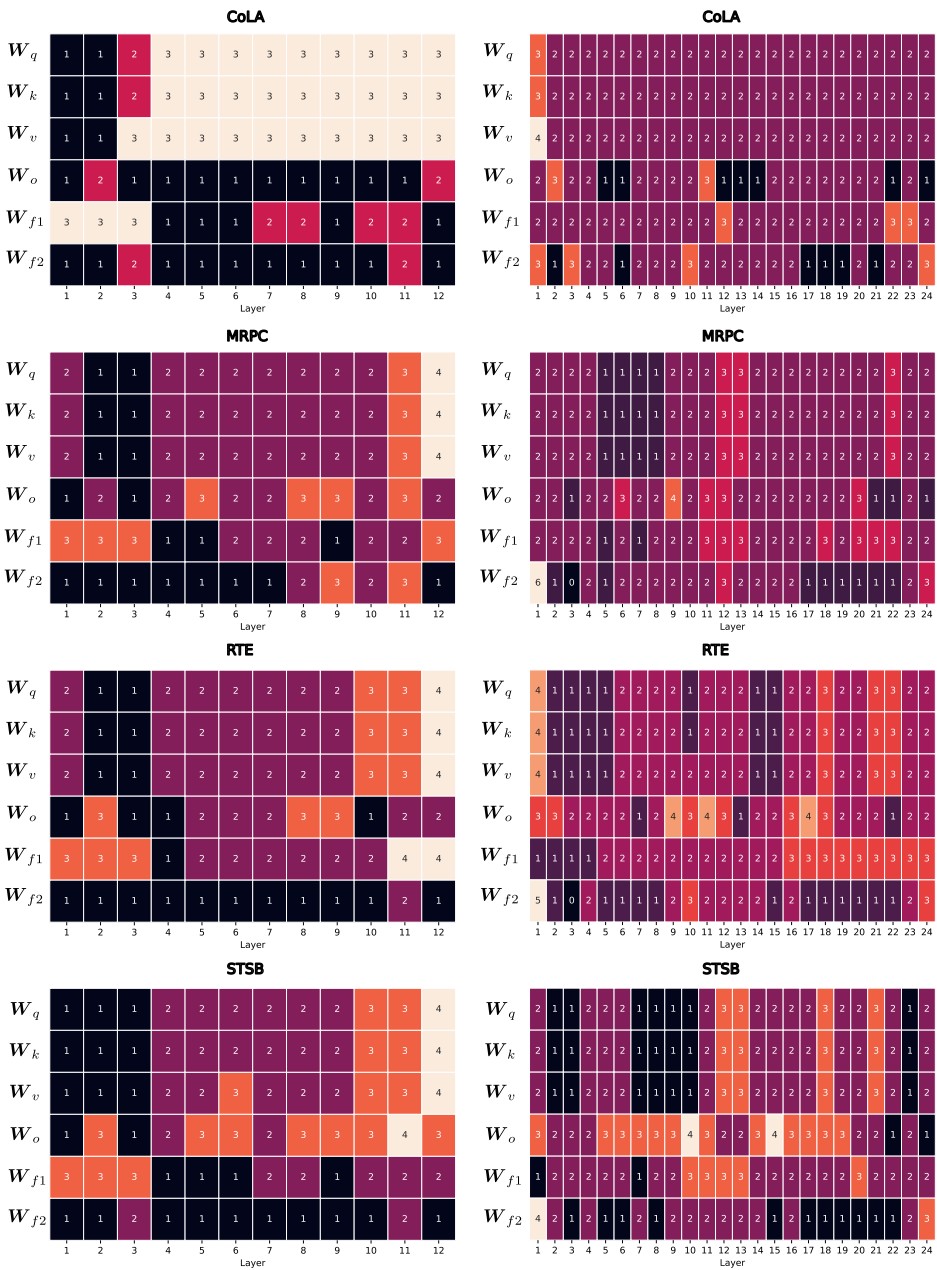

Figure 7: Rank distribution after initialization with EVA on four tasks of the GLUE benchmark (CoLA, MRPC, RTE, STSB) for DeBERTav3$_{\text{Base}}$ (left) and RoBERTa$_{\text{Large}}$ (right) with initial rank $r = 2$.

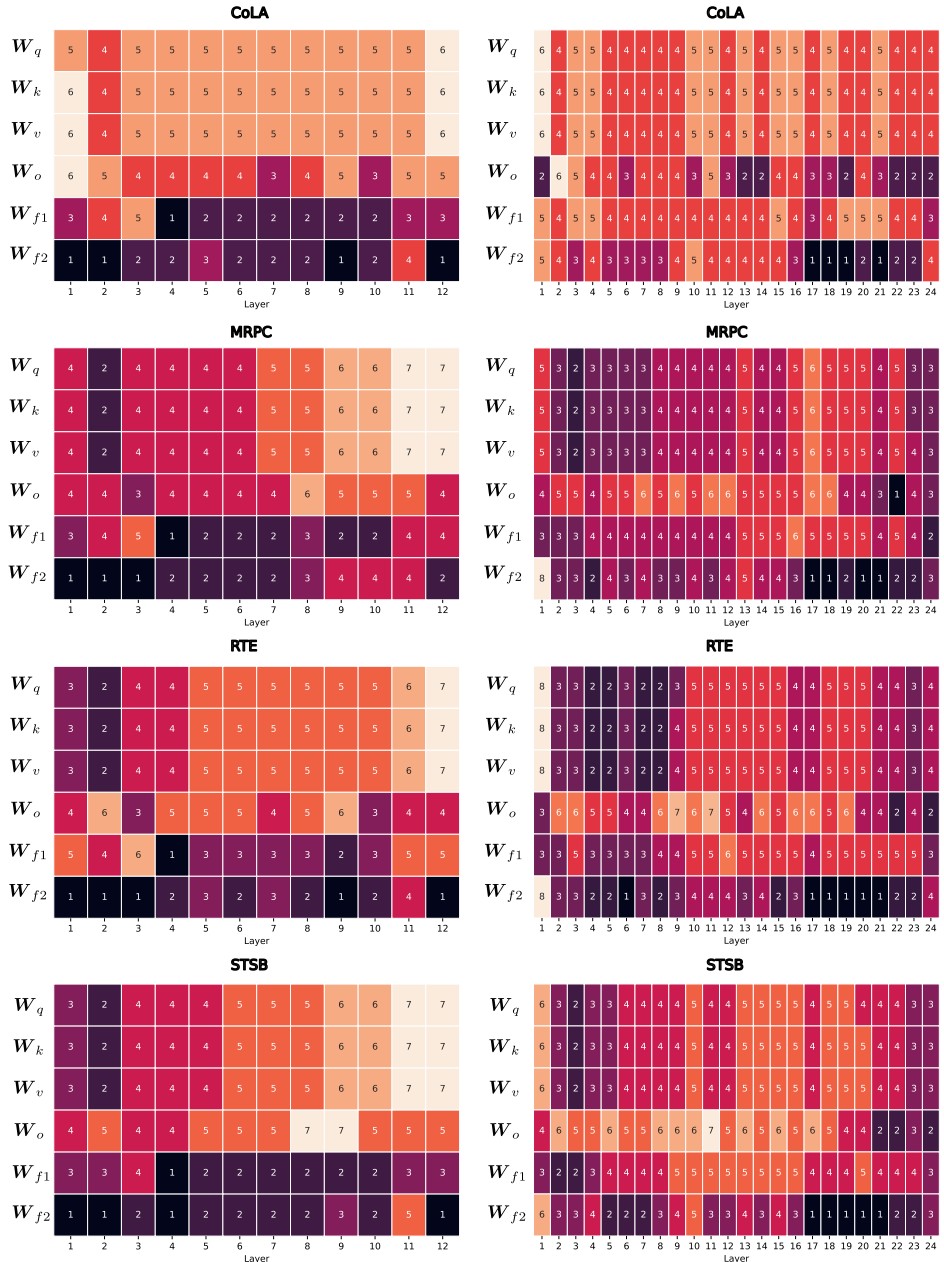

Figure 8: Rank distribution after initialization with EVA on four tasks of the GLUE benchmark (CoLA, MRPC, RTE, STSB) for DeBERTav3$_{\text{Base}}$ (left) and RoBERTa$_{\text{Large}}$ (right) with initial rank $r = 4$.

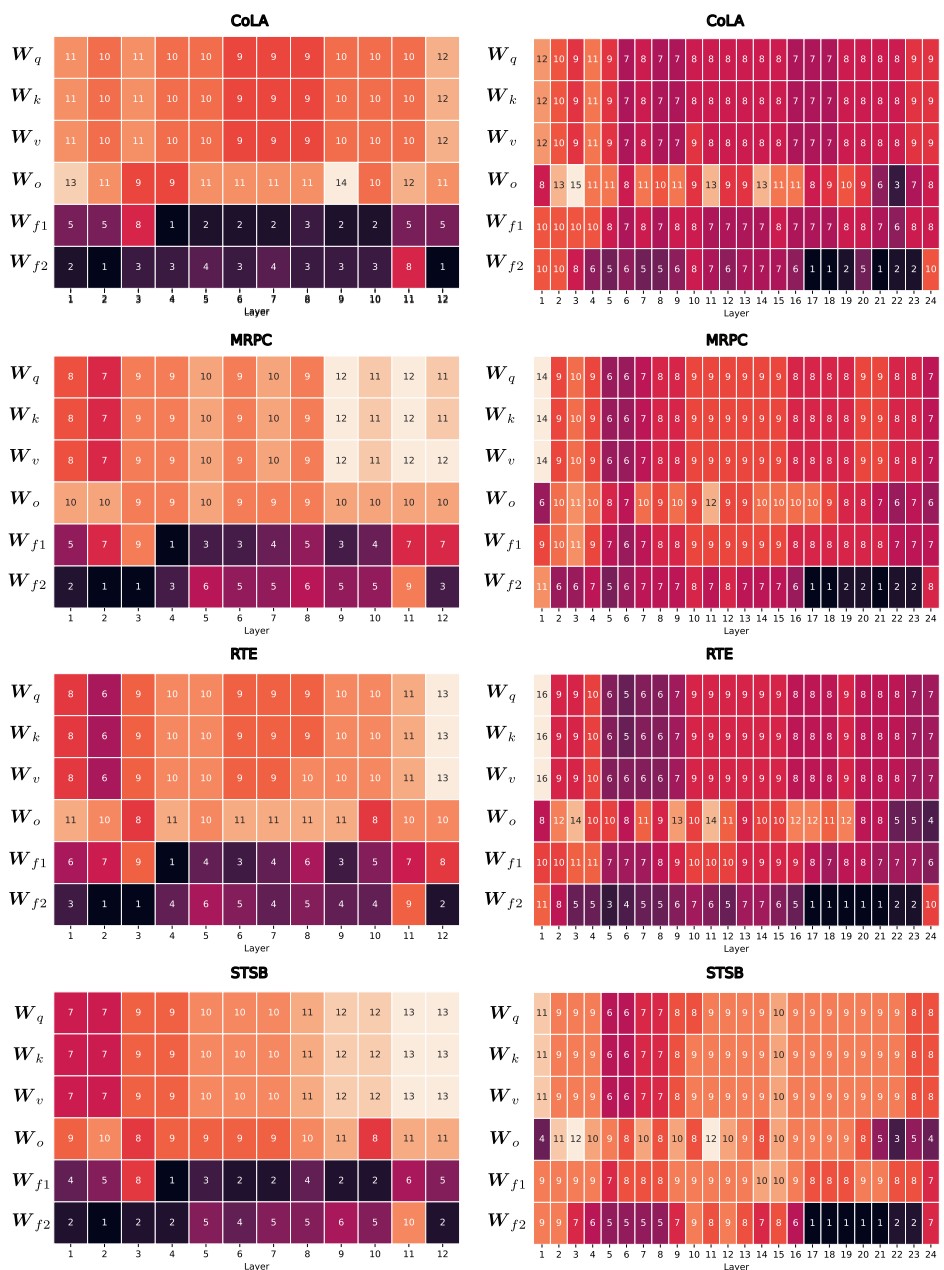

Figure 9: Rank distribution after initialization with EVA on four tasks of the GLUE benchmark (CoLA, MRPC, RTE, STSB) for DeBERTav3$_{\text{Base}}$ (left) and RoBERTa$_{\text{Large}}$ (right) with initial rank $r = 8$.

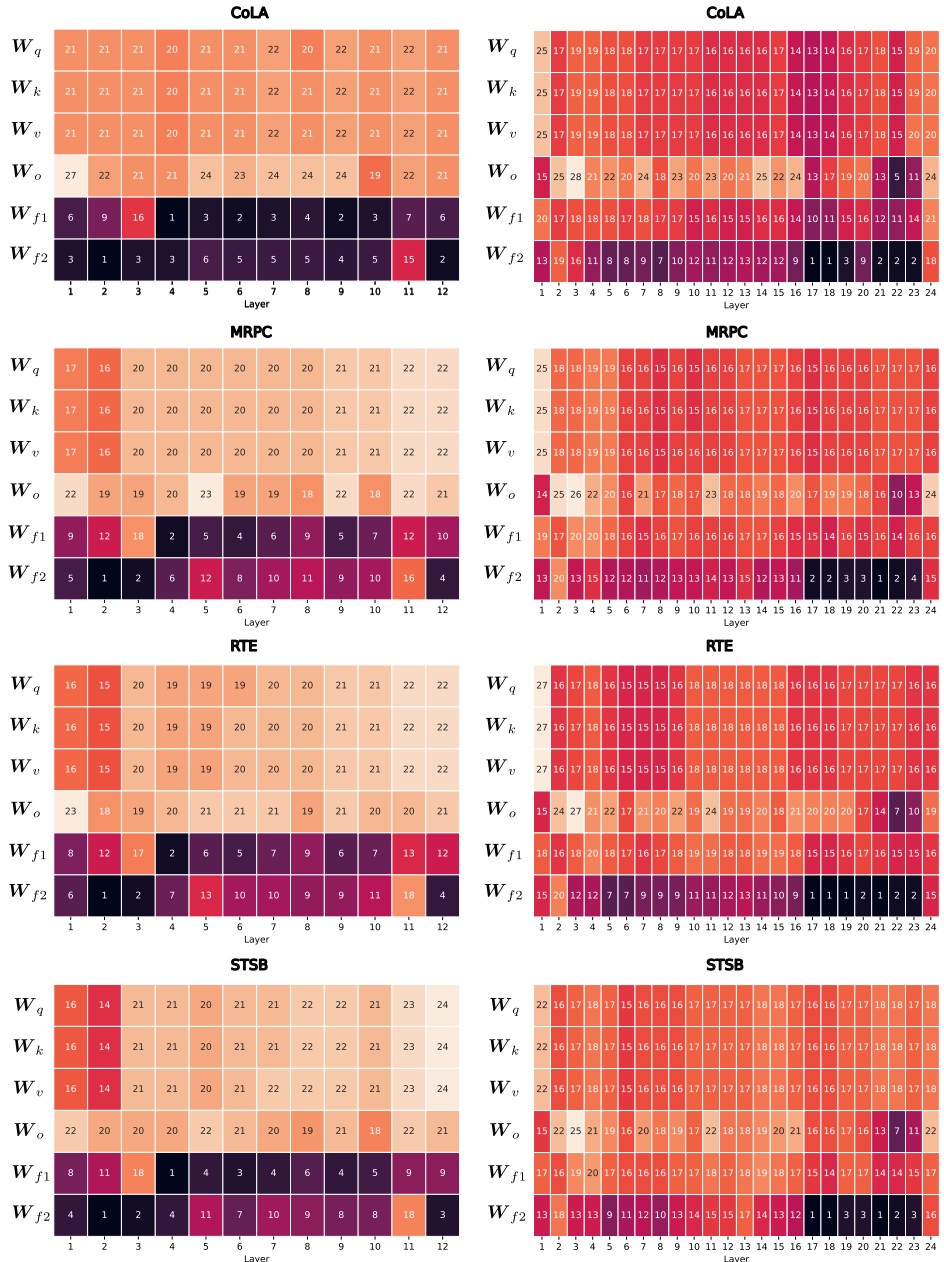

Figure 10: Rank distribution after initialization with EVA on four tasks of the GLUE benchmark (CoLA, MRPC, RTE, STSB) for DeBERTav3$_{\text{Base}}$ (left) and RoBERTa$_{\text{Large}}$ (right) with initial rank $r = 16$.

Table 15: GLUE benchmark suite statistics and evaluation metric for each corpus sorted by the number of examples in the training set.

| Corpus | #Train | #Dev | #Test | Metric |
|--------|--------|------|-------|--------|
| RTE    | 2.5 k  | 276  | 3 k   | Accuracy |
| MRPC   | 3.7 k  | 408  | 1.7 k | Accuracy |
| STS-B  | 7 k    | 1.5 k | 1.4 k | Pearson correlation |
| CoLA   | 8.5 k  | 1 k  | 1 k   | Matthew's correlation |
| SST-2  | 67 k   | 872  | 1.8 k | Accuracy |
| QNLI   | 108 k  | 5.7 k | 5.7 k | Accuracy |
| QQP    | 364 k  | 40 k | 391 k | Accuracy |
| MNLI   | 393 k  | 20 k | 20 k  | Accuracy |

train sizes of the datasets vary drastically, the 1K subset provides equal datasets across tasks. The number of classes also varies from as little as two to almost 400.

## D.2 Implementation details

We implemented a custom pipeline to fine-tune DINOv2-L/14 on VTAB-1K that supports LoRA, DoRA and EVA. To train AdaLora, PiSSA and OLoRA, we integrate their implementation from the `peft` library (Mangrulkar et al., 2022) into our pipeline. This pipeline is designed to be highly parallelizable and to be executed on individual GPUs. A single evaluation run of a L/14 model (all 19 datasets with hyperparameter tuning and evaluation) takes roughly 160 A100 GPU-hours but can be easily parallelized. A g/14 run takes roughly 140 H100 GPU-hours. A single evaluation run consists of 1140 hyperparameter tuning runs (19 datasets * 5 learning rates * 4 ranks * 3 seeds) and 95 evaluation runs (19 datasets * 5 seeds). Details to hyperparameter tuning are described below.

We use the original DINOv2 models (Oquab et al., 2023) and train a classification head on top of the [CLS] token, where we initialize the classification head weights with a normal distribution with $\sigma = 2e\text{-}5$ and bias with zeros. We train the classification head, LoRA matrices and biases. The images are resized to $224 \times 224$ resolution with bicubic interpolation and normalized with the per-channel mean and variance of ImageNet. We train all models with bfloat16 precision using the AdamW optimizer with a weight decay of $0.05$ for 30 epochs. We use a cosine learning rate schedule with a linear warm-up for the first 3 epochs. The batch size is set to 64 where we use gradient accumulation if the batch size does not fit into GPU memory. Full fine-tuning uses a layer-wise lr decay of $0.75$ (Clark et al., 2020).

## D.3 Hyperparameter search

We first fine-tune on the 800 train samples of the VTAB-1K datasets to find the best learning rate for the task. We sweep over `learning_rate` $\in \{2.5e\text{-}3, 1e\text{-}3, 7.5e\text{-}4, 5e\text{-}4, 2.5e\text{-}4\}$ and `rank` $\in \{2, 4, 8, 16\}$ and average the accuracy on the 200 validation samples over 3 different seeds to choose the best learning rate and rank for each dataset. For evaluation, we train on the union of train and validation set using five different seeds and report the average accuracy on the test set.

## D.4 Additional results

We show our main results in Table 20. To complement these results, we report the respective standard deviations in Table 21.

# E Decision Making

## E.1 Dataset statistics

Meta-World (Yu et al., 2020) is an established benchmark in RL for multi-task continuous control. The benchmark consists of 50 challenging robotic tasks simulated using a Sawyer robotic arm in the

Table 16: The best hyperparameters RoBERTa$_{Large}$ and DeBERTav3$_{Base}$ that were found via gridsearch for each task of the GLUE benchmark.

| Method | Dataset | MNLI | SST-2 | MRPC | CoLA | QNLI | QQP | RTE | STS-B |
|---|---|---|---|---|---|---|---|---|---|
| | Optimizer | | | | AdamW | | | | |
| | Warmup Ratio | | | | 0.06 | | | | |
| | LR Schedule | | | | Linear | | | | |
| RoBERTa$_{Large}$ LoRA | Batch Size | 8 | 16 | 8 | 8 | 8 | 8 | 16 | 8 |
| | # Epochs | 10 | 10 | 20 | 20 | 10 | 20 | 20 | 10 |
| | LoRA rank | 2 | 8 | 8 | 4 | 8 | 4 | 2 | 2 |
| | Learning rate | 4e-4 | 1e-3 | 4e-4 | 1e-3 | 1e-3 | 1e-3 | 1e-3 | 4e-4 |
| | LoRA $\alpha$ | | | | 1 | | | | |
| | Max Seq. Len. | | | | 512 | | | | |
| | DDP GPUs | | | | 4 | | | | |
| RoBERTa$_{Large}$ EVA | Batch Size | 8 | 16 | 8 | 8 | 8 | 8 | 16 | 8 |
| | # Epochs | 10 | 10 | 20 | 20 | 10 | 20 | 20 | 10 |
| | LoRA rank | 2 | 2 | 4 | 2 | 16 | 8 | 4 | 4 |
| | Learning rate | 4e-4 | 1e-3 | 4e-4 | 1e-3 | 4e-4 | 1e-3 | 1e-3 | 1e-3 |
| | LoRA $\alpha$ | | | | 1 | | | | |
| | Max Seq. Len. | | | | 512 | | | | |
| | DDP GPUs | | | | 4 | | | | |
| DeBERTav3$_{Base}$ LoRA | Batch Size | 32 | 32 | 16 | 32 | 64 | 32 | 32 | 16 |
| | # Epochs | 30 | 60 | 30 | 80 | 25 | 25 | 80 | 40 |
| | LoRA rank | 8 | 4 | 4 | 8 | 16 | 4 | 4 | 8 |
| | Learning rate | 4e-4 | 1e-3 | 4e-3 | 4e-3 | 4e-3 | 4e-3 | 4e-3 | 4e-3 |
| | LoRA $\alpha$ | | | | 1 | | | | |
| | Max Seq. Len. | | | | 512 | | | | |
| | DDP GPUs | | | | 4 | | | | |
| DeBERTav3$_{Base}$ EVA | Batch Size | 32 | 32 | 16 | 32 | 64 | 32 | 32 | 16 |
| | # Epochs | 30 | 60 | 30 | 80 | 25 | 25 | 80 | 40 |
| | LoRA rank | 8 | 2 | 4 | 8 | 16 | 4 | 2 | 2 |
| | Learning rate | 4e-4 | 4e-4 | 4e-3 | 4e-3 | 4e-3 | 4e-3 | 4e-3 | 4e-3 |
| | LoRA $\alpha$ | | | | 1 | | | | |
| | Max Seq. Len. | | | | 512 | | | | |
| | DDP GPUs | | | | 4 | | | | |

MuJoCo physics engine (Todorov et al., 2012). All 50 tasks in Meta-World share the same underlying robotic arm. Therefore, all tasks share a common state (39-dimensional continuous vector) and action space (6-dimensional). The reward functions in Meta-World are dense and based on the distance of the robotic arm to the target location or objects. All episodes last for 200 environment interactions.

For our experiments on Meta-World, we use the datasets released by Schmied et al. (2024). We follow Wołczyk et al. (2021) and Schmied et al. (2024), and split the 50 tasks into 40 pre-training tasks (MT40) and 10 fine-tuning tasks (CW10). The CW10 tasks are the following.

`hammer-v2`, `push-wall-v2`, `faucet-close-v2`, `push-back-v2`, `stick-pull-v2`, `stick-pull-v2`, `handle-press-side-v2`, `push-v2`, `shelf-place-v2`, `window-close-v2`, and `peg-unplug-side-v2`.

The datasets contain 2M transitions for each of the 50 tasks, which is equivalent to 80M transitions (320M tokens) for all training tasks. The average success rate and rewards for all MT40 tasks are 84% and 1414.62, respectively. We list the statistics per task in Table 22.

### E.2 Implementation details

We implemented our pipeline that supports training on Meta-World on top of the code-base provided by Schmied et al. (2024). Our custom implementation supports training LoRA, DoRA and EVA. Furthermore, we leverage the `peft` library (Mangrulkar et al., 2022) to train the remaining methods.

Table 17: Comparison of LoRA to EVA using RoBERTa$_{\text{Large}}$ on all tasks from GLUE for equal rank budgets. Mean and standard deviation of Matthew's correlation for CoLA, pearson correlation for STS-B, and accuracy for remaining datasets on the development set across 5 seeds are shown.

| Method | CoLA | MRPC | RTE | STS-B | MNLI | QNLI | QQP | SST-2 | Avg |
|---|---|---|---|---|---|---|---|---|---|
| LoRA$_{r=2}$ | $68.0_{\pm1.4}$ | $90.9_{\pm.8}$ | $88.1_{\pm1.1}$ | $92.3_{\pm.1}$ | $91.9_{\pm.1}$ | $94.8_{\pm.3}$ | $90.6_{\pm.1}$ | $96.1_{\pm.1}$ | 89.09 |
| EVA$_{r=2}$ | $69.1_{\pm1.4}$ | $90.8_{\pm.5}$ | $88.2_{\pm.7}$ | $92.5_{\pm.1}$ | $90.8_{\pm.1}$ | $94.9_{\pm.1}$ | $91.9_{\pm.1}$ | $96.2_{\pm.1}$ | 89.30 |
| LoRA$_{r=4}$ | $69.1_{\pm.5}$ | $90.7_{\pm.7}$ | $86.9_{\pm.2}$ | $92.3_{\pm.1}$ | $90.6_{\pm.1}$ | $94.7_{\pm.2}$ | $92.0_{\pm.0}$ | $96.0_{\pm.1}$ | 89.04 |
| EVA$_{r=4}$ | $69.5_{\pm1.4}$ | $91.4_{\pm.8}$ | $88.8_{\pm1.3}$ | $92.6_{\pm.1}$ | $90.7_{\pm.0}$ | $94.9_{\pm.1}$ | $91.8_{\pm.0}$ | $96.1_{\pm.1}$ | 89.48 |
| LoRA$_{r=8}$ | $68.8_{\pm1.0}$ | $91.1_{\pm.6}$ | $87.1_{0.7}$ | $92.2_{\pm.2}$ | $90.6_{\pm.2}$ | $94.8_{\pm.1}$ | $91.8_{\pm.0}$ | $96.2_{\pm.3}$ | 89.08 |
| EVA$_{r=8}$ | $69.0_{\pm1.4}$ | $91.1_{\pm.4}$ | $88.4_{\pm.6}$ | $92.6_{\pm.3}$ | $90.6_{\pm.1}$ | $94.9_{\pm.1}$ | $92.1_{\pm.1}$ | $96.1_{\pm.2}$ | 89.35 |
| LoRA$_{r=16}$ | $68.4_{\pm1.0}$ | $90.5_{\pm.5}$ | $88.0_{\pm.5}$ | $92.3_{\pm.1}$ | $90.6_{\pm.1}$ | $94.8_{\pm.1}$ | $91.9_{\pm.1}$ | $96.1_{\pm.1}$ | 89.08 |
| EVA$_{r=16}$ | $69.1_{\pm.8}$ | $91.2_{\pm.8}$ | $88.0_{\pm.5}$ | $92.6_{\pm.2}$ | $90.7_{\pm.0}$ | $95.0_{\pm.2}$ | $91.8_{\pm.0}$ | $96.2_{\pm.1}$ | 89.33 |

Table 18: Comparison of LoRA to EVA, EVA-Raw, and EVA-Max for RoBERTa$_{\text{Large}}$ on the GLUE tasks CoLA, MRPC, RTE, and STS-B. We report mean and standard deviation of Matthew's correlation for CoLA, pearson correlation for STS-B, matched accuracy for MNLI, and accuracy for remaining tasks across 5 seeds.

| Method | CoLA | MRPC | RTE | STS-B | Avg |
|---|---|---|---|---|---|
| LoRA | $69.1_{\pm.5}$ | $91.1_{\pm0.6}$ | $88.1_{\pm1.1}$ | $92.3_{\pm0.1}$ | 85.2 |
| EVA | $\mathbf{69.5_{\pm1.4}}$ | $\mathbf{91.4_{\pm0.8}}$ | $\mathbf{88.8_{\pm1.2}}$ | $\mathbf{92.6_{\pm0.1}}$ | **85.6** |
| EVA-Raw | $69.4_{\pm1.1}$ | $91.0_{\pm0.9}$ | $88.2_{\pm0.3}$ | $92.5_{\pm0.2}$ | 85.3 |
| EVA-Max | $69.1_{\pm0.5}$ | $91.2_{\pm0.5}$ | $88.4_{\pm1.2}$ | $92.5_{\pm0.2}$ | 85.3 |

For our experiments on Meta-World, we use a GPT2-like network architecture (Radford et al., 2019) with 4 Transformer layers, 8 heads, and hidden dimension of 512 resulting in 16M parameters. We use a context of 50 time steps, which amounts to a sequence length of 200, as each timestep contains states, actions, rewards, and RTGs. We embed states, actions, rewards, and return-to-gos (RTGs) using separate linear embedding layers per modality, as proposed by Chen et al. (2021a). We train with a batch size of 128 using a constant learning rate of $1e^{-4}$, 4000 linear warm-up steps followed by a cosine decay to $1e^{-6}$, using the AdamW optimizer (Loshchilov & Hutter, 2017). We employ a gradient clipping of 0.25, a weight decay of 0.01, and a dropout rate of 0.2. Our DT implementation employs global position embedding. For each task, we set the target return to the maximum return achieved in the respective training datasets, as proposed by (Schmied et al., 2024). Furthermore, we employ mixed precision (Micikevicius et al., 2017) and flash attention (Dao, 2023) to speed up the training.

We first **pre-train** a DT on all MT40 tasks (80M transitions) for 1M updates via next-action prediction by minimizing the mean-squared error. The resulting pre-trained model achieves an average success rate of 80% across all MT40 tasks. Then we **fine-tune** the DT on each of the CW10 downstream tasks for 100K updates with the same set of hyperparameters as used for pre-training. We run all our experiments on a public research cluster with 4xA100-40GB GPU nodes. A single EVA fine-tuning run for one task takes roughly 1 hour on an A100.

## E.3 Hyperparameter search

In line with previous experiments, we tune the rank for LoRA, DoRA, AdaLora and EVA, rank $\in$ $\{2, 4, 8, 16\}$. Further, we sweep over the same learning rates as for the GLUE tasks.

## E.4 Additional results

In Table 24, we show the full comparison of all the methods on CW10. EVA+DoRA consistently outperforms all competitors for the different rank budgets.

Table 19: Category, train size and classes of the VTAB-1K dataset.

| Category | Dataset | Train size | Classes |
|---|---|---:|---:|
| Natural | Caltech101 (Fei-Fei et al., 2006) | 3060 | 102 |
| Natural | CIFAR-100 (Krizhevsky, 2009) | 50000 | 100 |
| Natural | DTD (Cimpoi et al., 2014) | 3760 | 47 |
| Natural | Flowers102 (Nilsback & Zisserman, 2008) | 2040 | 102 |
| Natural | Pets (Parkhi et al., 2012) | 3680 | 37 |
| Natural | Sun397 (Xiao et al., 2010) | 87003 | 397 |
| Natural | SVHN (Netzer et al., 2011) | 73257 | 10 |
| Specialized | EuroSAT (Helber et al., 2019) | 21600 | 10 |
| Specialized | Resisc45 (Cheng et al., 2017) | 25200 | 45 |
| Specialized | Patch Camelyon (Veeling et al., 2018) | 294912 | 2 |
| Specialized | Retinopathy (Kaggle & EyePacs, 2015) | 46032 | 5 |
| Structured | Clevr/count (Johnson et al., 2017) | 70000 | 8 |
| Structured | Clevr/distance (Johnson et al., 2017) | 70000 | 6 |
| Structured | dSprites/location (Matthey et al., 2017) | 663552 | 16 |
| Structured | dSprites/orientation (Matthey et al., 2017) | 663552 | 16 |
| Structured | SmallNORB/azimuth (LeCun et al., 2004) | 36450 | 18 |
| Structured | SmallNORB/elevation (LeCun et al., 2004) | 36450 | 9 |
| Structured | DMLab (Beattie et al., 2016) | 88178 | 6 |
| Structured | KITTI/distance (Geiger et al., 2013) | 5711 | 4 |

Table 20: Fine-tuning DINOv2-g/14 on the VTAB-1K benchmark. Best average performance is highlighted in boldface. We report average accuracy across five seeds.

| | Natural | | | | | | | Specialized | | | | Structured | | | | | | | | |
|---|---|---|---|---|---|---|---|---|---|---|---|---|---|---|---|---|---|---|---|---|
| | Cifar100 | Caltech101 | DTD | Flower102 | Pets | SVHN | Sun397 | Camelyon | EuroSAT | Resisc45 | Retinopathy | Clevr-Count | Clevr-Dist | DMLab | KITTI-Dist | dSpr-Loc | dSpr-Ori | sNORB-Azim | sNORB-Ele | Average |
| FFT | 73.1 | 89.7 | 78.4 | 99.7 | 92.2 | 89.5 | 55.5 | 74.8 | 95.0 | 88.2 | 70.5 | 93.6 | 64.2 | **63.6** | 68.8 | 92.0 | **64.3** | **50.2** | **56.8** | 76.8 |
| LoRA | 85.9 | 92.2 | 82.2 | 99.7 | 94.5 | 64.1 | **63.6** | **88.8** | **97.0** | **92.6** | 76.6 | **97.7** | 65.3 | 62.1 | 83.6 | 90.6 | 63.0 | 37.1 | 52.3 | 78.4 |
| AdaLoRA | 85.4 | 92.5 | 81.4 | 99.7 | 95.2 | 90.5 | 62.2 | 87.1 | 96.4 | 91.2 | 76.6 | 94.4 | 64.4 | 60.3 | 83.7 | 85.4 | 61.0 | 32.9 | 46.0 | 78.2 |
| PiSSA | 85.5 | 93.6 | 82.3 | 99.7 | 94.6 | 92.8 | 62.3 | 87.1 | 96.6 | 91.9 | 76.3 | 95.0 | **66.3** | 63.2 | **84.9** | 90.5 | 60.1 | 36.3 | 48.6 | 79.4 |
| OLoRA | 85.5 | 93.0 | 82.1 | 99.7 | 95.1 | 78.3 | 62.1 | 86.7 | 96.3 | 91.9 | **76.8** | 94.3 | 66.0 | 62.4 | 71.3 | 89.0 | 60.9 | 34.3 | 49.5 | 77.6 |
| EVA | 85.6 | **93.9** | 82.2 | 99.7 | **95.9** | 93.2 | **63.6** | 86.8 | 96.6 | 92.3 | 76.1 | 96.1 | 65.1 | 61.1 | 83.3 | 91.4 | 61.6 | 35.0 | 55.0 | **79.7** |
| DoRA | 85.9 | 92.7 | 82.1 | 99.7 | 95.2 | 34.4 | 61.4 | 88.6 | 96.8 | 92.4 | **76.8** | 97.6 | 65.4 | 62.7 | 84.4 | 43.2 | 63.1 | 37.8 | 52.6 | 74.4 |
| EVA+DoRA | **86.2** | 92.1 | 81.9 | 99.7 | 94.9 | **93.8** | 62.4 | 88.3 | 96.6 | **92.6** | 76.7 | 97.2 | 65.5 | 54.1 | 83.7 | **93.3** | 62.3 | 37.5 | 54.5 | 79.6 |

# F   Incremental SVD convergence analysis

For simplicity, assume that $A = X_0^{i\top}$ and $B = X_1^{i\top}$ are two batches of activations for the weight matrix $W^i$ obtained by passing two subsequent batches of downstream data through the model. The aim is now to compute the SVD of the concatenated activation matrix $[AB] = U'\Sigma'V'^{\top}$ in constant memory. Further, we obtain $A = U_t\Sigma_t V_t^{\top}$ via SVD. Now let $\tilde{B}$ be the component of $B$ that is orthogonal to $U$, which can be obtained by QR decomposition or by $\tilde{B} = \mathrm{orth}(B - UU^{\top}B)$, where $\mathrm{orth}(\cdot)$ performs orthogonalization. Then the SVD of the concatenated activation matrix can be expressed in partitioned form as

$$[AB] = \begin{bmatrix} U & \tilde{B} \end{bmatrix} \begin{bmatrix} \Sigma & U^{\top}B \\ 0 & \tilde{B}^{\top}B \end{bmatrix} \begin{bmatrix} V^{\top} & 0 \\ 0 & I \end{bmatrix}. \tag{7}$$

Table 21: Standard deviations for the VTAB-1K results (Table 20) over 5 seeds.

| | Natural | | | | | | | Specialized | | | | Structured | | | | | | | | |
|---|---|---|---|---|---|---|---|---|---|---|---|---|---|---|---|---|---|---|---|---|
| | Cifar100 | Caltech101 | DTD | Flower102 | Pets | SVHN | Sun397 | Camelyon | EuroSAT | Resisc45 | Retinopathy | Clevr-Count | Clevr-Dist | DMLab | KITTI-Dist | dSpr-Loc | dSpr-Ori | sNORB-Azim | sNORB-Ele | Average |
| FFT | 1.5 | 1.1 | 1.6 | 0.0 | 0.4 | 1.2 | 0.9 | 14.9 | 0.4 | 0.6 | 2.7 | 1.7 | 0.9 | 1.2 | 23.6 | 0.5 | 0.4 | 1.6 | 1.9 | 3.0 |
| LoRA | 0.2 | 0.4 | 0.2 | 0.0 | 0.3 | 36.4 | **0.1** | 0.5 | 0.3 | 0.1 | 0.4 | **0.2** | 0.3 | 0.5 | 1.2 | 0.4 | 0.4 | 0.7 | 0.4 | 2.3 |
| AdaLoRA | **0.0** | **0.2** | 0.4 | 0.0 | 0.1 | 0.4 | **0.1** | 0.3 | 0.3 | 0.2 | 0.3 | 0.3 | 0.2 | 0.3 | 0.8 | 0.8 | 0.3 | 0.3 | 0.4 | **0.3** |
| PiSSA | 0.2 | 0.4 | 0.3 | 0.0 | 0.2 | 0.5 | 0.2 | 0.7 | 0.2 | **0.1** | 0.4 | 0.3 | 0.4 | **0.2** | 0.7 | **0.3** | 0.5 | 0.4 | 0.5 | 0.3 |
| OLoRA | 0.3 | 0.3 | 0.4 | 0.0 | 0.3 | 29.4 | 0.1 | 0.3 | **0.1** | 0.2 | **0.2** | 0.5 | **0.1** | 0.3 | 24.6 | 0.3 | 0.4 | 0.3 | 0.8 | 3.1 |
| EVA | 0.2 | 0.5 | **0.2** | 0.0 | **0.1** | **0.3** | **0.1** | **0.3** | 0.2 | 0.3 | 0.4 | 0.5 | 0.3 | 0.6 | 0.6 | 0.5 | 0.5 | **0.2** | 0.5 | 0.3 |
| DoRA | 0.1 | 0.2 | 0.5 | 0.0 | 0.2 | 29.7 | 0.4 | 0.7 | 0.1 | 0.2 | 0.4 | 0.4 | 0.3 | 0.3 | **0.6** | 36.2 | 0.5 | 0.3 | **0.3** | 3.8 |
| EVA+DoRA | 0.2 | 1.3 | 0.6 | 0.0 | 0.3 | 0.5 | 0.3 | 0.4 | 0.2 | 0.3 | 0.3 | 0.4 | 0.4 | 12.8 | 1.3 | 2.5 | **0.3** | 0.6 | 0.6 | 1.2 |

By setting $R = \begin{bmatrix} \Sigma & U^\top B \\ 0 & \tilde{B}B \end{bmatrix}$, we can obtain SVD of the concatenated activation matrix by performing SVD on $R$, $R = \tilde{U}\tilde{\Sigma}\tilde{V}^\top$, which is constant in time and memory as we only need to compute $U'$ and $\Sigma'$, which do not scale with the number of data samples. Hence, we perform

$$[A; B] = \left(\begin{bmatrix} U; \tilde{B} \end{bmatrix} \tilde{U}\right) \tilde{\Sigma} \left(\tilde{V}^\top \begin{bmatrix} V^\top & 0 \\ 0 & I \end{bmatrix}\right), \tag{8}$$

and subsequently obtain $U' = \begin{bmatrix} U\tilde{B} \end{bmatrix} \tilde{U}$ and $\Sigma' = \tilde{\Sigma}$.

As this algorithm incrementally updates the $U$ and $\Sigma$ components, we need to keep track of changing mean and variance estimates. For the mean, this is trivial, but the computation of running variances can introduce numerical instabilities. To counteract this, *young and cramer update* is commonly employed (Chan et al., 1983). The supporting proof that the covariance matrix of the original data matrix is equal to the covariance matrix of the concatenated matrix up to a constant factor is given in Ross et al. (2008). In our example, the left-singular values $U$ do not scale with the number of samples. However, in our case we have $A = X_t^i$ and $B = X_{t+1}^i$, i.e. transposed data matrices, therefore it is the right-singular values $V$ that do not depend on the number of samples and can be incrementally updated in constant time and memory. We show pseudocode for the incremental SVD algorithm in Algorithm 2. In the following sections, we analyze the behavior of this algorithm under different conditions, i.e. different batch sizes, etc.

### F.1 Complexity

The SVD computation introduces computational overhead in the initial training stage. Since we do not require gradient computation or storing of optimizer states, there is no overhead in terms of memory. SVD has a time complexity of $\mathcal{O}(\min(b^2d, bd^2))$ that can be reduced to $\mathcal{O}(k^2b)$ for $k << d$ by performing truncated SVD Halko et al. (2011). Let $T$ be the number of minibatches until all components are converged for $N$ weight matrices, then the time complexity is $\mathcal{O}(NTk^2b)$. In other words, the complexity scales linearly with the number of weight matrices and the number of minibatches. To speed up the computation of SVD, we provide an implementation that runs entirely on GPU.

### F.2 Batch Size invariance

We perform an analysis of the convergence of the components obtained via SVD. Specifically, we investigate the difference in components according to cosine similarity across different batch sizes. Previously, we have seen that the components obtained across different batch orderings are heavily

Table 22: Dataset statistics for all MT40 tasks from Schmied et al. (2024).

| Task | $|\mathcal{S}|$ | $|\mathcal{A}|$ | Success Rate | Reward |
|------|------|------|------|------|
| assembly-v2 | 39 | 4 | 0.0 | 1206.9 |
| basketball-v2 | 39 | 4 | 0.9 | 1375.95 |
| bin-picking-v2 | 39 | 4 | 0.0 | 474.81 |
| box-close-v2 | 39 | 4 | 0.0 | 759.15 |
| button-press-topdown-v2 | 39 | 4 | 1.0 | 1299.24 |
| button-press-topdown-wall-v2 | 39 | 4 | 1.0 | 1296.16 |
| button-press-v2 | 39 | 4 | 1.0 | 1430.44 |
| button-press-wall-v2 | 39 | 4 | 1.0 | 1508.16 |
| coffee-button-v2 | 39 | 4 | 1.0 | 1499.17 |
| coffee-pull-v2 | 39 | 4 | 1.0 | 1313.88 |
| coffee-push-v2 | 39 | 4 | 0.6 | 508.14 |
| dial-turn-v2 | 39 | 4 | 0.8 | 1674.29 |
| disassemble-v2 | 39 | 4 | 1.0 | 1396.55 |
| door-close-v2 | 39 | 4 | 1.0 | 1535.4 |
| door-lock-v2 | 39 | 4 | 1.0 | 1712.65 |
| door-open-v2 | 39 | 4 | 1.0 | 1544.32 |
| door-unlock-v2 | 39 | 4 | 1.0 | 1733.64 |
| drawer-close-v2 | 39 | 4 | 1.0 | 1845.92 |
| drawer-open-v2 | 39 | 4 | 1.0 | 1710.65 |
| faucet-open-v2 | 39 | 4 | 0.9 | 1727.98 |
| hand-insert-v2 | 39 | 4 | 1.0 | 1607.17 |
| handle-press-v2 | 39 | 4 | 1.0 | 1854.79 |
| handle-pull-side-v2 | 39 | 4 | 1.0 | 1613.72 |
| handle-pull-v2 | 39 | 4 | 1.0 | 1581.75 |
| lever-pull-v2 | 39 | 4 | 1.0 | 1449.05 |
| peg-insert-side-v2 | 39 | 4 | 1.0 | 1545.19 |
| pick-out-of-hole-v2 | 39 | 4 | 1.0 | 1435.64 |
| pick-place-v2 | 39 | 4 | 0.0 | 6.59 |
| pick-place-wall-v2 | 39 | 4 | 0.1 | 702.59 |
| plate-slide-back-side-v2 | 39 | 4 | 1.0 | 1766.24 |
| plate-slide-back-v2 | 39 | 4 | 1.0 | 1773.56 |
| plate-slide-side-v2 | 39 | 4 | 1.0 | 1663.35 |
| plate-slide-v2 | 39 | 4 | 1.0 | 1667.35 |
| reach-v2 | 39 | 4 | 1.0 | 1858.99 |
| reach-wall-v2 | 39 | 4 | 1.0 | 1831.14 |
| soccer-v2 | 39 | 4 | 0.4 | 445.84 |
| stick-push-v2 | 39 | 4 | 1.0 | 1470.71 |
| sweep-into-v2 | 39 | 4 | 1.0 | 1761.69 |
| sweep-v2 | 39 | 4 | 1.0 | 1458.35 |
| window-open-v2 | 39 | 4 | 1.0 | 1537.59 |
| Average | - | - | $0.84 \pm 0.34$ | $1414.62 \pm 439.39$ |

Table 23: Results for single task fine-tuning experiments on the Meta-World benchmark. We report mean success rates and standard error across three seeds for every task.

| | faucet-close | hammer | handle-press | peg-unplug | push-back | push | push-wall | shelf-place | stick-pull | window-close | Average |
|---|---|---|---|---|---|---|---|---|---|---|---|
| FFT | $1.0_{\pm.0}$ | $0.97_{\pm.03}$ | $1.0_{\pm.0}$ | $0.77_{\pm.05}$ | $0.87_{\pm.05}$ | $1.0_{\pm.0}$ | $1.0_{\pm.0}$ | $1.0_{\pm.0}$ | $0.63_{\pm.03}$ | $1.0_{\pm.0}$ | $0.92$ |
| LoRA | $1.0_{\pm.0}$ | $1.0_{\pm.0}$ | $1.0_{\pm.0}$ | $0.6_{\pm.05}$ | $0.63_{\pm.1}$ | $1.0_{\pm.0}$ | $1.0_{\pm.0}$ | $1.0_{\pm.0}$ | $0.4_{\pm.09}$ | $1.0_{\pm.0}$ | $0.86$ |
| AdaLoRA | $1.0_{\pm.0}$ | $0.97_{\pm.03}$ | $1.0_{\pm.0}$ | $0.4_{\pm.09}$ | $0.57_{\pm.1}$ | $0.97_{\pm.03}$ | $0.97_{\pm.03}$ | $1.0_{\pm.0}$ | $0.13_{\pm.07}$ | $1.0_{\pm.0}$ | $0.80$ |
| PiSSA | $1.0_{\pm.0}$ | $1.0_{\pm.0}$ | $1.0_{\pm.0}$ | $0.43_{\pm0.11}$ | $0.57_{\pm0.03}$ | $1.0_{\pm.0}$ | $1.0_{\pm.0}$ | $1.0_{\pm.0}$ | $0.53_{\pm0.1}$ | $1.0_{\pm.0}$ | $0.85$ |
| OLoRA | $1.0_{\pm.0}$ | $0.97_{\pm0.03}$ | $1.0_{\pm.0}$ | $0.57_{\pm0.1}$ | $0.63_{\pm0.03}$ | $1.0_{\pm.0}$ | $1.0_{\pm.0}$ | $1.0_{\pm.0}$ | $0.6_{\pm0.12}$ | $1.0_{\pm.0}$ | $0.88$ |
| EVA | $1.0_{\pm.0}$ | $0.97_{\pm.03}$ | $1.0_{\pm.0}$ | $0.63_{\pm.03}$ | $0.77_{\pm.05}$ | $1.0_{\pm.0}$ | $1.0_{\pm.0}$ | $1.0_{\pm.0}$ | $0.63_{\pm.07}$ | $1.0_{\pm.0}$ | $0.90$ |
| DoRA | $1.0_{\pm.0}$ | $1.0_{\pm.0}$ | $1.0_{\pm.0}$ | $0.6_{\pm1.2}$ | $1.0_{\pm.0}$ | $1.0_{\pm.0}$ | $1.0_{\pm.0}$ | $1.0_{\pm.0}$ | $\mathbf{0.67_{\pm1.5}}$ | $1.0_{\pm.0}$ | $0.93$ |
| EVA+DoRA | $1.0_{\pm.0}$ | $1.0_{\pm.0}$ | $1.0_{\pm.0}$ | $\mathbf{0.8_{\pm.08}}$ | $1.0_{\pm.0}$ | $1.0_{\pm.0}$ | $1.0_{\pm.0}$ | $1.0_{\pm.0}$ | $0.63_{\pm.03}$ | $1.0_{\pm.0}$ | $\mathbf{0.94}$ |

Table 24: Rank-wise comparison for all methods on CW10. We fine-tune a 12M DT on 10 tasks individually and report the mean success rates/rewards ($\pm$ standard error) for every task.

| Method | Rank | faucet-close | hammer | handle-press-side | peg-unplug-side | push-back | push | push-wall | shelf-place | stick-pull | window-close | Average |
|---|---|---|---|---|---|---|---|---|---|---|---|---|
| FFT | - | $0.97_{\pm0.03}$ | $0.93_{\pm0.03}$ | $1.0_{\pm0.0}$ | $0.6_{\pm0.05}$ | $0.7_{\pm0.12}$ | $1.0_{\pm0.0}$ | $0.93_{\pm0.03}$ | $1.0_{\pm0.0}$ | $0.57_{\pm0.07}$ | $1.0_{\pm0.0}$ | $0.87_{\pm0.03}$ |
| LoRA | 2 | $1.0_{\pm0.0}$ | $1.0_{\pm0.0}$ | $1.0_{\pm0.0}$ | $0.6_{\pm0.05}$ | $0.57_{\pm0.07}$ | $0.97_{\pm0.03}$ | $0.93_{\pm0.03}$ | $1.0_{\pm0.0}$ | $0.37_{\pm0.1}$ | $1.\pm0.0$ | $0.84_{\pm0.04}$ |
| | 4 | $1.0_{\pm0.0}$ | $0.97_{\pm0.03}$ | $1.0_{\pm0.0}$ | $0.47_{\pm0.12}$ | $0.63_{\pm0.1}$ | $0.97_{\pm0.03}$ | $1.0_{\pm0.0}$ | $1.0_{\pm0.0}$ | $0.23_{\pm0.12}$ | $1.0_{\pm0.0}$ | $0.83_{\pm0.05}$ |
| | 8 | $1.0_{\pm0.0}$ | $0.97_{\pm0.03}$ | $1.0_{\pm0.0}$ | $0.43_{\pm0.05}$ | $0.4_{\pm0.09}$ | $0.97_{\pm0.03}$ | $0.93_{\pm0.03}$ | $1.0_{\pm0.0}$ | $0.23_{\pm0.12}$ | $1.0_{\pm0.0}$ | $0.79_{\pm0.06}$ |
| | 16 | $1.0_{\pm0.0}$ | $0.97_{\pm0.03}$ | $1.0_{\pm0.0}$ | $0.43_{\pm0.03}$ | $0.47_{\pm0.03}$ | $1.0_{\pm0.0}$ | $0.97_{\pm0.03}$ | $1.0_{\pm0.0}$ | $0.4_{\pm0.09}$ | $1.0_{\pm0.0}$ | $0.82_{\pm0.05}$ |
| DoRA | 2 | $1.0_{\pm0.0}$ | $1.0_{\pm0.0}$ | $1.0_{\pm0.0}$ | $0.57_{\pm0.05}$ | $1.0_{\pm0.0}$ | $1.0_{\pm0.0}$ | $1.0_{\pm0.0}$ | $1.0_{\pm0.0}$ | $0.33_{\pm0.11}$ | $1.0_{\pm0.0}$ | $0.89_{\pm0.04}$ |
| | 4 | $1.0_{\pm0.0}$ | $1.0_{\pm0.0}$ | $1.0_{\pm0.0}$ | $0.6_{\pm0.12}$ | $1.0_{\pm0.0}$ | $1.0_{\pm0.0}$ | $1.0_{\pm0.0}$ | $1.0_{\pm0.0}$ | $0.43_{\pm0.12}$ | $1.0_{\pm0.0}$ | $0.9_{\pm0.04}$ |
| | 8 | $1.0_{\pm0.0}$ | $1.0_{\pm0.0}$ | $1.0_{\pm0.0}$ | $0.47_{\pm0.12}$ | $0.93_{\pm0.05}$ | $1.0_{\pm0.0}$ | $1.0_{\pm0.0}$ | $1.0_{\pm0.0}$ | $0.57_{\pm0.15}$ | $1.0_{\pm0.0}$ | $0.9_{\pm0.04}$ |
| | 16 | $1.0_{\pm0.0}$ | $1.0_{\pm0.0}$ | $1.0_{\pm0.0}$ | $0.57_{\pm0.12}$ | $1.0_{\pm0.0}$ | $1.0_{\pm0.0}$ | $1.0_{\pm0.0}$ | $1.0_{\pm0.0}$ | $0.67_{\pm0.15}$ | $1.0_{\pm0.0}$ | $0.92_{\pm0.03}$ |
| AdaLoRA | 2 | $1.0_{\pm0.0}$ | $0.97_{\pm0.03}$ | $1.0_{\pm0.0}$ | $0.37_{\pm0.05}$ | $0.37_{\pm0.05}$ | $0.93_{\pm0.05}$ | $0.97_{\pm0.03}$ | $1.0_{\pm0.0}$ | $0.13_{\pm0.07}$ | $1.0_{\pm0.0}$ | $0.77_{\pm0.06}$ |
| | 4 | $1.0_{\pm0.0}$ | $0.97_{\pm0.03}$ | $1.0_{\pm0.0}$ | $0.37_{\pm0.07}$ | $0.57_{\pm0.1}$ | $0.97_{\pm0.03}$ | $0.9_{\pm0.08}$ | $1.0_{\pm0.0}$ | $0.13_{\pm0.07}$ | $1.0_{\pm0.0}$ | $0.79_{\pm0.06}$ |
| | 8 | $1.0_{\pm0.0}$ | $0.97_{\pm0.03}$ | $1.0_{\pm0.0}$ | $0.3_{\pm0.05}$ | $0.57_{\pm0.14}$ | $0.93_{\pm0.03}$ | $0.87_{\pm0.07}$ | $1.0_{\pm0.0}$ | $0.0_{\pm0.0}$ | $1.0_{\pm0.0}$ | $0.76_{\pm0.06}$ |
| | 16 | $1.0_{\pm0.0}$ | $0.97_{\pm0.03}$ | $1.0_{\pm0.0}$ | $0.4_{\pm0.09}$ | $0.57_{\pm0.12}$ | $0.97_{\pm0.03}$ | $0.93_{\pm0.05}$ | $1.0_{\pm0.0}$ | $0.0_{\pm0.0}$ | $1.0_{\pm0.0}$ | $0.78_{\pm0.06}$ |
| OLoRA | 2 | $1.0_{\pm0.0}$ | $0.9_{\pm0.05}$ | $1.0_{\pm0.0}$ | $0.47_{\pm0.03}$ | $0.33_{\pm0.03}$ | $0.97_{\pm0.03}$ | $0.97_{0.03}$ | $1.0_{\pm0.0}$ | $0.27_{\pm0.11}$ | $1.0_{\pm0.0}$ | $0.79_{\pm0.05}$ |
| | 4 | $1.0_{\pm0.0}$ | $0.9_{\pm0.05}$ | $1.0_{\pm0.0}$ | $0.43_{\pm0.03}$ | $0.63_{\pm0.12}$ | $1.0_{\pm0.0}$ | $1.0_{0.0}$ | $1.0_{\pm0.0}$ | $0.6_{\pm0.12}$ | $1.0_{\pm0.0}$ | $0.86_{\pm0.04}$ |
| | 8 | $1.0_{\pm0.0}$ | $0.97_{\pm0.03}$ | $1.0_{\pm0.0}$ | $0.57_{\pm0.1}$ | $0.5_{\pm0.08}$ | $1.0_{\pm0.0}$ | $1.0_{0.0}$ | $1.0_{\pm0.0}$ | $0.53_{\pm0.14}$ | $1.0_{\pm0.0}$ | $0.86_{\pm0.04}$ |
| | 16 | $1.0_{\pm0.0}$ | $0.97_{\pm0.03}$ | $1.0_{\pm0.0}$ | $0.4_{\pm0.05}$ | $0.63_{\pm0.03}$ | $1.0_{\pm0.0}$ | $1.0_{0.0}$ | $1.0_{\pm0.0}$ | $0.43_{\pm0.05}$ | $1.0_{\pm0.0}$ | $0.84_{\pm0.04}$ |
| PiSSA | 2 | $1.0_{\pm0.0}$ | $0.97_{\pm0.03}$ | $1.0_{\pm0.0}$ | $0.43_{\pm0.11}$ | $0.53_{\pm0.07}$ | $0.97_{\pm0.03}$ | $0.9_{0.08}$ | $1.0_{\pm0.0}$ | $0.33_{\pm0.17}$ | $1.0_{\pm0.0}$ | $0.81_{\pm0.05}$ |
| | 4 | $1.0_{\pm0.0}$ | $1.0_{\pm0.0}$ | $1.0_{\pm0.0}$ | $0.37_{\pm0.07}$ | $0.7_{\pm0.05}$ | $0.97_{\pm0.03}$ | $1.0_{0.0}$ | $1.0_{\pm0.0}$ | $0.07_{\pm0.05}$ | $1.0_{\pm0.0}$ | $0.81_{\pm0.06}$ |
| | 8 | $1.0_{\pm0.0}$ | $0.97_{\pm0.03}$ | $1.0_{\pm0.0}$ | $0.3_{\pm0.0}$ | $0.57_{\pm0.03}$ | $0.97_{\pm0.03}$ | $1.0_{0.0}$ | $1.0_{\pm0.0}$ | $0.53_{\pm0.1}$ | $1.0_{\pm0.0}$ | $0.83_{\pm0.05}$ |
| | 16 | $1.0_{\pm0.0}$ | $0.93_{\pm0.03}$ | $1.0_{\pm0.0}$ | $0.33_{\pm0.12}$ | $0.47_{\pm0.03}$ | $1.0_{\pm0.0}$ | $0.97_{0.03}$ | $1.0_{\pm0.0}$ | $0.47_{\pm0.11}$ | $1.0_{\pm0.0}$ | $0.82_{\pm0.05}$ |
| EVA | 2 | $1.0_{\pm0.0}$ | $0.97_{\pm0.03}$ | $1.0_{\pm0.0}$ | $0.43_{\pm0.07}$ | $0.77_{\pm0.05}$ | $0.97_{\pm0.03}$ | $1.0_{\pm0.0}$ | $1.0_{\pm0.0}$ | $0.63_{\pm0.07}$ | $1.0_{\pm0.0}$ | $0.88_{\pm0.04}$ |
| | 4 | $1.0_{\pm0.0}$ | $0.97_{\pm0.03}$ | $1.0_{\pm0.0}$ | $0.43_{\pm0.05}$ | $0.47_{\pm0.12}$ | $1.0_{\pm0.0}$ | $0.97_{\pm0.03}$ | $1.0_{\pm0.0}$ | $0.23_{\pm0.05}$ | $1.0_{\pm0.0}$ | $0.81_{\pm0.05}$ |
| | 8 | $1.0_{\pm0.0}$ | $0.97_{\pm0.03}$ | $1.0_{\pm0.0}$ | $0.63_{\pm0.03}$ | $0.7_{\pm0.08}$ | $1.0_{\pm0.0}$ | $1.0_{\pm0.0}$ | $1.0_{\pm0.0}$ | $0.23_{\pm0.03}$ | $1.0_{\pm0.0}$ | $0.85_{\pm0.05}$ |
| | 16 | $1.0_{\pm0.0}$ | $0.97_{\pm0.03}$ | $1.0_{\pm0.0}$ | $0.53_{\pm0.03}$ | $0.77_{\pm0.07}$ | $1.0_{\pm0.0}$ | $1.0_{\pm0.0}$ | $1.0_{\pm0.0}$ | $0.0_{\pm0.0}$ | $1.0_{\pm0.0}$ | $0.83_{\pm0.06}$ |
| EVA + DoRA | 2 | $1.0_{\pm0.0}$ | $1.0_{\pm0.0}$ | $1.0_{\pm0.0}$ | $0.8_{\pm0.08}$ | $0.97_{\pm0.03}$ | $1.0_{\pm0.0}$ | $1.0_{\pm0.0}$ | $1.0_{\pm0.0}$ | $0.43_{\pm0.12}$ | $1.0_{\pm0.0}$ | $0.92_{\pm0.03}$ |
| | 4 | $1.0_{\pm0.0}$ | $1.0_{\pm0.0}$ | $1.0_{\pm0.0}$ | $0.8_{\pm0.05}$ | $0.93_{\pm0.03}$ | $1.0_{\pm0.0}$ | $1.0_{\pm0.0}$ | $1.0_{\pm0.0}$ | $0.63_{\pm0.03}$ | $1.0_{\pm0.0}$ | $0.94_{\pm0.02}$ |
| | 8 | $1.0_{\pm0.0}$ | $1.0_{\pm0.0}$ | $1.0_{\pm0.0}$ | $0.63_{\pm0.19}$ | $0.87_{\pm0.07}$ | $1.0_{\pm0.0}$ | $1.0_{\pm0.0}$ | $1.0_{\pm0.0}$ | $0.57_{\pm0.03}$ | $1.0_{\pm0.0}$ | $0.91_{\pm0.04}$ |
| | 16 | $1.0_{\pm0.0}$ | $1.0_{\pm0.0}$ | $1.0_{\pm0.0}$ | $0.67_{\pm0.2}$ | $1.0_{\pm0.0}$ | $1.0_{\pm0.0}$ | $1.0_{\pm0.0}$ | $1.0_{\pm0.0}$ | $0.5_{\pm0.16}$ | $1.0_{\pm0.0}$ | $0.92_{\pm0.04}$ |

correlated. In Figure 11 (right), we visualize the cosine similarities between the SVD components for different batch sizes, namely 4, 8, 16, and 32 for Llama-2-7B on the MetaMathQA dataset. We observe that the components correlate strongly and remain mostly invariant to the batch size. This indicates that smaller batch sizes may be used for obtaining the initialization, which results in less computational overhead. In the case of Llama-2-7B on MetaMathQA, this means that we can use a batch size of 4 since it induces a computational overhead of around 100 seconds. Afterwards, we can continue the fine-tuning process with a larger batch size.

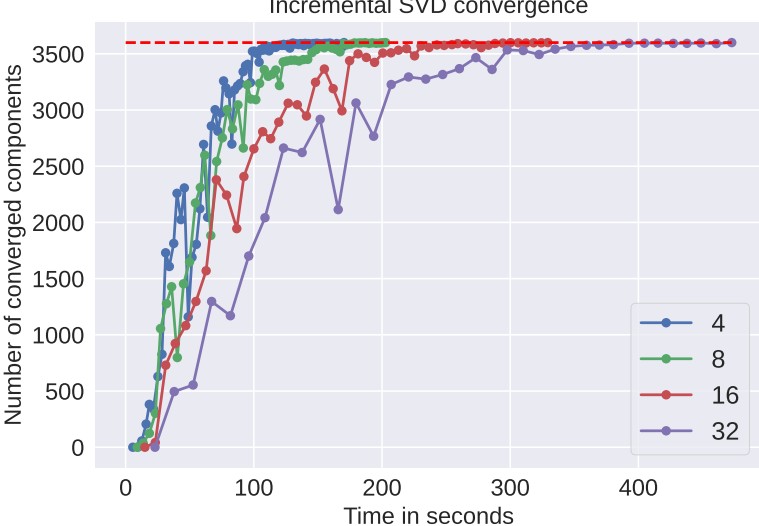

Figure 11: Time in seconds until convergence of incremental SVD components for different batch sizes for Llama-2-7B on the MetaMathQA dataset. The dashed line indicates the total number of components.

### F.3 Convergence speed

The data-driven initialization of EVA relies on incremental SVD on minibatches of activations in the initial training stage. In Figure 11, we show that this process converges for Llama-2-7B on MetaMathQA for different minibatch sizes. Using a minibatch size of 4 the computation for EVA's initialization lasts for approximately 80 seconds, which corresponds to around 90 minibatches. For a batch size of 32 the computation of the SVD components takes around 500 seconds.

### F.4 Batch order invariance

In Figure 12 left, we additionally show that the main components obtained via SVD mostly remain consistent across different batch orders for a batch size of 4, again for Llama-2-7B on MetaMathQA. To this end, we plot the cosine similarity between components obtained via incremental SVD after rank redistribution. These results indicate that these models exhibit certain activation patterns that remain consistent across different batch orders, which leads to a robust initialization for EVA.

### F.5 Excluding ignored tokens for SVD

For some datasets we notice that masking out tokens for the SVD computation which are ignored for the loss calculation during fine-tunine can be advantageous. However, this can result in a significant reduction of the effective batch size for SVD if the number of completion tokens is small. An example where this is the case in our experiments is the common-sense reasoning tasks which have long prompts, but completion tokens are only one word per sample. This setting can lead to cases where SVD does not converge for lower batch sizes. We therefore do not mask out the prompt tokens in our experiments. Another setting where masking ignored tokens can be advantageous is multi-turn conversation where the model is only trained on the assistant tokens. To achieve the results in Table 13 we mask out user tokens together with the prompt for the SVD computation.

## G  Rank redistribution analysis

To illuminate the rank redistribution process, we visualize the resulting ranks for each weight matrix after SVD for Llama-2-7B on the MetaMathQA dataset for different values of $\rho$. Setting $\rho = 1$ results in a uniform rank distribution as in standard LoRA. However, setting $\rho > 1$ alters the number of ranks per weight matrix. In Figure 13 we visualize the number of ranks assigned to each weight

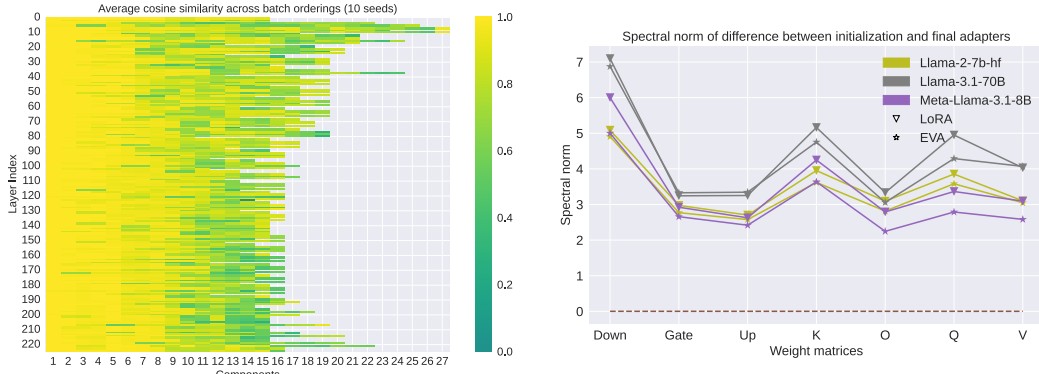

Figure 12: **Left:** Average cosine similarity between SVD components across 10 random seeds for permuting the batch order. The first 10 components remain mostly consistent across all permutations. While the remaining components vary, they strongly correlate with each other. **Right:** Average spectral norm of difference between weight matrices at initialization and after training for LoRA and EVA applied to Llama-2-7B, Llama-3.1-8B, and Llama-3.1-70B. EVA's initialization is closer to the final adapter than LoRA's.

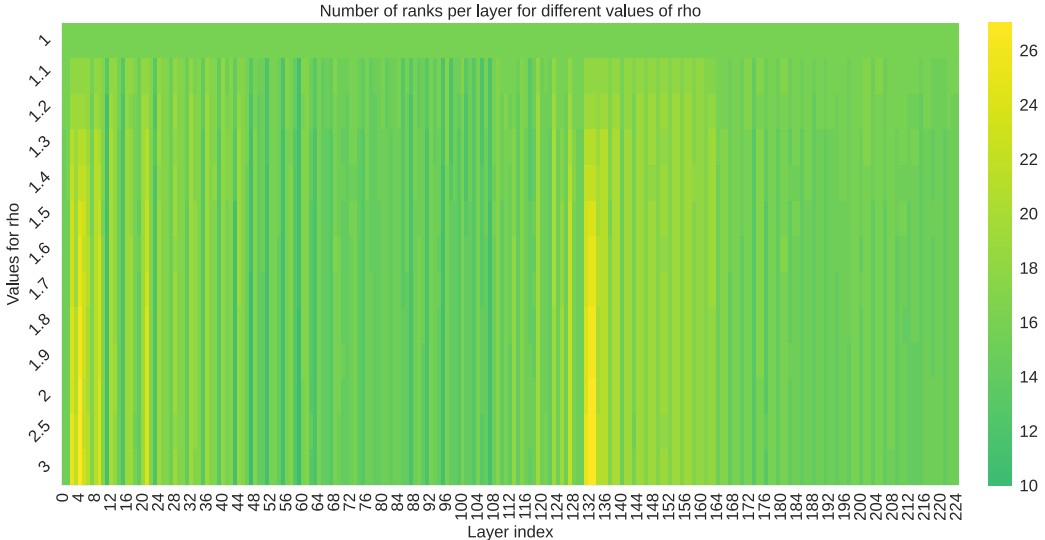

Figure 13: The resulting rank allocation per weight matrix in each layer for Llama-2-7B on the MetaMathQA dataset with different values of $\rho$. The first row represents a uniform distribution where each weight matrix receives the same rank $r = 16$. The most change occurs for $\rho < 1.5$. The redistribution converges for larger values of $\rho$.

matrix for different values of $\rho > 1$ and in Figure 14 we visualize the corresponding deltas. Both visualizations clearly illustrate that the greatest change occurs for values of $\rho < 1.5$. Setting $\rho$ to higher values results in less and less change. Interestingly, some ranks still change when going from $\rho = 2.5$ to $\rho = 3$. Finally, we conduct a hyperparameter search in which we search over different values of $\rho \in \{1, 1.1, 1.2, 1.3, 1.4, 1.5, 1.6, 1.7, 1.8, 1.9, 2, 2.5, 3\}$. We report the results in Figure 15. We find that for Llama-2-7B on MetaMathQA a uniform distribution performs favorably. The second best performance is shared by $\rho = 1.5$ and $\rho = 2$. Therefore, we always search for $\rho = 1$ and $\rho = 2$ for all our remaining experiments when we apply EVA and select the best performing one.

## H  Supplementary Proofs

For convenience we first repeat both Theorem 3.1 and Theorem 3.2 and provide a proof afterwards.

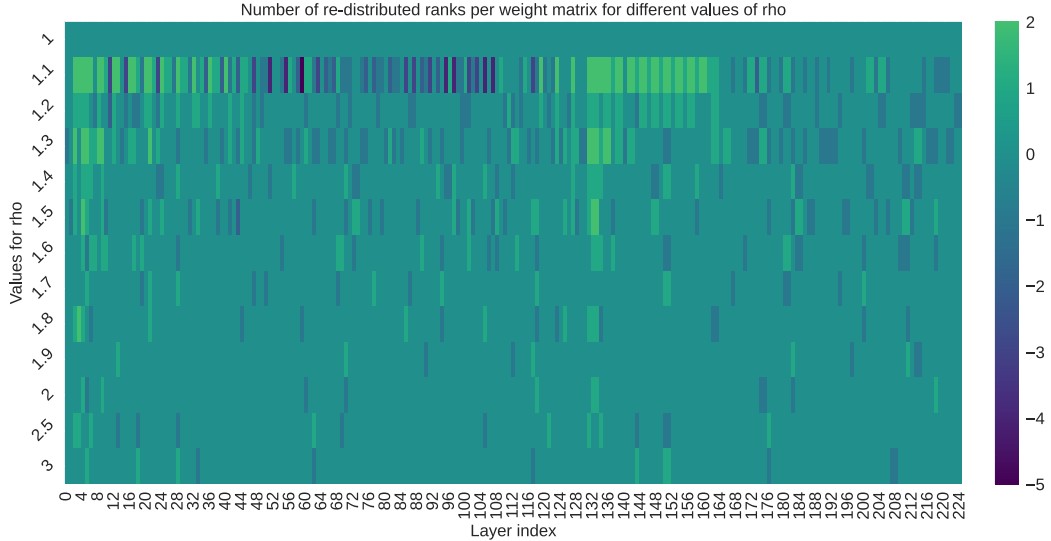

Figure 14: Deltas between rank distributions per weight matrix in each layer for Llama-2-7B on the MetaMathQA dataset with different values of $\rho$. The first row represents a uniform distribution where each weight matrix receives the same rank $r = 16$. The most change occurs in the range $\rho \in [1, 1.5]$. Larger values of $\rho$ do not induce additional significant changes to the rank distribution.

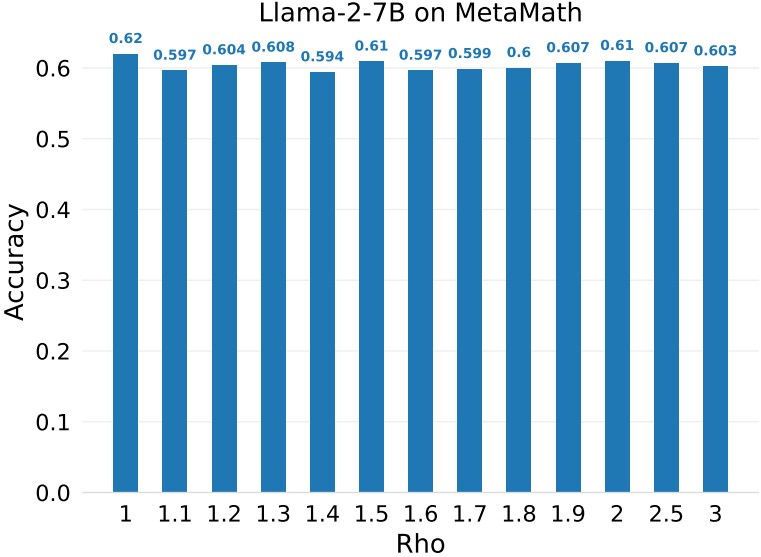

Figure 15: Accuracy for different values of $\rho$ when fine-tuning Llama-2-7B on the MetaMathQA dataset.

## Optimality of SVD-Based Initialization for Maximizing Explained Variance

Let $\boldsymbol{X} \in \mathbb{R}^{b \times d}$ be a matrix of activation vectors obtained from a pretrained model, where $b$ is the number of samples and $d$ is the feature dimension. Suppose we wish to adapt a weight matrix $\boldsymbol{W} \in \mathbb{R}^{k \times d}$ using a low-rank update of the form $\Delta \boldsymbol{W} = BA$, where $\boldsymbol{B} \in \mathbb{R}^{k \times r}$, $\boldsymbol{A} \in \mathbb{R}^{r \times d}$, and $r \ll \min(k, d)$. Let $\boldsymbol{X} = \boldsymbol{U}\boldsymbol{\Sigma}\boldsymbol{V}^\top$ be the singular value decomposition (SVD) of the activation matrix with $\sigma_1 \geq \sigma_2 \geq \cdots \geq 0$ being the singular values of $\boldsymbol{\Sigma}$. Then the top $r$ right singular vectors $\boldsymbol{V}_{:r} \in \mathbb{R}^{d \times r}$ solve the following optimization problem:

$$\boldsymbol{V}_{:r} = \arg \max_{\boldsymbol{V} \in \mathbb{R}^{d \times r}, \boldsymbol{V}^\top \boldsymbol{V} = I} \mathrm{Tr}(\boldsymbol{V}^\top \boldsymbol{X}^\top \boldsymbol{X} \boldsymbol{V}),$$

and also minimize the Frobenius norm reconstruction error:

$$\boldsymbol{V}_{:r} = \arg \min_{\boldsymbol{M} \in \mathbb{R}^{b \times d}, \mathrm{rank}(\boldsymbol{M}) \leq r} \|\boldsymbol{X} - \boldsymbol{M}\|_F^2.$$

Hence, $\boldsymbol{V}_{:r}$ forms the optimal basis for capturing the maximum variance of activations under a rank-$r$ projection.

### Proof.

First, we define the empirical covariance matrix of activations as

$$\boldsymbol{S} = \frac{1}{n-1} \boldsymbol{X}^\top \boldsymbol{X}, \tag{9}$$

with $\boldsymbol{V}$ being the eigenvectors of $\boldsymbol{S}$. Projecting $\boldsymbol{X}$ onto a subspace spanned by the orthonormal basis $\boldsymbol{V} \in \mathbb{R}^{d \times r}$ the total captured variance is

$$\mathrm{Var}_{\boldsymbol{V}(\boldsymbol{X})} = \mathrm{Tr}(\boldsymbol{V}^\top \boldsymbol{X}^\top \boldsymbol{X} \boldsymbol{V}) = \mathrm{Tr}(\boldsymbol{V}^\top \boldsymbol{S} \boldsymbol{V}). \tag{10}$$

The trace is maximized when $\boldsymbol{V}$ contains the eigenvectors corresponding to the top $r$ eigenvalues of $\boldsymbol{S}$, which are the top right singular vectors of $\boldsymbol{X}$. This is verified by the Eckart–Young–Mirsky theorem (Eckart & Young, 1936). The best rank-$r$ approximation to $\boldsymbol{X}$ in the Frobenius norm is given by

$$\boldsymbol{X}_{:r} = \sum_{i=1}^{r} \sigma_i \boldsymbol{u}_i \boldsymbol{v}_i^\top, \tag{11}$$

with $\sigma_1 \geq \sigma_2 \geq \cdots \geq 0$, which uses the top $r$ right singular vectors. Hence the Eckart-Young theorem directly proves both reconstruction with respect to Frobenius norm, as well as maximizing the trace operator.

### Gradient Signal Amplification via EVA Initialization.

Let $\Delta \boldsymbol{W} = \boldsymbol{B}\boldsymbol{A}$ be a low-rank adaptation to a pretrained weight matrix $\boldsymbol{W} \in \mathbb{R}^{k \times d}$, where $\boldsymbol{B} \in \mathbb{R}^{k \times r}$, $\boldsymbol{A} \in \mathbb{R}^{r \times d}$, and $r \ll \min(k, d)$. Let $\boldsymbol{x} \in \mathbb{R}^d$ be the activation input to this layer. Assume activations $\boldsymbol{x}$ are drawn from a distribution with covariance matrix $\boldsymbol{\Sigma} = \mathbb{E}[\boldsymbol{x}\boldsymbol{x}^\top]$. Then initializing $\boldsymbol{A}$ with the top right singular vectors of a sample activation matrix $\boldsymbol{X} \in \mathbb{R}^{b \times d}$ maximizes the expected squared gradient norm:

$$\mathbb{E}\left[\left\|\frac{\partial \mathcal{L}}{\partial \boldsymbol{B}}\right\|_F^2\right] \propto \mathrm{Tr}(\boldsymbol{A}^\top \boldsymbol{\Sigma} \boldsymbol{A}).$$

### Proof.

Let us consider the forward pass

$$\hat{\boldsymbol{y}} = (\boldsymbol{W} + \boldsymbol{B}\boldsymbol{A})\boldsymbol{x} \tag{12}$$

with loss function $\mathcal{L}(\hat{\boldsymbol{y}}, \boldsymbol{y})$ and target $\boldsymbol{y}$. The gradient with respect to $\boldsymbol{B}$ is

$$\frac{\partial \mathcal{L}}{\partial \boldsymbol{B}} = \frac{\partial \mathcal{L}}{\partial \boldsymbol{y}} \boldsymbol{x}^\top \boldsymbol{A}^\top = \boldsymbol{g}\boldsymbol{x}^\top \boldsymbol{A}^\top. \tag{13}$$

The squared Frobenius norm of the gradient is

$$\left\|\frac{\partial \mathcal{L}}{\partial \boldsymbol{B}}\right\|_F^2 = \mathrm{Tr}(\boldsymbol{A}\boldsymbol{x}(\boldsymbol{g}^\top \boldsymbol{g})\boldsymbol{x}^\top \boldsymbol{A}^\top) = (\boldsymbol{g}^\top \boldsymbol{g})\mathrm{Tr}(\boldsymbol{A}\boldsymbol{x}\boldsymbol{x}^\top \boldsymbol{A}^\top). \tag{14}$$

Since $\boldsymbol{BA}$ is initialized to be $\boldsymbol{BA} = 0$ at the beginning of fine-tuning and the gradient $\boldsymbol{g}$ is entirely governed by the behavior of the pretrained model, we can make the assumption that the gradient is statistically independent of the input ($\boldsymbol{x} \perp \boldsymbol{g}$) and

$$\text{Cov}(\boldsymbol{x}, \boldsymbol{g}) = \mathbb{E}[\boldsymbol{x}\boldsymbol{g}^\top] - \mathbb{E}[\boldsymbol{x}] \cdot \mathbb{E}[\boldsymbol{g}^\top] = 0. \tag{15}$$

Hence, by taking the expectation over $\boldsymbol{x} \sim \mathcal{D}$ with $\mathbb{E}[\boldsymbol{x}\boldsymbol{x}^\top] = \boldsymbol{\Sigma}$ we obtain

$$\mathbb{E}\left[\left\|\frac{\partial \mathcal{L}}{\partial \boldsymbol{B}}\right\|_F^2\right] \propto \text{Tr}(\boldsymbol{A}\boldsymbol{\Sigma}\boldsymbol{A}^\top). \tag{16}$$

Again, the trace is maximized when the rows of $\boldsymbol{A}$ are aligned with the top eigenvectors of $\boldsymbol{\Sigma}$, that is, the principal directions of the activations, as proven by the Eckart-Young theorem Eckart & Young (1936).

---

**Algorithm 2** Incremental SVD algorithm from Ross et al. (2008)

---

**Input:** Sequence of data batches $\{\boldsymbol{A}^0, \ldots, \boldsymbol{A}^T\}$, truncated SVD $\text{SVD}(\cdot)$, orthogonalization function $\text{orth}(\cdot)$, running variance update function $\text{young\_cramer\_update}(\cdot, \cdot)$

1: $\bar{\boldsymbol{m}}^0 \leftarrow \frac{1}{b}\sum_{i=0}^b \boldsymbol{A}_{:,i}$, $\boldsymbol{\sigma}^0 \leftarrow \frac{\sum_{i=0}^b(\boldsymbol{A}_{:,i} - \bar{\boldsymbol{m}}^0)^2}{b-1}$   ▷ initialize incremental mean/variance
2: $\boldsymbol{U_0}\boldsymbol{\Sigma_0}\boldsymbol{V}^\top \leftarrow \text{SVD}(\boldsymbol{A}^0 - \bar{\boldsymbol{a}}^0)$   ▷ Perform initial SVD on $\boldsymbol{A}$ to get initial components
3: **for** $i$ in $1, \ldots, T$ **do**
4: $\quad \bar{\boldsymbol{a}}^i \leftarrow \frac{1}{b}\sum_b \boldsymbol{A}_{:,i}^i$, $\bar{\boldsymbol{m}}^i \leftarrow \bar{\boldsymbol{m}}^i + \frac{\boldsymbol{a}^i - \bar{\boldsymbol{m}}^{i-1}}{b(i+1)}$   ▷ compute mean vectors
5: $\quad \boldsymbol{\sigma}^i \leftarrow \text{young\_cramer\_update}(\boldsymbol{\sigma}^{i-1}, \boldsymbol{A}^i)$   ▷ Update running variance
6: $\quad \hat{\boldsymbol{A}}^i \leftarrow \left[\boldsymbol{A}^i - \bar{\boldsymbol{a}}^i; \sqrt{\frac{b(i+1)}{2b}}\left(\bar{\boldsymbol{m}}^i - \bar{\boldsymbol{a}}^i\right)\right]$   ▷ concatenate mean correction factor
7: $\quad \tilde{\boldsymbol{A}}^i \leftarrow \text{orth}(\hat{\boldsymbol{A}}^i - \boldsymbol{U}_{i-1}\boldsymbol{U}_{i-1}^\top \hat{\boldsymbol{A}}^i)$   ▷ Obtain orthogonal component to $\boldsymbol{U}$
8: $\quad \boldsymbol{R} = \begin{bmatrix} \boldsymbol{\Sigma_{i-1}} & \boldsymbol{U}_{i-1}\top\hat{\boldsymbol{A}}^i \\ \boldsymbol{0} & \tilde{\boldsymbol{A}}^i\hat{\boldsymbol{A}}^i \end{bmatrix}$   ▷ Define matrix $\boldsymbol{R}$
9: $\quad \tilde{\boldsymbol{U}}\tilde{\boldsymbol{\Sigma}}\tilde{\boldsymbol{V}}^\top \leftarrow \text{SVD}(\boldsymbol{R})$   ▷ Perform SVD on $\boldsymbol{R}$
10: $\quad \boldsymbol{U}_i \leftarrow \left[\boldsymbol{U}_{i-1}; \tilde{\boldsymbol{A}}^i\right]\tilde{\boldsymbol{U}}$, $\boldsymbol{\Sigma}_i \leftarrow \tilde{\boldsymbol{\Sigma}}$   ▷ Update SVD components
11: **end for**

---

# I Ablation Studies

Finally, we conduct ablation studies on EVA to investigate important factors that contribute to its performance. Specifically, we investigate the impact of scale and direction. To this end, we use the VTAB-1K dataset because it comprises a diverse set of tasks and allows for a systematic investigation on in-domain (natural) and out-of-distribution (specialized and structured) data. We report results for our ablation studies in Table 25 and explain the different settings in the following paragraphs.

**Effect of scale.** To investigate the effect of scale on initialization, we add a setting that uses whitening (EVA-whiten). Whitening scales the initialization by the reciprocal of their eigenvalues, which alters scale, but preserves directions. We found that whitening can significantly improve performance in structured (out-of-distribution) tasks, even leading to a slightly higher average score than EVA. This indicates that scale is especially important for structured data. However, EVA-whiten experiences a slight performance drop in natural and specialized tasks.

**Effect of directions.** To address the importance of the directions of the components, we randomly permute its rows (EVA-perm). This preserves scale while corrupting directions and the $\ell_2$ norm of $\boldsymbol{A}$. Additionally, we add a setting where we randomly rotate $\boldsymbol{A}$ (EVA-rot), which preserves the $\ell_2$ norm but alters directions. We find that altering directions leads to a drop in performance on structured tasks, while changing the $\ell_2$ norm leads to a drop on natural tasks. Both EVA-perm and EVA-rot lead to worse average performance across all tasks compared to EVA.

**Effect of rank redistribution.** We conduct an experiment in which we randomly initialize $\boldsymbol{A}$ after performing rank redistribution (LoRA redist). This setting gives insights on the effect of the

redistribution and whether its benefits are bound to EVA. Redistribution has a positive effect on LoRA on natural tasks, but a negative effect on both structured and specialized tasks. This illustrates that rank redistribution is most beneficial in combination with EVA's initialization of $A$.

Generally, we can say that EVA performs particularly well on natural images and whitening can enhance its performance on out-of-distribution images. The decisive factor with respect to this improvement seems to be a controlled change in the scale of initialization induced by the singular values. Therefore, by changing the scale in a controlled manner, we can make EVA more compatible for different kinds of data. The results for EVA-perm confirm that the scale is the decisive factor for initialization.

Table 25: Group-wise averages for DINOv2-g/14 ablation studies on the VTAB-1K benchmark.

| Method | Nat. | Spec. | Struct. | All |
|---|---|---|---|---|
| LoRA | 83.2 | **88.8** | 69.0 | 78.4 |
| LoRA-redist | 87.3 | 88.0 | 68.2 | 79.4 |
| EVA-whiten | 87.5 | 87.5 | **69.1** | **79.8** |
| EVA-rot | **87.7** | 88.0 | 68.2 | 79.6 |
| EVA-perm | 87.4 | 87.8 | 68.3 | 79.5 |
| EVA | **87.7** | 87.9 | 68.6 | 79.7 |

## J  Further Discussions

**Alternative data-driven initialization schemes.** We investigated alternative data-driven initialization schemes such as Kernel-PCA (Schölkopf et al., 1997) or Linear Discriminant Analysis (Fisher, 1936, LDA). Kernel-PCA can account for non-linearities in the data but scales with the number of datapoints, which is impractical. For LDA, we observed convergence instabilities during incremental updates. In our setting we deal with sequences, therefore the number of datapoints grows fast, making Kernel-PCA impractical. LDA projects the data onto a subspace that maximizes linear separability between classes. Such an initialization scheme may be particularly interesting for classification tasks like GLUE or VTAB-1K.

**Additional latency of SVD.** EVA leads to performance improvements over LoRA, but introduces additional latency at the beginning of training to compute the initialization. In Figure 5 (left) we demonstrate that this process constitutes merely 0.2% of the actual training time for Llama-2-7B on MetaMathQA. In addition, in Appendix F we show that this process is largely invariant to the batch size and order, meaning smaller batch sizes may be used, resulting in additional speedup. Since the SVD computation does not require backpropagation and storing of optimizer states, there is no memory overhead.

**How to initialize $B$?** We follow Hu et al. (2022) and initialize $B = 0$. All other initialization methods initialize $B \neq 0$, which requires altering the pre-trained model weights. In our experiments, EVA usually outperformed CorDA and LoRA-GA, even though they are both data-driven and leverage similar information. Therefore, setting $B = 0$ could also be a driving factor for improved performance. We leave this investigation to future work. Finally, restoring the base model after fine-tuning requires computing the delta of the weights before and after training for $B \neq 0$. In contrast, EVA and LoRA can fully restore the base model's weights by simply unloading the adapter weights.

