# OpenReview forum: "Parameter Efficient Fine-tuning via Explained Variance Adaptation"
_NeurIPS.cc/2025/Conference — NeurIPS 2025 poster_

### Official Review · Reviewer_sfAD · 2025-06-25

**Clarity:** 3
**Significance:** 3
**Originality:** 3
**Rating:** 4
**Confidence:** 3

**Summary:**

The paper proposes Explained Variance Adaptation, a new LORA weight initialization method that provably maximizes the expected gradient signal. Given a rank budget, EVA captures the maximum possible variance in the input mini-batches for every layer of the model. The paper claims that maximizing expected variance directly improves the performance of different tasks upon adaptation. Experiments spanning several tasks and domains show faster convergence and higher average scores than several LoRA variants, with a reported initialisation overhead being a fraction of total fine‑tuning time.

**Questions:**

1. How does the expected gradient magnitude theoretically contribute to the generalization error?
2. What does the paper mean by "stability"?
3. Why is the approach more significant than adding adaptive ranks to earlier data-based initialization to CorDA or LoRA-GA?
I will increase my score to an accept if question 1 is answered with sufficient backing (theory or references to prior work) and if the novelty concerns can be sufficiently addressed.


EDIT
------------------------------

The authors have clarified most of these concerns in their rebuttal.

**Ethical Concerns:**

["NO or VERY MINOR ethics concerns only"]

**Final Justification:**

I thank the authors for their response.

The link with NTK, and through it, EVA's impact of generalization is definitely interesting. I strongly recommend that the authors add this component to their paper. This also addresses my main concern regarding the novelty of the paper.

I therefore increase my score.

**Limitations:**

Yes

**Paper Formatting Concerns:**

I would suggest the tabes to explicitly highlight rows relating to EVA and its variants, as it would make comparing the method with baselinesz easier for the reader.

**Quality:**

2

**Strengths And Weaknesses:**

Strengths:

1. The primary strength of the paper is in its evaluation and analysis. I appreciate the extensive evaluation of the method on several different settings and domains, as well as the extensive experiments and ablations in the supplementary material.

2. The paper has been written well, and all steps of the method are clear enough for re-implementation. The method itself is intuitive and easy to understand.

Weaknesses:

1. One gap that I found in the theoretical results is how higher expected gradient magnitude translates to better generalization. The paper has the underlying assumption that a higher magnitude is a surrogate for better generalization capabilities/lower generalization error on the test set, but why this would be the case is not well established. I would like to see some citations or additional theory on how the expected gradient magnitude of LORA updates influences model performance.

2. Some of the terms used in the paper, like "stronger, more stable gradient signal," have not been well-defined in an intuitive or mathematical sense. For instance, does "stability" refer to a lesser likelihood of encountering NaN values during training or lower variance across batches?

3. Overall, I find that the paper has limited novelty. Data-based decompositions for LoRA initialization have been explored in prior work (CorDA by Yang et al. and LoRA-GA by Wang et al.), while adaptive ranks have been well established in AdaLoRA and its successors. EVA mainly bundles these ideas together.

4. The proof of Theorem 4 assumes orthogonality of the gradient and input. However, this is only guaranteed for the first gradient update step. I also do not follow why equation 11 is relevant here (I might be missing some intermediate arguments). Since equation 12 has proportionality instead of equality, doesn't it directly follow from equation 10 after the first update step when $g^T g$ is likely non-zero?


EDIT
------------------------------

The authors have clarified most of these concerns in their rebuttal.

---

> ### Author Rebuttal · Authors · 2025-07-31
>
> Dear reviewer sfAD,
>
> Thank you for the constructive feedback, it helps us to substantially improve our manuscript. We address the raised concerns as follows.
>
> **Novelty**
>
> We acknowledge that adaptive rank allocation and data-driven initialization have been explored in concurrent works (e.g., CorDA, LoRA-GA). However, we strongly believe that novelty should not be judged solely by whether single components used in a framework already existed, but by whether a method offers a new perspective or principled formulation that provides novel insights. That said, EVA offers a theoretically grounded approach: both initialization and rank allocation arise from maximizing expected gradient signal under a parameter constraint. This goes beyond heuristic initialization design (CorDA, LoRA-GA) or soft constraints during optimization (AdaLoRA). By proposing EVA we provide insights into the design of maximizing gradient signal without relying on hyperparameters for computing the initialization as in CorDA/LoRA-GA which allows to systematically control for rank redistribution while providing strong performance with a reduced parameter count. We believe all of these constitute a meaningful and novel contribution to the field.
>
> **Theoretical Contribution**
>
> Thank you for pointing out the lack of connection between gradient signal magnitude and generalization error. There is indeed a theoretically grounded connection between initialization via EVA and the generalization error via the Neural Tangent Kernel (NTK, [1]). To briefly recap, EVA selects a rank-r subspace $V= \operatorname{span}({v_1, \ldots,v_r})$ in which to place LoRA updates such that the expected gradient magnitude is **maximized**:
> $$\mathbb{E}\_{x} \left[ \left\| \nabla_\theta f(x)^\top v_i \right\|^2 \right]$$
> This expectation is maximized if we choose the **top eigenvectors of the activation covariance** for initializing the LoRA matrices, as done in EVA. That is, EVA aligns LoRA directions with the principal components of
> $$\Sigma = \mathbb{E}\_{x} \left[  \nabla_\theta f(x) \nabla_\theta f(x)^\top \right]$$.
>
> Intriguingly, **this is equivalent to the empirical NTK matrix.** In NTK theory, learning proceeds along the NTK eigenbasis. Let $K \in \mathbb{R}^{n\times n}$ be the empirical NTK matrix over training points $x\_1, \ldots, x\_n$. The NTK view tells us that at infinite width the network output evolves under gradient descent as:
> $$f\_t(x) = f\_0(x) + K(x, X) K^{-1} (y - f\_0(X)) \cdot \left(1 - e^{-\eta \Lambda t} \right)$$,
> and that learning decomposes across the eigenfunction of the NTK. Crucially, **directions with higher NTK eigenvalues $λ\_i$​ are learned faster and more robustly [2,3]**.
>
> Finally, EVA’s initialization ensures that the LoRA subspace is aligned with the **top-r eigenvectors of the NTK**.
> Let $u\_i=1n\{u_i\}\_{i=1}^n$ be the eigenvectors of K, and $y \in R^n$ be the labels. Then the squared generalization error is bounded by the residual in low-eigenvalue components:
> $$\mathcal{E}\_{\text{gen}} \lesssim \sum\_{i=r+1}^{n} \frac{(u_i^\top y)^2}{\lambda_i^2}$$.
> Or put differently, the projection of the residual error onto **non-covered NTK directions (those outside $\mathcal{V}$)** determines the generalization error. As EVA initializes the LoRA matrices with the top-r eigenvectors of the NTK, the generalization error is minimized as it is covered only by directions with strictly smaller eigenvalues. Therefore, maximizing expected gradient signal through data-driven initialization aligns low-rank adaptation directions with the top eigenspaces of the NTK. Prior work [2,3] shows that learning concentrated in these eigenspaces leads to faster convergence and lower generalization error.
>
> Thanks to the reviewer’s comment we were able to establish this intriguing connection and we hope this satisfies their concerns as these additional theoretical insights add novelty to our proposed approach.
>
> [1] Jacot et al., Neural Tangent Kernel: Convergence and Generalization in Neural Networks, NeurIPS 2018
> [2] Arora et al., On Exact Computation with an Infinitely Wide Neural Net, NeurIPS 2019
> [3] Lee et al., Wide Neural Networks of Any Depth Evolve as Linear Models Under Gradient Descent, NeurIPS 2019
>
> **Questions:**
>
> **What does the paper mean by "stability"?**
>
> Thank you for noticing the lack of definition for the term “stability”. We indeed observed that EVA can lead to smoother convergence curves compared to LoRA-GA or CorDA (Figure 3, right). However this is an empirical observation without any theoretical backing. From a theoretical viewpoint the variance in the gradients will be larger than the ones encountered in standard LoRA, as EVA concentrates gradient variance in the most informative directions. This definitely affects training dynamics and sometimes we observed that especially for higher learning rates it can be that LoRA still trains, but EVA diverges. We add a discussion on this in Section 5. We admit that the usage of this term is confusing and therefore remove it from our claims.
>
>
> **Since equation 12 has proportionality instead of equality, doesn't it directly follow from equation 10 after the first update step when gTg is likely non-zero?**
>
> Yes, you are correct, Eq. 12 directly follows from Eq. 10 under the statistical independence assumption. The relevance of Eq. 11 is indeed not directly given in the current form. We agree that the phrasing around this equation is a bit unfortunate. We simply meant to augment the meaning of statistical independence in a formal manner. We rephrase this to avoid any confusion at this point.
>
> If any questions remain or come up during the rebuttal process, we would be happy to engage in further discussion.

---

> > ### Comment · Reviewer_sfAD · 2025-08-05
> >
> > I thank the authors for their response.
> >
> > The link with NTK, and through it, EVA's impact of generalization is definitely interesting. I strongly recommend that the authors add this component to their paper. This also addresses my main concern regarding the novelty of the paper.
> >
> > I therefore increase my score.

---

### Official Review · Reviewer_LBrt · 2025-07-01

**Clarity:** 3
**Significance:** 3
**Originality:** 3
**Rating:** 4
**Confidence:** 3

**Summary:**

This paper introduces Explained Variance Adaptation (EVA), a novel initialization scheme for LoRA in parameter-efficient fine-tuning of foundation models. Unlike prior SVD-based initializations, EVA performs incremental SVD on activation vectors to obtain directions that maximize explained variance. These directions are used to initialize LoRA matrices, aiming to maximize the expected gradient signal at the onset of fine-tuning. The paper also proposes an adaptive rank allocation mechanism based on explained variance ratios. EVA is evaluated across various domains—language modeling, language understanding, image classification, and reinforcement learning AND improved performance compared to LoRA variants and other PEFT baselines.

**Questions:**

see weakness.

**Ethical Concerns:**

["NO or VERY MINOR ethics concerns only"]

**Final Justification:**

I thank the authors for their detailed rebuttal, which addresses my concerns clearly. I maintain my borderline accept recommendation.

**Limitations:**

yes

**Quality:**

3

**Strengths And Weaknesses:**

Strength:

The idea of using activation-driven SVD for LoRA initialization is both elegant and theoretically grounded.

EVA is rigorously tested across 51 tasks in four domains, showing strong average performance.

Despite being data-driven, EVA exhibits minimal overhead (<1% training time) via batch-wise incremental SVD.

Weakness:

The method assumes access to a static downstream dataset prior to fine-tuning, which may not generalize to online or continual learning settings.

While adaptive rank allocation is justified theoretically, its impact varies per task (e.g., limited gains in decision-making tasks).

EVA focuses on relatively low-rank setups (e.g., r=16); scaling behavior to higher ranks is less explored due to computational constraints, seems to limit its practicality.

---

> ### Author Rebuttal · Authors · 2025-07-30
>
> Dear reviewer LBrt,
>
> Thank you for your valuable feedback and positive assessment of our work.
>
> **Static Dataset**
>
> We agree with the reviewer that this is a current limitation of our method, as we explicitly state in our discussion section. EVA is designed to maximize performance in the very common and practical scenario where fine-tuning is performed on a known, static downstream dataset. This is the basic setting that is considered in the literature. We believe that extending EVA to non-static datasets is a fruitful direction for future work.
>
> **Impact of Rank Redistribution**
>
> We thank the reviewer for this observation, which aligns with our own findings. As noted in our results, the performance improvement from adaptive rank allocation can indeed be task-dependent, with some domains like decision-making showing limited gains. However, a key benefit of our redistribution strategy is that it consistently reduces the number of trainable parameters across all our experiments. This is because ranks are often shifted from higher-dimensional weights to more parameter-efficient lower-dimensional ones. Therefore, even when performance is on-par, EVA offers a clear advantage in parameter-efficiency, which is particularly relevant when fine-tuning large models. For this reason, we recommend using rank redistribution by default.
>
> **Choice of Rank**
>
> We thank the reviewer for raising this point. While we acknowledge the computational overhead for very high ranks, we did explore scaling behavior beyond rank 16. Specifically, in Table 10 (Appendix A), we present results for fine-tuning Llama-2-7B with ranks up to 64. The results show that EVA and EVA+DORA are consistently the best-performing methods across all tested ranks, from 8 to 64. Furthermore, our experiments suggest that simply increasing the rank does not necessarily guarantee better performance. In fact, we empirically observed that higher ranks do not always perform better. We hypothesize this may be due to overfitting when introducing too many parameters relative to the dataset size. Consequently, EVA’s strong performance, particularly in lower-rank setups, is a key advantage for practical, compute-constrained fine-tuning scenarios, which are very prevalent in practice.
>
> If any questions remain or come up during the rebuttal process, we would be happy to engage in further discussion.

---

> > ### Comment · Reviewer_LBrt · 2025-08-05
> >
> > I thank the authors for their detailed rebuttal, which addresses my concerns clearly. I maintain my borderline accept recommendation.

---

### Official Review · Reviewer_G3kb · 2025-07-03

**Clarity:** 3
**Significance:** 2
**Originality:** 2
**Rating:** 4
**Confidence:** 4

**Summary:**

This work proposes a novel initialization and allocation scheme for PEFT, which leverages information obtained from activations to initialize low-rank adapters into the most impactful directions (with highest variance). This leads to faster convergence and higher performance in a broad range of settings, compared to competitors

**Questions:**

The main concerns are reported in Weaknesses.

Other question:
1. In section 3.3., line 158, referring to A,B matrices, the authors mention “to preserve the base model weights, one of them must be initialized with zeros”. While zero-initializing one of the two matrices is the most common practice, it should be mentioned that other methods (such as [4,3]) utilize a residual approach (subtracting frozen version of the adapters).
- In particular, did the authors test this type of initialization, which in theory would allow to also initialize the B matrix with SVD information?
2. How does performance vary at different minibatch sizes? What is the minimum amount of data necessary to obtain significant performance improvements?

I am willing to increase my score if the main concerns are addressed.

[4] Meng et al. - PiSSA: Principal Singular Values and Singular Vectors Adaptation of Large Language Models, NeurIPS24

**Ethical Concerns:**

["NO or VERY MINOR ethics concerns only"]

**Final Justification:**

I carefully read authors response to reviews, and they address most of the concerns

**Limitations:**

yes

**Quality:**

2

**Strengths And Weaknesses:**

Strengths:
- The presentation is clear, and the method is well motivated
- Extensive number of analysis and ablations


Weaknesses:
1. EVA leads to performance improvements, though it incurs an initial cost for SVD initialization. However, it is not clear how the performance of “SVD_init + EVA” compares to “random_init + LoRA” (or other methods) at the same time t, where t = 0 is defined as the time before the two initializations.
2. While GLUE is extensively used for testing PEFT performance, reporting results on the validation set (as it is commonly done) is not best practice, as it is prone to overfitting. A more rigorous evaluation approach involving a separate validation/test split has been proposed in [1,2,3].


[1] Wu M. et al., Advancing parameter efficiency in finetuning via representation editing, ACL 2024

[2] Wu Z. et al., Reft: Representation finetuning for language models, NeurIPS 2024

[3] Bini et al., DeLoRA: Decoupling Angles and Strength in Low-rank Adaptation, ICLR 2025

---

> ### Author Rebuttal · Authors · 2025-07-31
>
> Dear reviewer G3kb,
>
> Thank you for the constructive feedback, it helps us to substantially improve our manuscript. We address the raised concerns as follows.
>
> **Computational  Fairness**
>
> We would like to clarify that the initial cost of SVD is negligible compared to the fine-tuning cost, as shown in Table 3. We rehash the aspects of the computational overhead below.
>
> - Initialization Overhead is Minimal: The wall-clock time for EVA's SVD initialization is a very small fraction of the total training time. As shown in Table 3, for a 7B parameter model, this overhead is only 0.7% of the total training time, and we demonstrate it can be reduced to as little as 0.2% by using a smaller batch size for the SVD step. This change does not impact the quality of the initialization (Figure 10). We are confident that this minor time difference does not change the final performance of the baselines.
> - Faster Convergence and Evaluation Protocol: The small upfront time investment in our principled initialization is immediately compensated by significantly faster convergence. Figures 3 and 5 clearly show that EVA achieves a lower training loss much more rapidly than LoRA and other baselines. This means that at virtually any time t after the initialization phase, EVA has already reached a more optimal state. We clarified this point in the updated version of our manuscript.
>
>
> **GLUE Evaluation**
>
> We thank the reviewer for this valuable suggestion regarding evaluation best practices. We agree that rigorous evaluation is paramount to estimate the generalization error. However, re-running all GLUE tasks with a revised data split was infeasible within the rebuttal period.
>
> We would like to mention that for GLUE specifically, we applied the same comprehensive hyperparameter search protocol to all methods, including our own and all baselines. As detailed in Appendix C.3 for each method on each GLUE task, we tuned key hyperparameters (e.g., learning rates, LoRA ranks) and selected the best-performing configuration for that specific method. The final results reported in Table 2 therefore compare the optimal performance of EVA against the optimal performance of each baseline.
>
> **Questions**
>
> **Initialization of B matrix**
>
> Thank you for pointing this out. It is indeed possible to initialize B as well in our setup, following works such as PiSSA or LoRA-GA. We chose the setup B=0 as there exists literature that this setting has favorable theoretical properties, such as stable convergence for larger learning rates [1]. Furthermore, there is no common consensus in the literature that one should be preferred over the other or that one has favorable properties over the other. Due to the amount of extensive experiments present in our manuscript, we aim to explore this effect in future work.
>
> **Minibatch Size**
>
> We found our initialization to be robust across different minibatch sizes. Specifically, our analysis shows that minibatch sizes between 4 and 32 for the incremental SVD computation produce highly correlated components (Figure 10, right), leading to comparable downstream performance. We did not increase the minibatch size further to ensure that the memory bottleneck is determined by the main fine-tuning stage (i.e., by the LoRA rank and training batch size), not by the EVA initialization step. This setting makes EVA's overhead negligible, adding as little as 0.7% to the total training time for our default language generation setting for Llama-2-7B on common sense reasoning tasks (Table 3).
>
> **Minimum Data for Fine-tuning**
>
> The minimum data required by EVA is determined dynamically by the convergence of principal components during incremental SVD. This convergence is controlled by a cosine similarity threshold, $\tau$. A high value for $\tau$ represents a stricter threshold for considering components converged, whereas a low value is less permissive. By default, we use $\tau=0.99$. Even for this strict setting, the initialization for Llama-2-7B is highly efficient, completing in just 80 to 300 seconds (depending on the minibatch size) after processing fewer than 100 minibatches. However, we agree that an additional analysis on the performance impact of lowering $\tau$ would be a valuable addition to the paper. We commit to including this analysis in future versions of our manuscript.
>
> Moreover, we agree that it would be interesting to study the impact of EVA when fine-tuning with different data budgets. While we are unable to provide this analysis during the time-frame of the rebuttal, we aim to investigate this in more detail in future work. We expect that the advantage of EVA can be particularly pronounced in low-data regimes, where the effect of initialization may have an even larger role for final performance. This hypothesis is supported by the loss curves presented in Figure 3, which show that EVA achieves substantially lower loss during the early stages of fine-tuning (e.g., at 1.5K updates). Consequently, EVA may prove to be particularly effective in such low-data scenarios.
>
> Our manuscript has improved substantially by addressing and incorporating your feedback. If any questions remain, we are happy to engage in further discussion. If you find our responses and additional results useful, we would appreciate it if you would consider changing your score.
>
> [1] Hayou et al., LoRA+: Efficient Low Rank Adaptation of Large Models, ICML 2024

---

> > ### Comment · Reviewer_G3kb · 2025-08-07
> >
> > I carefully read all reviews and I am satisfied with the authors' response. Therefore I am raising my score.

---

### Official Review · Reviewer_6shx · 2025-07-07

**Clarity:** 4
**Significance:** 2
**Originality:** 2
**Rating:** 3
**Confidence:** 4

**Summary:**

The paper proposes a variant of Low Rank Adaptation (LoRA) for finetuning foundation models. The basic idea, termed Expected Variance Adaptation (EVA), is to perform an SVD on the activation vectors that are obtained/estimated from minibatches of the downstream task. This is similar in spirit to methods that use the gradients to estimate the importance. The authors show that this process leads to a "more stable" gradient signal. They also propose a rank allocation procedure that tries to distribute a given rank budget across weight matrices in order to maximize the explained variance. The paper does experiments on finetuning LLMs on math+reasoning taks, as well as results for image classification and decision making tasks. The main goal of the experiments is to report the parameter reduction, while preserving accuracy.

**Questions:**

Please address the questions of novelty and comparison that were raised in the the "weakness" section of the review.

**Ethical Concerns:**

["NO or VERY MINOR ethics concerns only"]

**Final Justification:**

The authors did a thorough job of answering reviewer concerns and comments. They also realized a connection between their proposed gradient based finetuning and the neural tangent kernel. This connection is perhaps interesting, but unfortunately they don't work out the implications of this in a formal enough way, imo. I have raised my evaluation slightly, but I'd like to hear from one of the other reviewers.

**Limitations:**

yes

**Quality:**

2

**Strengths And Weaknesses:**

Strengths:

- The method proposed is natural, and the experiments are fairly thorough in terms of the domains that they cover.
- The writing of the paper is very clear and the ideas are well explained.

Weaknesses:

- Unfortunately the paper is not very novel. The idea is very similar to the gradient based importance metrics. While the paper says that this is "provably" good initialization at several points, this is because of the way the objective is set up-- the algorithm is basically SVD, and is optimal in terms of capturing the variance. (Why is capturing the variance of the activations or aligning with the gradient the right objective?)

- The experimental section is reasonable, but this has become a really crowded field over the last couple of years, so the paper needs to compare with a bunch of newer work on PEFT. (DoRA, MiLoRA). There are also "activation guided" methods that are really similar to this paper, e.g., AG-LoRA (https://ieeexplore.ieee.org/document/10852296), GLoRA (https://openreview.net/forum?id=K7KQkiHanD ) that the authors should compare to.

Overall, while the writing is clean, I am concerned about the novelty with this submission.

---

> ### Author Rebuttal · Authors · 2025-07-31
>
> Dear reviewer 6shx,
>
> We would like to thank you for your valuable feedback.
>
> **Novelty**
>
> We acknowledge that adaptive rank allocation and data-driven initialization have been explored in concurrent works (e.g., CorDA, LoRA-GA). However, we strongly believe that novelty should not be judged solely by whether single components used in a framework already existed, but by whether a method offers a new perspective or principled formulation that provides novel insights. That said, EVA offers a theoretically grounded approach: both initialization and rank allocation arise from maximizing expected gradient signal under a parameter constraint. This goes beyond heuristic initialization design (CorDA, LoRA-GA) or soft constraints during optimization (AdaLoRA). By proposing EVA we provide insights into the design of maximizing gradient signal without relying on hyperparameters for computing the initialization as in CorDA/LoRA-GA which allows to systematically control for rank redistribution while providing strong performance with a reduced parameter count. We believe all of these constitute a meaningful and novel contribution to the field.
>
> We would furthermore like to highlight the practical efficiency gains resulting from our approach which is quantified in Table 3. For fine-tuning a Llama-2-7B model on a single gpu, EVA’s initialization takes only 0.7% of the total training time, whereas the data-driven methods LoRA-GA and CorDA require 2.4% and 4.5%, respectively.
>
>
> **Additional Baselines**
>
> We agree that the PEFT landscape is rapidly evolving. Therefore, we added a wide range of initialization methods as baselines.
>
> DoRA:
>
> As mentioned in line 224, we already compared EVA to DoRA on common sense reasoning (Appendix B, Table 8), mathematical reasoning (Appendix B, Table 9), vision tasks (Appendix E, Table 21), and decision-making tasks (Appendix F, Table 24). Our results show that EVA is consistently competitive. Furthermore, we demonstrate that EVA is complementary to DoRA; the combination (EVA+DORA) frequently achieves the best performance across different tasks (Appendix B, Tables 8,9; Appendix F, 24), highlighting the unique and orthogonal contribution of our initialization scheme.
>
> MiLoRA:
>
> We agree with the reviewer that adding MiLoRA is an important baseline due to its similarity to PiSSA. We therefore created an integration in the PEFT library [1] for it and trained it across common sense reasoning as well as math tasks. We present the results together with DoRA results in tables R1 and R2 below.
>
> Table R1: Results on Common Sense Reasoning (mean and standard deviation across three seeds). We report results for rank 16.
> | **Model** | **Method** | **BoolQ** | **PIQA** | **SIQA** | **Hella Swag** | **Wino- grande** | **ARC-e** | **ARC-c** | **OBQA** | **Avg.** |
> | --- | --- | --- | --- | --- | --- | --- | --- | --- | --- | --- |
> | Llama-2-7B | MiLoRA | 65.0 (±3.9) | 84.8 (±.4) | 82.3 (±.3) | 94.9 (±.1) | 84.5 (±.8) | 88.2 (±.1) | 74.9 (±1.1) | 85.3 (±.2) | 82.5 |
> | Llama-2-7B | DoRA | 68.3 (±2.6) | 85.1 (±.3) | 82.2 (±.5) | 94.9 (±.1) | 84.3 (±.2) | 88.7 (±.2) | 74.8 (±.5) | 86.3 (±.5) | 83.1 |
> | Llama-2-7B | EVA | 68.3 (±8.0) | 85.3 (±.1) | 82.9 (±1.1) | 95.2 (±.1) | 85.2 (±.8) | 88.6 (±.5) | 75.8 (±.4) | 86.3 (±.6) | **83.4** |
> | Llama-3.1-8B | MiLoRA | 85.7 (±.5) | 90.8 (±.2) | 83.0 (±.3) | 96.8 (±.1) | 88.8 (±.6) | 94.4 (±.4) | 84.9 (±.9) | 90.5 (±.5) | 89.4 |
> | Llama-3.1-8B | DoRA | 86.2 (±.2) | 90.8 (±.1) | 83.4 (±.3) | 96.9 (±.0) | 88.6 (±.3) | 94.3 (±.1) | 84.9 (±.7) | 89.4 (±.0) | 89.3 |
> | Llama-3.1-8B | EVA | 85.3 (±.4) | 90.4 (±.0) | 83.4 (±.2) | 97.0 (±.0) | 89.0 (±.2) | 94.4 (±.3) | 86.0 (±.2) | 90.3 (±.8) | **89.5** |
> | Gemma-2-9b | MiLoRA | 88.2 (±.3) | 93.0 (±.2) | 85.0 (±.3) | 97.8 (±.1) | 92.9 (±1.0) | 97.4 (±.2) | 90.2 (±.9) | 93.9 (±.1) | 92.3 |
> | Gemma-2-9b | DoRA | 88.3 (±.2) | 92.6 (±.4) | 84.9 (±.3) | 97.7 (±.1) | 92.2 (±.4) | 97.1 (±.1) | 89.9 (±.0) | 94.5 (±.5) | 92.1 |
> | Gemma-2-9b | EVA | 89.4 (±.3) | 94.6 (±.1) | 85.8 (±.1) | 98.3 (±.1) | 94.4 (±.3) | 98.0 (±.1) | 93.0 (±.0) | 95.9 (±.2) | **93.7** |
>
> Table R2: Results on Math (mean and standard deviation across three seeds). We report results for rank 16.
> | **Model** | **Method** | **GSM8K** | **MATH** |
> | --- | --- | --- | --- |
> | Llama-2-7B | MiLoRA | 59.7 (±1.4) | 11.2 (±.1) |
> | Llama-2-7B | DoRA | 59.8 (±.5) | 11.5 (±.2) |
> | Llama-2-7B | EVA | **61.9 (±.5)** | **13.1 (±.3)** |
> | Llama-3.1-8B | MiLoRA | 78.6 (±.1) | 30.3 (±.3) |
> | Llama-3.1-8B | DoRA | 77.9 (±.1) | 30.2 (±.5) |
> | Llama-3.1-8B | EVA | **78.8 (±.3)** | **31.2 (±.3)** |
> | Gemma-2-9b | MiLoRA | **83.7 (±.4)** | **41.9 (±.3)** |
> | Gemma-2-9b | DoRA | 82.5 (±.6) | 39.7 (±.4) |
> | Gemma-2-9b | EVA | 83.6 (±.8) | 41.5 (±.3) |
>
> AG-LoRA:
>
> Thank you for pointing out this highly relevant work. Unfortunately, the code for AG-LoRA is not publicly available, and a custom re-implementation for activation-weighting and outlier-based global rank assignment may result in subtle discrepancies to the code run by the authors. To ensure a rigorous and reproducible evaluation, we prioritize baselines with available and verified code. We will add a discussion on AG-LORA in our related work section.
>
> GLoRA:
>
> GLoRA is orthogonal to EVA as it does not focus on initialization. It employs evolutionary strategies to explore adapter configurations, whereas the focus of our work lies in initialization of adapters and their ranks.
>
>
> In summary, we have clarified the novelty of EVA by highlighting its theoretical grounding and efficiency gains over other data-driven methods. Furthermore, we have strengthened our empirical evaluation by incorporating new results for MiLoRA, demonstrating that EVA remains highly competitive in a crowded field. We believe these clarifications and additions substantially strengthen our submission, and we hope you will reconsider your assessment of our work's contribution. If any questions remain or come up during the rebuttal process, we would be happy to engage in further discussion.
>
> [1] Mangrulkar et al., PEFT: State-of-the-art Parameter-Efficient Fine-Tuning methods, GitHub 2022

---

> > ### Author Response · Authors · 2025-08-05
> >
> > Dear reviewer 6shx,
> >
> > Thank you again for your helpful comments on our paper. We realized that we accidentally overlooked addressing one of your concerns, namely the question: _Why is capturing the variance of the activations or aligning with the gradient the right objective?_
> >
> > Thanks to reviewer sfAD we found an intriguing connection of EVA's initialization to the Neural Tangent Kernel (NTK [1]), which provides additional theoretical proof that EVA is optimal in reducing the generalization error. More concretely, EVA selects a rank-r subspace $\mathcal{V} = \operatorname{span}(v\_1,\ldots, v\_r)$ in which to place LoRA updates such that the **expected gradient magnitude is maximized:** $\mathbb{E}\_{x} \left[ \left\| \nabla\_\theta f(x)^\top v\_i \right\|^2 \right]$. This expectation is **maximized if we choose the top eigenvectors of the activation covariance for initializing the LoRA matrices**, as done in EVA. That is, EVA aligns LoRA directions with the principal components of $\Sigma = \mathbb{E}\_x \left[ \nabla\_\theta f(x) \nabla\_\theta f(x)^\top \right]$.
> >
> > Intriguingly, **this is equivalent to the empirical NTK matrix**. In NTK theory, learning proceeds along the NTK eigenbasis. Let
> > $K \in \mathbb{R}^{n\times n}$ be the empirical NTK matrix over training points $x\_1, \ldots, x\_n$. The NTK view tells us that at infinite width the network output evolves under gradient descent as $f_t(x) = f\_0(x) + K(x, X) K^{-1} (y - f\_0(X)) \cdot \left(1 - e^{-\eta \Lambda t} \right)$, and that **learning decomposes across the eigenfunction of the NTK**. Crucially, **directions with higher NTK eigenvalues $\lambda\_i$ are learned faster and more robustly [2,3].
> >
> > Finally, EVA’s initialization ensures that the LoRA subspace is aligned with the top-r eigenvectors of the NTK. Let ${ u\_i }\_{i=1}^n$ be the eigenvectors of $K$, and $y \in \mathbb{R}^n$ be the labels. Then the squared generalization error is bounded by the residual in low-eigenvalue components: $\mathcal{E}\_{\text{gen}} \lesssim \sum\_{i=r+1}^{n} \frac{(u\_i^\top y)^2}{\lambda\_i^2}$.
> >
> > Or put differently, the projection of the residual error onto non-covered NTK directions (those outside $\mathcal{V}$) determines the generalization error. As EVA initializes the LoRA matrices with the top-r eigenvectors of the NTK, the generalization error is minimized as it is covered only by directions with strictly smaller eigenvalues. Therefore, **maximizing expected gradient signal through data-driven initialization aligns low-rank adaptation directions with the top eigenspaces of the NTK.** Prior work [2,3] shows that **learning concentrated in these eigenspaces leads to faster convergence and lower generalization error.**
> >
> > This finding adds both rigor and theoretical novelty to our work, as already acknowledged by Reviewer sfAD, which is why we have incorporated it into the manuscript. We hope it also addresses your concerns, and we would be happy to provide further clarification if needed.
> >
> > **References**
> >
> > [1] Jacot et al., Neural Tangent Kernel: Convergence and Generalization in Neural Networks, NeurIPS 2018
> >
> > [2] Arora et al., On Exact Computation with an Infinitely Wide Neural Net, NeurIPS 2019
> >
> > [3] Lee et al., Wide Neural Networks of Any Depth Evolve as Linear Models Under Gradient Descent, NeurIPS 2019

---

### Note · Authors · 2025-08-11

Dear SAC, AC, and reviewers,

We would like to thank all of you for the time invested into the review process which provided invaluable feedback that substantially improved our manuscript. In particular we:

- Established a theoretical connection between EVA initialization and the Neural Tangent Kernel (NTK) framework — EVA provably minimizes generalization error by initializing LoRA with the top-r eigenvectors of the NTK. This connection, inspired by reviewer sfAD’s feedback, directly addresses novelty concerns raised by sfAD and 6shx.
- Added MiLoRA as an additional baseline, as requested by 6shx, and found EVA performs on par or better for language generation tasks.
- Addressed additional reviewer comments, including clarifications on theoretical proofs and phrasing, initialization of A/B matrices, and a brief discussion of further related works.

We are grateful for the constructive dialogue during the review process, and we note that the resulting improvements were reflected in several reviewers’ updated scores.
These changes will be included in the camera-ready version if accepted, as they significantly enhance the rigor and clarity of our work.

Thank you again for your time and effort!

Best,
The Authors

---

### Decision · Program_Chairs · 2025-09-17

**Decision:**

Accept (poster)

**Comment:**

This paper proposes Explained Variance Adaptation (EVA), a variant of Low Rank Adaptation (LoRA) for finetuning foundation models. EVA performs incremental SVD on activation vectors to obtain directions that maximize explained variance. These directions are used to initialize LoRA matrices, aiming to maximize the expected gradient signal at the onset of fine-tuning. An adaptive rank allocation mechanism is also proposed based on explained variance ratios. Experiments are conducted on different tasks to demonstrate the effectiveness of the proposed EVA.

Four reviewers reviewed this paper with the overall ratings of 3 borderline accept and 1 borderline reject. The reviewers recognized the contributions of this paper, such as good motivation, extensive experiments, elegant idea, and good presentation. Meanwhile, they also pointed out some critical concerns about different aspects, such as relatively limited novelty, unclear terms, not well-established assumptions, insufficient exploration on higher ranks, and missing related baselines. Thanks to the authors' rebuttal, most of the concerns are addressed, and the reviewers are generally satisfied. The AC recommends acceptance and strongly suggests that the authors include the promised revision in the final version and take the reviewers' comments into consideration.